# MINDCUBE: SPATIAL MENTAL MODELING FROM LIMITED VIEWS

**Qineng Wang**[1][*] **Baiqiao Yin**[1,3][*] **Pingyue Zhang**[1] **Jianshu Zhang**[1] **Kangrui Wang**[1] **Zihan Wang**[1]

**Jieyu Zhang**[4] **Keshigeyan Chandrasegaran**[2] **Han Liu**[1] **Ranjay Krishna**[4] **Saining Xie**[3]

**Jiajun Wu**[2][†] **Li Fei-Fei**[2][†] **Manling Li**[1][†]

[1] Northwestern University  [2] Stanford University  [3] New York University  [4] University of Washington

🌐 Website  ⓞ Code  🤗 Dataset

## ABSTRACT

Can Vision-Language Models (VLMs) imagine the full scene from just a few views, like humans do? Humans form *spatial mental models* naturally, internal representations of *unseen space*, to reason about layout, perspective, and motion. Our MINDCUBE benchmark with $21,154$ questions across $3,268$ images exposes this critical gap, where existing VLMs exhibit near-random performance. Using MINDCUBE, we systematically evaluate how well VLMs build robust spatial mental models through representing positions (cognitive mapping), orientations (perspective-taking), and dynamics (mental simulation for "what-if" movements). We then explore three approaches to help approximate spatial mental models in VLMs, focusing on incorporating unseen intermediate views, natural language reasoning chains, and cognitive maps. The significant improvement comes from a synergistic approach, "map-then-reason", that jointly trains the model to first generate a cognitive map and then reason upon it. By training models to reason over these internal maps, we boosted accuracy from $37.8\%$ to $57.8\%$ ($+20.0\%$). Adding reinforcement learning pushed performance even further to $61.3\%$ ($+23.5\%$). Our key insight is that such scaffolding of spatial mental models, actively constructing and utilizing internal structured spatial representations with flexible reasoning processes, significantly improves understanding of unobservable space.

## 1 INTRODUCTION

For Vision-Language Models (VLMs) (OpenAI, 2024) to move beyond passive perception (Li et al., 2023) to interact with partially observable environments (Yang et al., 2024), it is fundamental to reason about unseen spatial relationships from limited views. Consider how effortlessly a human can infer the layout of a room or the hidden objects behind furniture, all by integrating information from several egocentric observations. For example, given the second viewpoint in Figure 1, human can easily infer the unseen objects behind the "*plant*" are the "*tissue box*" and the "*hand sanitizer*", including their position, pose, and their relationship with objects that are not simultaneously visible. We humans build and update a mental model of our surroundings, even when objects are out of sight. This is enabled by a core cognitive function referred to as **spatial mental model** (Johnson-Laird, 1980; 1983): an internal representation of the environment that allows for consistent understanding and inference about space, independent of the current viewpoint. VLMs, despite their impressive progress, struggle to synthesize spatial information from limited views, maintain spatial consistency across views, and reason about objects not directly visible (Ma et al., 2025a).

This gap calls for specialized evaluation settings, which must include: (a) reasoning with partial observations where objects are occluded or out of view (such as "*hand sanitizer*" in the second viewpoint in Figure 1), (b) maintaining cross-view consistency across shifting viewpoints (such as through anchor objects "*plant*"), and (c) mental simulation to infer hidden spatial relationships (such as "*what if turning left and moving forward*"). To fill this gap, we introduce MINDCUBE, featuring

---

[*]Equal contribution, listed in alphabetical order.
[†]Equal advising.

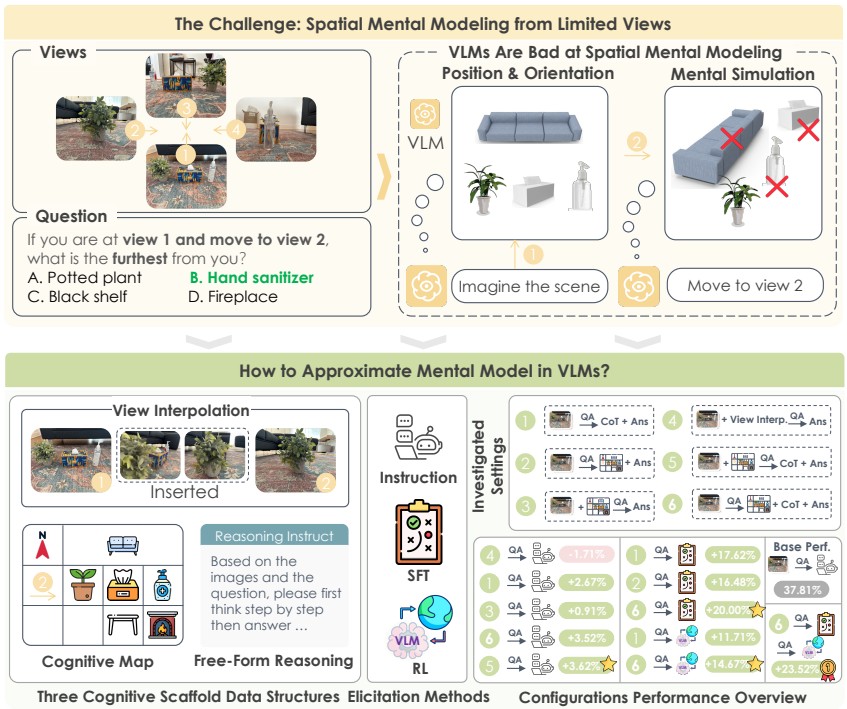

Figure 1: **Top**: VLMs cannot maintain a coherent mental model when evaluating on the MINDCUBE benchmark. **Bottom**: We study how we can help build spatial mental models through external (scaling of views, cognitive map input) and internal strategies (fine-tuning, cognitive map elicitation). We find joint cognitive map and reasoning setting yields the highest gain (+23.52%).

21, 154 questions and 3, 268 images, organized into 976 multi-view groups through various types of viewpoint transformations (i.e., ROTATION, AMONG, AROUND in Figure 2). We annotate questions with a focus on objects that are not visible in the current query view. As shown in Figure 2, we systematically design question types requiring "what-if" mental simulations from the given view (such as "*what if turning to left*"), perspective taking (such as "*what if taking the sofa's perspective*"), complex relation reasoning queries (referencing either the agent or other objects).

Our extensive evaluations of 17 state-of-the-art VLMs on MINDCUBE reveal that both open-weight and closed-source models perform only marginally better than random guessing. This poor performance motivates a central question: **How can we facilitate spatial mental models to reason effectively from partial observations?**

Inspired by spatial cognition (Ramakrishnan et al., 2025; Lee et al., 2025; Zha et al., 2025) operating through *visual imagery*, *linguistic reasoning*, or *explicit cognitive maps*, to build consistent spatial awareness across different views, we investigate three approaches to determine whether intermediate representations can assist approximating spatial mental models in VLMs. **View Interpolation** enhances the input by providing additional views and thereby offering more information using recorded video, which unexpectedly is not helpful, highlighting the importance of reasoning directly from *limited* views. **Free-form Natural Language Reasoning** verbalizes the mental simulation process, achieving performance gains (+2.7%). **Structured Cognitive Map** simulates global spatial memory from an allocentric (bird's-eye) perspective with orientation and view augmentation. Interestingly, providing ground truth cognitive maps directly to answer questions will not yield strong improvements (−5.81%), only actively engaging reasoning with a map achieves strong improvements (+3.62%). Despite the effectiveness of reasoning over maps, building accurate spatial mental models exhibit a significant bottleneck attributed to VLMs' intrinsic ability, evidenced by low Isomorphic Rates (< 10%) with ground truth maps during generation. Recognizing this limitation, we train VLMs by constructing 10, 000 reasoning chains and ground truth cognitive maps, investigating how to effectively guide spatial mental models toward achieving accuracy. While SFT on free-form reasoning chains proved more effective with a gain of +2.8%, guiding models to first build cognitive maps and then perform free-form reasoning over them achieved the best performance, resulting in a total gain of +5.1%, proving scaffolding spatial mental models via

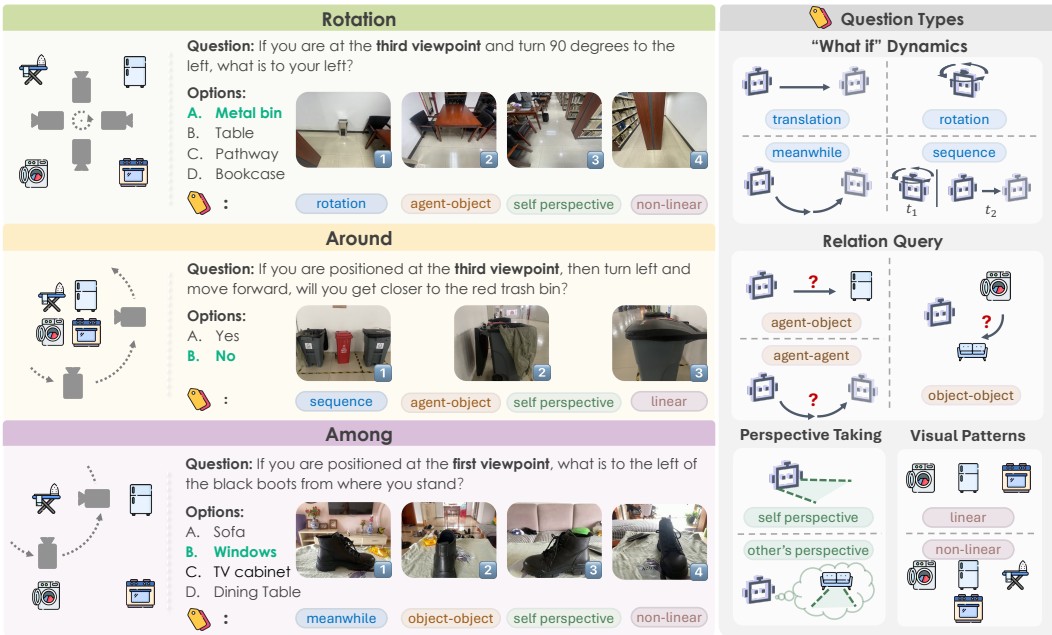

Figure 2: MINDCUBE taxonomy and examples. Left: Three camera movement patterns (ROTATION, AROUND, AMONG) with corresponding spatial QA examples. **Right**: Four-dimensional taxonomy categorizing MINDCUBE questions types.

actively constructing and utilizing internal structured spatial representations with flexible reasoning processes is highly effective. We also use Reinforcement Learning (RL) to further boost post-SFT performance, guiding models to think in terms of building and reasoning over cognitive maps by injecting structured thinking before RL training, using our SFT model. This approach leads to a significant improvement, raising task accuracy from a baseline of $37.8\%$ to $61.3\%$. Our empirical evidence substantiates a critical finding: **autonomously generating and leveraging internal mental representations help VLMs exhibit superior performance in spatial reasoning tasks, as compared to conventional approaches such as view interpolation or externally-supplied maps.**

## 2   MINDCUBE BENCHMARK AND EVALUATION

### 2.1   MINDCUBE BENCHMARK

**Overview.**  We introduce MINDCUBE, a benchmark for evaluating VLMs' spatial reasoning under partial observations and dynamic viewpoints. MINDCUBE features multi-view orthogonal images paired with spatial reasoning questions, enabling fine-grained analysis of spatial mental modeling performance. It targets key challenges such as maintaining object consistency across views and reasoning about occluded or invisible elements.

**Settings.**  MINDCUBE incorporates three distinct settings—**Rotation**, **Around** and **Among** (visualized in left of Figure 2). In the **Rotation** setting, the challenge lies in interpreting multiple orthogonal views from a static and rotational observation point, requiring models to form a holistic understanding of the environment despite only incremental visibility shifts. The **Around** setting leverages occlusion to force VLMs to maintain object permanence even with partial visibility and to convert lateral (left-right) relations in frontal views into depth (front-back) cues in side views. The **Among** setting maintain spatial consistency and overcome visibility constraints as views are captured around a central object with adjacent ones, each view showing the central object positioned before one surrounding element. VLMs need to share information across views, deducing the overall spatial arrangement and relationships even when not all elements are visible simultaneously. Table 1 (left) summarizes the benchmark's overall data distribution. Details on benchmark design about settings and taxonomies and curation are provided in the Appendix B, C and  B.2.2.

Table 1: Left: MINDCUBE data statistics. The number next to the setting (ROTATION, AMONG, AROUND) means the total QA pairs. Numbers next to each dataset (e.g., Arkitscenes) mean QA pairs/image groups. For example, "865/53" for Arkitscenes in ROTATION means 865 QA pairs and 53 image groups from it. Right: Performance of VLMs on MINDCUBE. Dark blue indicates the best result among all models and light blue indicates the second best result among all models.

| Rotation (1081) | |
| --- | --- |
| Arkitscenes | 865/53 |
| Self collected | 216/9 |
| **Img groups** | **62** |

| Among (18204) | |
| --- | --- |
| WildRGB-D | 17500/710 |
| DL3DV-10K | 704/24 |
| **Img groups** | **733** |

| Around (1869) | |
| --- | --- |
| DL3DV-10K | 789/109 |
| Self collected | 1080/71 |
| **Img groups** | **180** |

| Method | Overall | Rotation | Among | Around |
| --- | --- | --- | --- | --- |
| *Baseline* | | | | |
| *Random (chance)* | *32.35* | *36.36* | *32.29* | *30.66* |
| *Random (frequency)* | *33.02* | *38.30* | *32.66* | *35.79* |
| *Open-Weight Multi Image Models* | | | | |
| LLaVA-Onevision-7B Li et al. (2024a) | 47.43 | 36.45 | 48.42 | 44.09 |
| LLaVA-Video-Qwen-7B Zhang et al. (2024d) | 41.96 | 35.71 | 43.55 | 30.12 |
| LongVA-7B Zhang et al. (2024c) | 29.46 | 35.89 | 29.55 | 24.88 |
| mPLUG-Owl3-7B-241101 Ye et al. (2024) | 44.85 | 37.84 | 47.11 | 26.91 |
| InternVL3-8B Zhu et al. (2025) | 37.50 | 26.00 | 42.03 | 36.00 |
| Qwen2.5-VL-7B-Instruct Bai et al. (2025) | 29.26 | 38.76 | 29.50 | 21.35 |
| Qwen2.5-VL-3B-Instruct Bai et al. (2025) | 33.21 | 37.37 | 33.26 | 30.34 |
| DeepSeek-VL2-Small Lu et al. (2024) | 47.62 | 37.00 | 50.38 | 26.91 |
| Gemma-3-12B-it Team et al. (2025) | 46.67 | 38.39 | 48.38 | 34.63 |
| Mantis-8B (SigLip) Jiang et al. (2024) | 41.05 | 37.65 | 40.23 | 50.99 |
| *Proprietary Models* | | | | |
| GPT-5-2025-08-07 OpenAI (2025) | 47.59 | 93.33 | 34.17 | 41.63 |
| Gemini-2.5-pro-2025-06 Team (2025) | 47.05 | 85.50 | 25.95 | 38.40 |
| Claude-4-Sonnet-20250514 Anthropic (2025) | 44.75 | 48.42 | 44.21 | 47.62 |
| *Spatial Models* | | | | |
| RoboBrain Ji et al. (2025) | 37.38 | 35.80 | 38.28 | 29.53 |
| SpaceMantis Chen et al. (2024a) | 22.81 | 37.65 | 21.26 | 29.32 |
| Spatial-MLLM Wu et al. (2025a) | 32.06 | 38.39 | 20.92 | 32.82 |
| Space-Qwen Chen et al. (2024a) | 33.28 | 38.02 | 33.71 | 26.32 |

**Dataset Curation.** The MINDCUBE dataset was created through a pipeline: We first selected multi-view image groups matching our taxonomy's movement patterns (Figure 2) and spatial criteria. These were then annotated with key spatial information. Finally, we algorithmically generated taxonomy-aligned questions with targeted distractors. Details are included in the Appendix B.1.

## 2.2 EVALUATION ON MINDCUBE

We evaluate VLMs' spatial mental modeling abilities on MINDCUBE using a diverse set of models (Table 1, right; setup details in the Appendix C). Results reveal a striking performance gap: the best model, DeepSeek-VL2-Small, achieves only 47.62% accuracy, well above chance but far from human-level C.3. While some models show strength in specific areas—notably GPT-5 in ROTATION (93.33%) and Mantis-8B (SigLip) in AROUND (50.99%)—no single model excels across all categories. We also observe that proprietary models generally outperform the open-source ones. Spatial fine-tuning also yielded varied outcomes without consistently reaching top performance. Overall, neither multi-image input nor spatial fine-tuning reliably improves spatial reasoning, raising a key question: **How can we help VLMs develop or approximate these crucial spatial reasoning capabilities?**

## 3 WHICH SCAFFOLDS BEST GUIDE SPATIAL MENTAL MODELING?

To address the identified gap, we first evaluate whether structured data forms can scaffold spatial reasoning in frozen VLMs by approximating spatial mental models under limited views.

### 3.1 DATA STRUCTURES AS COGNITIVE SCAFFOLDS FOR SPATIAL MENTAL MODELS

We investigate whether certain data structures can act as cognitive scaffolds that help form spatial mental models in VLMs from limited visual observations. In cognitive science, spatial mental models are internal representations encoding the relative configuration of objects and viewpoints. Rather than metric-precise maps, they are schematic, manipulable constructs that support reasoning across fragmented observations and unseen perspectives (Johnson-Laird, 1983; Tversky, 1993; Tversky et al., 1994; Tversky, 2003). For instance, humans can mentally simulate turning or infer what lies behind them, suggesting that such representations are flexible, incomplete, yet functionally

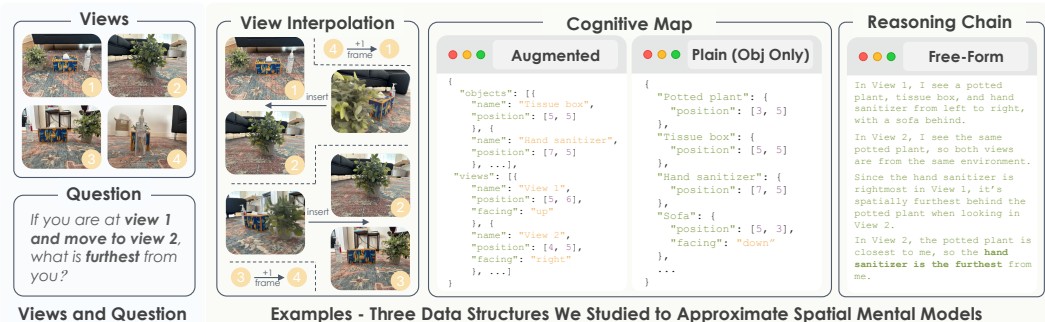

Figure 3: Grounded examples of our three data structures that approximate spatial mental models.

effective. Drawing on this literature, we define three data structures below (detailed introduction can be found in Appendix D.1), each targeting distinct cognitive properties (integration, transformation, inference) of spatial mental models, with grounded examples in Figure 3:

1. **View Interpolation**. Interpolating between sparse views introduces perceptual continuity, echoing the process of *mental animation* (Hegarty, 1992) and supporting internal transformation such as imagined rotation. This structure scaffolds the dynamic updating capability of spatial mental models. Figure 3 shows a one-frame inserting example that replaces the original question images.

2. **Augmented Cognitive Map**. A cognitive map is a 2D schematic representation of object layouts in space. Such maps resemble Tversky's *cognitive collages* (Tversky, 1993), and they capture locally coherent but fragmented structures. Recent studies (Yang et al., 2024; Yeh et al., 2025) on VLM-based spatial intelligence typically adopt a *plain* form that only encodes object positions in a top-down view. We propose an *augmented* variant that incorporates discrete views, with both objects and views annotated by position and orientation, thereby approaching the relational consistency of *spatial mental models*.

3. **Free Form Reasoning**. Open-ended, step-by-step natural language reasoning offers a *procedural approximation* of how spatial models are constructed and queried. While less rigid than map-like structures, such reasoning reflects the inferential function of spatial mental models, especially under ambiguous or incomplete observations (Tversky et al., 1994).

## 3.2 EXPERIMENT SETUP

We conduct controlled experiments with fixed input formats to test whether structured scaffolds can help without retraining. Each condition introduces a different structure to support internal modeling.

**Configurations and Evaluation Metrics.** Each experiment is defined by two orthogonal axes: *Input Structure* (what spatial evidence VLMs receive) and *Output Format* (the required response type). As the experimental foundation of this paper, we begin with the ten possible configurations listed in Table 2, from which we investigate a representative subset. Specifically, our grounded cognitive maps are generated using the object arrangements annotation described in Section 2.1, and examples for all configurations are provided in the Appendix D.3. In the frozen VLMs evaluation setup, we exclude the `Aug-CGMap-Out` and `Plain-CGMap-Out` settings, as VLMs tend to conflate map generation with reasoning, even when instructed otherwise. Beyond evaluating task performance using QA accuracy, we also introduce two well-defined graph metrics for generated cognitive maps: (1) *Overall Similarity*, a weighted score combining directional and facing consistency; and (2) *Isomorphic Rate*, measuring whether all pairwise object relations match the ground truth under optimal alignment. Full definitions are provided in the Appendix D.2.

**Model and Evaluation Data** We conduct all experiments using *Qwen2.5-VL-3B-Instruct* (Bai et al., 2025) with all evaluations performed on MINDCUBE-TINY, a diagnostic subset sampled from MINDCUBE, containing 1,050 questions in total. Detailed statistics are: 600 from AMONG, 250 from AROUND, and 200 from ROTATION.

## 3.3 DO SCAFFOLDS IMPROVE SPATIAL MENTAL MODELING WITHOUT TRAINING?

We evaluate how well the seven input configurations defined in Table 2 support spatial mental modeling in VLMs under limited views, without any model updates. Results are shown in Table 3 (left).

Table 2: Input–output configurations used in all experiments. The suffix "-In" means the cognitive map is given to the model as input, whereas "-Out" means the cognitive map is predicted as an intermediate output before answering. "Aug" indicates maps with object and camera annotations; "Plain" indicates maps without these augmentations. VI = View Interpolation, CGMap = Cognitive Map, FFR = Free-form reasoning. Figure 3 shows visual examples of input structures.

| Name | What the model receives (input) | What the model produces (output) |
|---|---|---|
| Raw-QA | Raw views + question text | Direct answer |
| VI-1 | Raw views + **1** interpolated view + question text | Direct answer |
| VI-2 | Raw views + **2** interpolated views + question text | Direct answer |
| FFR | Raw views + question text | Free-form reasoning → answer |
| Aug-CGMap-In | Augmented cognitive map (objects + camera) + question text | Direct answer |
| Aug-CGMap-Out | Raw views + question text | Augmented cognitive map → answer |
| Plain-CGMap-Out | Raw views + question text | Plain cognitive map → answer |
| Aug-CGMap-FFR-Out | Raw views + question text | Augmented cognitive map + free-form reasoning → answer |
| Plain-CGMap-FFR-Out | Raw views + question text | Plain cognitive map + free-form reasoning → answer |
| CGMap-In-FFR-Out | Augmented cognitive map (objects + camera) + question text | Free-form reasoning → answer |

**How far can structure alone go?** We begin with the baseline: raw input views and direct answering (Raw-QA), which achieves 37.81% accuracy. Adding interpolated views, which we hope to simulate smoother perceptual transitions, leads to no meaningful gain (↑ 0.09%). We include a further analysis on VI in Appendix E.3. Similarly, providing a pre-computed augmented cognitive map as direct input (Aug-CGMap-In) severely degrades performance to 32.00%. In contrast, enabling free-form reasoning (FFR) alone or combined with other settings provides a substantial boost to 41.33%. These results suggest: *structure alone, whether visual or spatial, is not enough.* Without engaging reasoning, VLMs struggle to leverage even well-formed spatial cues to improve spatial mental models.

Table 3: Left: QA accuracy (%) of *Qwen2.5-VL-3B-Instruct* on the MINDCUBE-TINY benchmark under different configs for frozen VLMs. Right: Graph metrics for two cog map output settings.

| Config. | Overall | Rotation | Among | Around |
|---|---|---|---|---|
| Raw-QA | 37.81 | 34.00 | 36.00 | 45.20 |
| VI-1 | 37.90↑ | 35.50 | 37.33 | 41.20 |
| VI-2 | 37.81− | 35.50 | 36.50 | 42.80 |
| Aug-CGMap-In | 32.00↓ | 35.00 | 30.50 | 33.20 |
| FFR | 40.48↑ | 32.00 | 36.00 | 58.00 |
| Aug-CGMap-FFR-Out | 40.57↑ | 21.00 | **43.00** | 50.40 |
| Plain-CGMap-FFR-Out | 41.33↑ | 25.00 | 39.67 | **58.40** |
| CGMap-In-FFR-Out | **41.43**↑ | **37.00** | 41.67 | 44.40 |

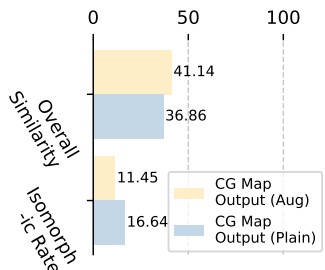

**Can we prompt the model to think spatially?** The answer appears to be yes. Prompting the model to generate a cognitive map (Aug-CGMap-FFR-Out, Plain-CGMap-FFR-Out) before answering leads to further improvements over free-form reasoning alone (FFR) from 40.48% to 41.43%. This suggests that generating a map may encourage the model to first form a global understanding of the scene, which in turn supports more structured reasoning. Both map forms have a great format-following ability, yet fail to generate accurate maps. Overall, augmented maps perform worse. In Table 3 (Right), despite generating syntactically valid maps for both formats, similarity to grounded maps is low (< 50%), reflecting limited mapping ability. Notably, both augmented and plain maps have low isomorphism rates (0.10%, 7.43%). The reason that the isomorphic rate for augmented map setting is nearly zero is likely because the added view-level details increase generation errors. Detailed case examples can be found in the Appendix E.

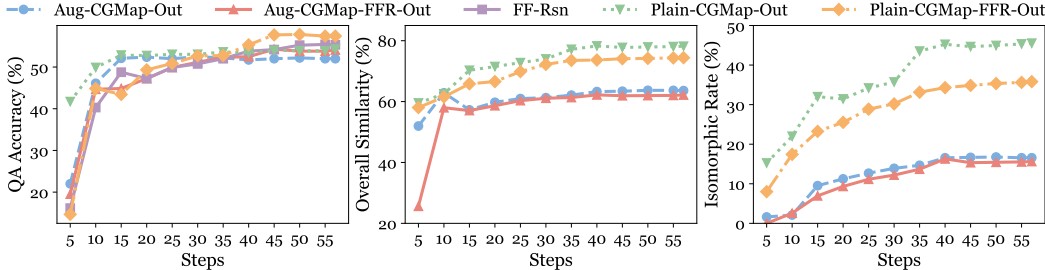

Figure 4: SFT per 5 step training performance on task accuracy and graph metrics.

> 💡 **Key Takeaways: Scaffolding Spatial Mental Models in *Frozen VLMs***
> - *Explicit reasoning is crucial for improving performance.*
> - *Reasoning acts as a necessary mechanism to ground spatial structure in frozen settings.*
> - *Passive structures (like maps as input) alone and visual continuity offer little benefit.*

## 4 CAN WE TRAIN FOR THE EMERGENCE OF SPATIAL MENTAL MODELS VIA VLMS' USE OF SCAFFOLDS?

So far, prompting frozen VLMs with external scaffolds, such as interpolated views or cognitive maps, has yielded limited gains. These techniques fail to tackle the core limitation: VLMs do not form internal spatial representations or reason through space effectively. To go further, we want to know: Can supervised fine-tuning (SFT) and Reinforcement learning (RL) teach VLMs to build and leverage spatial mental models from within?

### 4.1 DESIGNING A ROBUST EXPERIMENTAL FRAMEWORK

To ensure consistency and comparability, we inherit experimental configurations detailed in Sections 3.1 and 3.2. Specifically, we retain: (1) the two effective data scaffolds—Cognitive Maps (Object-only / Object + Camera) and Free-Form Reasoning, (2) the base model *Qwen2.5-VL-3B-Instruct*, (3) the evaluation benchmark MINDCUBE-TINY, and (4) all established evaluation metrics. View interpolation is excluded due to its limited performance gains in earlier validations.

**SFT Task Configurations.** Drawing on insights from Section 3.3, we use selected configurations from Table 2 to evaluate the incremental impact of cognitive map generation and free-form reasoning in SFT. These include baseline QA without explicit reasoning (`Raw-QA`), reasoning guided by generated maps only (`Plain-CGMap-Out`, `Aug-CGMap-Out`), reasoning-augmented prompts (`FFR`), and a fully integrated setup that asks VLMs to generate both maps and reasoning (`Aug-CGMap-FFR-Out` and `Plain-CGMap-FFR-Out`).

**RL Task Configurations and Reward Design.** We employ the VAGEN framework (Wang* et al., 2025) for VLM policy optimization, using Group Relative Policy Optimization (GRPO) (Shao et al., 2024) as our core algorithm. We evaluate RL variants along two axes: the output format (`FFR-only` vs. `CGMap-FFR`) and the initialization (from scratch vs. from the best SFT checkpoint), yielding six configurations in total (Table 4). Detailed settings can be found in the Appendix G.1.

**Grounded Cognitive Maps and Free-Form Reasoning Chain.** Grounded cognitive maps are not only used as the input in the `Aug-CGMap-In` and `CGMap-In-FFR-Out` setting for the frozen VLMs in the Section 3.2, but also as the training and comparison data. We curate such grounded cognitive maps through a template-based method, where we always select the front image in our annotation as the "up" direction. We also manually constructed grounded reasoning chains using detailed image annotations and structured question templates, ensuring logical coherence and clear grounding in observable spatial relations (see an example in Figure 3). The detailed grounded cognitive maps and reasoning data generation pipelines are shown in the Appendix F.1.1 and F.1.2. We also evaluate the effect of removing viewpoint descriptors from the question text in Appendix F.8, confirming that the map-then-reason advantage holds even without textual directional cues.

## 4.2 DO THE EMERGENCE OF SPATIAL MENTAL MODELS BENEFIT FROM TRAINING?

We explore several SFT configurations (results shown in Table 4), guided by a series of core questions. Fine-tuning directly on raw QA pairs, without spatial supervision, raises accuracy from 37.81% to 52.67%. This suggests VLMs can absorb some spatial cues from QA data alone. We use this setup as the baseline for evaluating methods that explicitly incorporate spatial structures. Primary modifications in SFT phase include adjusted training hyperparameters (detailed in the Appendix F.2) and the input-output configurations.

Table 4: QA accuracy (%) and cognitive map generation quality of *Qwen2.5-VL-3B-Instruct* under both SFT and RL on MINDCUBE-TINY. FFR refers to free-form reasoning. Bolded means the best within that training category (SFT or RL).

| | Config. | MINDCUBE-TINY QA Accuracy (%) | | | | Generated Cognitive Map (%) | |
|---|---|---|---|---|---|---|---|
| | | Overall | Rotation | Among | Around | Overall Sim. | Isom. Rate |
| SFT | Raw-QA | 52.67 | 34.50 | 52.50 | 67.60 | – | – |
| | FFR | 55.43↑ | 36.00 | 57.17 | 66.80 | – | – |
| | Aug-CGMap-Out | 52.48 | 30.00 | 52.17 | 71.20 | 61.28 | 13.90 |
| | Plain-CGMap-Out | 54.29↑ | 32.00 | 53.67 | **73.60** | 78.18 | **45.52** |
| | Aug-CGMap-FFR-Out | 54.29↑ | **41.50** | 52.33 | 69.20 | 61.92 | 15.33 |
| | Plain-CGMap-FFR-Out | **57.81↑** | 36.50 | **61.17** | 66.80 | 74.18 | 35.33 |
| RL | RL-FFR (from scratch) | 49.52 | 26.50 | 51.50 | 63.20 | – | – |
| | RL-Aug-CGMap-FFR-Out (from scratch) | 52.48 | **36.00** | 51.50 | 68.00 | 55.71 | 0.00 |
| | RL-Plain-CGMap-FFR-Out (from scratch) | 50.86 | 34.00 | 50.50 | 65.20 | 29.59 | 6.67 |
| | RL-FFR (from SFT) | 59.14 | 31.50 | 66.00 | 64.80 | – | – |
| | RL-Aug-CGMap-FFR-Out (from SFT) | 60.86 | **36.00** | 66.00 | **68.40** | 62.48 | 16.95 |
| | RL-Plain-CGMap-FFR-Out (from SFT) | **61.33** | 29.50 | **69.17** | 68.00 | **73.36** | 35.33 |

**Can structured approximations of mental models alone meaningfully improve performance?**
As shown in Table 4, supervised fine-tuning on explicit cognitive maps, either *Augmented* or *Plain*, leads to substantial improvements in graph structure quality. However, the effect on end-task accuracy remains limited. Aug-CGMap-Out (52.48%) shows no improvement over Raw-QA (52.67%), while Plain-CGMap-Out (54.29%) offers only a modest gain. FFR alone yields a moderate gain (55.43%), yet still falls short of the joint approach. This means that a scaffold alone is not sufficient to automatically translate into performance gains.

**Generating both cognitive maps and free-form reasoning is the most effective approximation.** Among all configurations, the combination of generating a plain map and then reasoning (Plain-CGMap-FFR-Out) yields performance gain (↑5.14% compared to Raw QA-SFT), surpassing models that rely on only map generation or reasoning alone. This suggests a synergy between structured spatial modeling and natural language inference. The training dynamics reveal a crucial trade-off that explains this synergy. As shown in Figure 4 (b, c), models trained solely on map generation (Plain-CGMap-Out) learn the target structure very rapidly, quickly reaching high similarity and isomorphism. However, their QA accuracy soon plateaus (Figure 4a), suggesting the model learns the structure without fully grasping its functional utility. In contrast, the top-performing Plain-CGMap-FFR-Out model learns the map structure more slowly and never reaches the same level of structural perfection. Yet, its QA accuracy continues to increase and surpass all other configurations. This suggests that the joint pressure of the reasoning task forces the model not just to replicate a structure, but to build a functionally effective spatial representation, which can lead to improvement for overall spatial understanding despite being imperfect.

> 💡 **Key Takeaways: Explicit Training for the Emergence of Spatial Mental Models**
> - *Joint cogmap and reasoning setting yields optimal performance through synergistic effects.*
> - *Neither map generation nor reasoning alone matches the performance of the joint approach.*

## 4.3 CAN REINFORCEMENT LEARNING FURTHER REFINE SPATIAL MENTAL MODELS?

While SFT establishes a strong baseline for spatial mental modeling, emerging evidence from models like DeepSeek R1 (Guo et al., 2025) suggests reinforcement learning (RL) can offer additional gains by optimizing behavior through outcome-driven feedback. We ask: Can reward-guided refinement help VLMs build sharper spatial models and reason more effectively?

RL lets a model *feel* the consequences of its spatial thoughts through reward, but does that feedback alone forge a genuine "mental map", or must we first teach the model what a map looks like? Table 4 summarizes key settings and answers this question in two parts.

**RL in a vacuum is not enough.** Training from scratch with sparse rewards provides insufficient guidance for building robust spatial representations. When asked to produce free-form reasoning (`RL-FFR (from scratch)`), the model achieves only 49.52% overall accuracy. This result, while an improvement over initial baselines, confirms that task-level rewards alone are too unstructured to effectively teach spatial abstraction.

**Structured outputs provide modest benefits when learned from scratch.** Introducing a cognitive map structure provides only marginal improvement (`RL-Aug-CGMap-FFR-Out`: 52.48%, `RL-Plain-CGMap-FFR-Out`: 50.86%). In both cases, the model fails to learn meaningful geometry, with low similarity scores and near-zero isomorphism rates. This suggests that without a prior concept of a "good" map, RL struggles to exploit the provided structural format, even if it can learn to fill it out validly.

**RL performs better when it trains from SFT checkpoint.** The most substantial improvements occur when warm-starting RL from an optimal SFT checkpoint. All three from-SFT configurations significantly outperform their from-scratch counterparts, with `RL-Plain-CGMap-FFR-Out (from SFT)` achieving the highest accuracy of 61.33% (↑3.52% over the best SFT model, ↑8.85% over the best RL-from-scratch). Notably, even `RL-FFR (from SFT)` reaches 59.14%, confirming that SFT initialization is critical. However, the map-then-reason configurations consistently outperform FFR-only, reinforcing the advantage of structured spatial scaffolding. The `Plain-CGMap` variant continues to produce geometrically superior maps (35.33% vs. 16.95% isomorphism rate), suggesting that simpler map formats allow RL to better preserve spatial structure. These results indicate that RL's primary role is polishing and refining the strong priors learned during SFT, and raising the performance ceiling beyond what SFT alone can achieve.

> 💡 **Key Takeaways: Reinforcement Learning for the Emergence of Spatial Mental Models**
> - *Combining cognitive maps with reasoning consistently improves all learning outcomes.*
> - *Starting from scratch, RL provides only marginal gains for spatial reasoning; its true power is unlocked when building upon a strong SFT foundation.*

### 4.4 EFFECT OF OBJECT PRESENTATION ORDER IN COGNITIVE MAP SUPERVISION

When constructing cognitive map supervision, the order in which objects are listed in the textual map description is a design choice that may influence learning dynamics. We investigate two settings: (1) **Fixed Spatial Order**, where objects follow a consistent spatial convention (e.g., clockwise from the camera's viewpoint), and (2) **Randomized Order**, where the object sequence is shuffled independently for each training example. We examine this factor for both SFT and RL, since the SFT checkpoint also serves as the initialization for RL training. All primary results reported in Table 4 use randomized order.

Table 5: SFT and RL results under fixed spatial object order. The map-then-reason approach (`Plain-CGMap-FFR-Out`) remains the best-performing configuration, consistent with the randomized-order results in Table 4.

| | Config. | Overall | Rotation | Among | Around | Overall Sim. | Isom. Rate |
|---|---|---|---|---|---|---|---|
| **SFT** | Raw-QA | 52.28 | 34.50 | 52.50 | 66.00 | – | – |
| | FFR | 53.52↑ | 36.00 | 54.67 | 64.80 | – | – |
| | Aug-CGMap-Out | 54.19↑ | 35.50 | 53.17 | **71.60** | 74.30 | 43.24 |
| | Plain-CGMap-Out | 54.38↑ | 35.50 | 53.50 | **71.60** | **91.73** | **89.05** |
| | Aug-CGMap-FFR-Out | 55.24↑ | **49.50** | 52.50 | 66.40 | 75.27 | 46.00 |
| | Plain-CGMap-FFR-Out | **60.76↑** | 47.50 | **62.33** | 67.60 | 88.79 | 73.81 |
| **RL** | RL-Aug-CGMap-FFR-Out (from SFT) | **70.67** | **53.00** | 76.83 | **70.00** | 85.53 | 58.86 |
| | RL-Plain-CGMap-FFR-Out (from SFT) | **70.67** | 48.00 | **79.17** | 68.40 | **85.79** | **71.52** |

**Both settings exhibit consistent trends.** As shown in Table 5 and Table 4, the relative ranking of configurations is preserved regardless of object order: `Plain-CGMap-FFR-Out` consistently achieves the highest QA accuracy in SFT, and RL from SFT continues to yield the strongest overall results. The training dynamics (Figure 4 vs. Figure 5) further confirm that in both settings, mod-

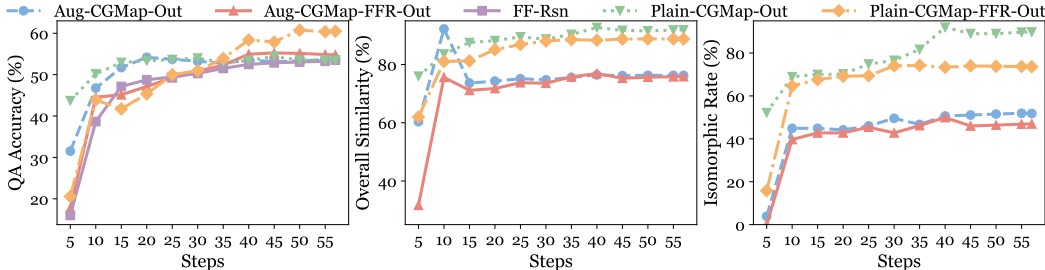

Figure 5: SFT training dynamics under fixed spatial object order. Compared to randomized order (Figure 4), the overall learning trends are consistent: `Plain-CGMap-FFR-Out` achieves the highest QA accuracy despite not producing the most structurally perfect maps.

els trained solely on map generation learn the target structure rapidly but plateau in QA accuracy, while the joint map-and-reasoning model learns maps slowly yet continues to improve on the end task. We further discuss why randomized order better evaluates genuine spatial understanding in Appendix F.9.

## 5  RELATED WORKS

**Spatial Cognition.** Spatial cognition encompasses skills like mental rotation, spatial visualization, and object assembly, essential for perceiving and manipulating spatial relationships in both 2D and 3D environments (Xu et al., 2025b; Zha et al., 2025; Wang et al., 2025). At the core of these abilities are Spatial Mental Models (SMMs) (Johnson-Laird, 1980; 1983), which are internal representations that allow for consistent understanding of space. Recently, much effort has been dedicated to evaluating spatial cognition in VLMs (Zhan et al., 2025; Ma et al., 2025a; Lee et al., 2025; Zhang et al., 2025). Moreover, some methods are proposed to enhance spatial understanding, such as coordinate-aware prompting (Cai et al., 2024), CoT reasoning (Ma et al., 2025b; Liu et al., 2025b), explicit spatial representation alignment (Cheng et al., 2024; Chen et al., 2024a), and an RL-based approach (Pan & Liu, 2025). However, existing benchmarks (Lee et al., 2025; Zhan et al., 2025; Chen et al., 2025; Qi et al., 2025; Zhang et al., 2025; Ma et al., 2025a; Ramakrishnan et al., 2025; Tang et al., 2025b; Fu et al., 2024; Yang et al., 2024; Zhang et al., 2024a) and approaches often neglect the mental-level spatial reasoning that underpins human cognition, leaving a gap between machine and human capabilities. To bridge this gap, a new approach is needed that trains VLMs to reason about space not only through visual data but also through mental-level spatial reasoning, aligning more closely with human spatial cognition.

**Multi Views understanding.** Multiview spatial understanding leverages multiple viewpoints to reconstruct 3D structures and overcome single-view limitations. Efficient techniques optimize view processing, while reconstruction methods (Wang et al., 2025; Liu et al., 2025a; Fu et al., 2025; Qu et al., 2025), view synthesis methods (Sun et al., 2018; Zhang et al., 2024e; Sargent et al., 2023) and multiview equivariant learning (You et al., 2024) enhance geometric consistency. Topological representations like Zhang et al. (2024b) encode object relations for holistic reasoning, while frameworks such as Hong et al. (2023) advance open-vocabulary concept learning from multiview data via neural fields and vision-language fusion. LMMs augmented with multiview inputs (Daxberger et al., 2025; Wu et al., 2025a; Fan et al., 2025; Zheng et al., 2025; Lee et al., 2025; Zhao et al., 2025; Xu et al., 2025a) demonstrate marked improvements in spatial tasks like geometric understanding and perspective taking. Yet, they struggle with multiview consistency understanding due to fragmented reasoning and 2D-to-3D projection ambiguities, leaving a gap for robust spatial AI.

## 6  CONCLUSION

We introduced MINDCUBE to study how VLMs can approximate spatial mental models from limited views, a core cognitive ability for reasoning in partially observable environments. Moving beyond benchmarking, we explored *how* internal representations can be scaffolded through structured data and reasoning. Our key finding is that *constructing and reasoning over self-generated cognitive maps*, rather than relying on view interpolation or externally provided maps, yields the most effective approximation of spatial mental models across all elicitation methods (input-output configurations, supervised fine-tuning, and reinforcement learning). Initializing RL from a well-trained SFT checkpoint further optimizes the process, further improving spatial reasoning performance.

## ETHICS STATEMENT

The MINDCUBE benchmark was developed using a combination of publicly available, anonymized datasets (ArkitScenes, WildRGB-D, DL3DV-10K) and self-collected imagery. For our self-collected data, care was taken to capture indoor and outdoor scenes without including personally identifiable information (PII) or sensitive content. All human annotators involved in the data curation and evaluation phases were compensated at rates significantly exceeding their local minimum wage.

We acknowledge several limitations and ethical considerations. The datasets used, while diverse, may not fully represent the vast range of global environments, potentially introducing geographic or cultural biases into the model's spatial understanding. Furthermore, the training, fine-tuning, and evaluation of the large-scale Vision-Language Models discussed in this paper carry a significant computational and environmental cost. While our research is intended to advance the scientific understanding of AI cognition, we recognize that technologies enhancing spatial reasoning in machines could have dual-use applications.

## REPRODUCIBILITY STATEMENT

To ensure the reproducibility of our findings, we have included our complete codebase for data processing, model training, and evaluation in the supplementary materials as a .zip archive. Furthermore, the full MINDCUBE benchmark, encompassing all of our training data, test data, annotations, and evaluation protocols, will be released in a public repository to facilitate further research and verification by the community.

### ACKNOWLEDGMENTS

This work is in part supported by the Stanford Institute for Human-Centered AI (HAI), ONR N00014-23-1-2355, ONR MURI N00014-22-1-2740, ONR MURI N00014-21-1-2801, and DSO National Laboratories Agreement DSOCO25017.

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

# Appendix

## Table of Contents

## A    THE USE OF LARGE LANGUAGE MODELS

We used large language models (LLMs), including Google's Gemini 2.5 Pro and OpenAI's GPT-5, as auxiliary tools to assist with writing, editing, and conducting the literature review for this manuscript. All content was critically reviewed, fact-checked, and revised by the human authors to ensure its scientific validity and originality. The authors are fully responsible for all statements and conclusions presented in this paper. Specifically, we use LLMs for polishing our wording and writing, and we use LLMs to retrieve several related works.

## B    MINDCUBE BENCHMARK

### B.1    DETAILS FOR DATA COLLECTION AND ANNOTATION

**Image Collection and Selection.** Our MINDCUBE benchmark comprises $3,268$ images (2,302 indoor/outdoor images from publicly released dataset and 400 self-collected images), where we implement a comprehensive image selection methodology encompassing four distinct view dynamics, incorporating various data sources and processing procedures, as shown in Fig.2.

For rotation view dynamics, we implement a three-stage filtering strategy to extract meaningful camera trajectories and key frames from ArkitScenes Baruch et al. (2021) dataset.

In the first stage, we analyze the top-down view of camera poses within each scene to identify two types of trajectories: linear paths and small rotational arcs. A linear trajectory is characterized by consistently oriented cameras exhibiting significant displacement perpendicular to their viewing direction. A rotational arc trajectory is identified when three to four camera positions demonstrate approximately 90-degree relative orientation changes while being distributed along an approximate circular arc. The second stage focuses on selecting two critical frames from the previously identified

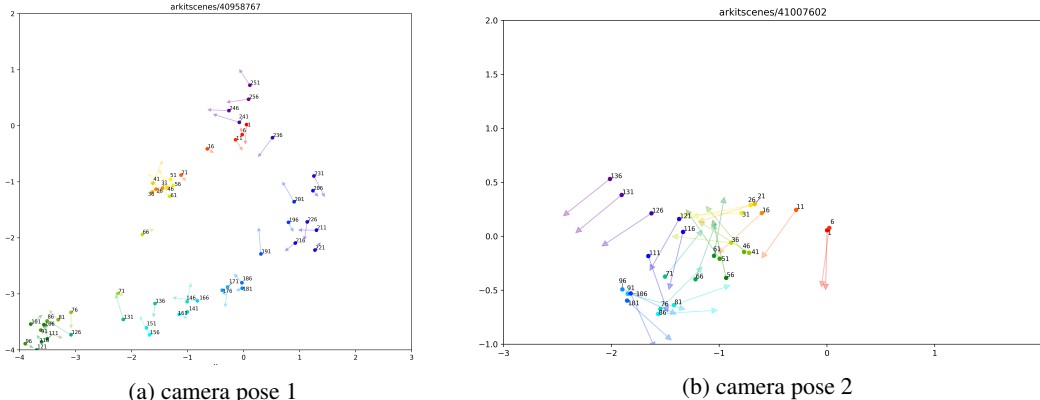

(a) camera pose 1                                        (b) camera pose 2

Figure 1: Examples of camera poses in ArkitScenes

translation segments. The selection criteria mandate that: (1) the camera movement direction must be parallel to the object arrangement direction, (2) this movement should be aligned with the horizontal axis, (3) the first frame should only capture objects A and B, while the second frame should only capture objects B and C, and (4) both frames must be free from motion blur and exhibit clear object visibility.

The third stage processes the rotation segments to extract three or four key frames. These frames must satisfy several conditions: (1) the camera positions should appear to originate from a stationary rotating camera, even if slight circular movement exists, (2) the camera orientations should align with standard cardinal directions (approximately 90 degrees apart), and (3) each frame should contain no more than three semantically distinct primary objects that occupy over 50% of the frame area relative to the background.

For among view dynamics, image groups are manually selected from DL3DV-10KLing et al. (2023) and WildRGB-DXia et al. (2024) datasets. We employ a single-stage selection process to identify

four key frames representing cardinal viewpoints (front, left, right, and back) from 360-degree scene captures. The selection criteria are: (1) camera orientations must align with standard directions, ensuring that the central object, its background objects, and the camera's line of sight are collinear and parallel or perpendicular to standard scene elements such as tables or walls, (2) we reject sets where three or more frames share identical semantic background information, and (3) we discard sets where three or more frames have severely occluded background objects that cannot be reconstructed from information in the other frames.

For around view dynamics, image groups are manually curated from the DL3DV-10KLing et al. (2023) dataset and assigned sequential identifiers. The front view (designated as view 1) must provide clear visibility of all relevant information. This view is established as the reference point for subsequent views in the sequence.

This structured approach to image selection and processing yields a rich dataset that supports subsequent model training and testing procedures. The methodology ensures comprehensive coverage of spatial relationships, occlusion states, and view-dependent object characteristics across multiple viewing scenarios.

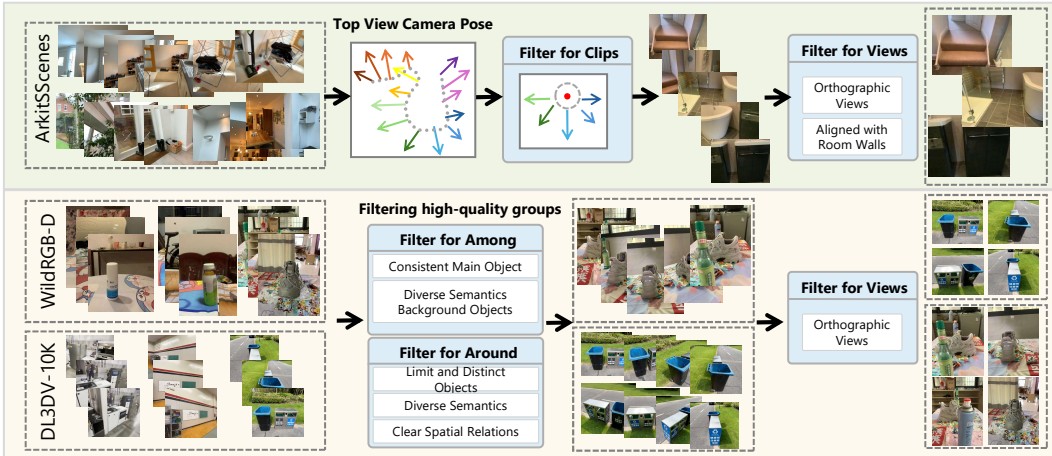

Figure 2: MINDCUBE Bench construction pipeline.

**Data Annotation.** After collecting and filtering the images, we follow a two-phase paradigm for annotation: We establish a systematic image annotation protocol to ensure data consistency and accuracy. The annotation framework encompasses four key dimensions: spatial relationship identification, object grouping rules, semantic orientation determination, and occlusion level assessment. We provide a pdf of the annotation interface in the supplementary material.

Regarding spatial relationship identification, annotators are required to identify primary object entities within images and determine their spatial relationships. These relationships are primarily categorized into two types: front-back relationships typically involving two primary objects, with priority given to objects directly behind as key entities; and left-right relationships encompassing two to four primary objects, where adjacent objects with front-back relationships can be considered as a unified entity.

To enhance annotation efficiency and semantic completeness, this study introduces object grouping rules. Multiple objects can be annotated as a unified entity when they collectively form clear spatial relationships with other primary objects. Each object may include attribute descriptors (e.g., color, material) to enhance semantic expression. Combined object entities must maintain distinct spatial relationships with other primary objects.

For objects with definitive semantic fronts, the following information must be recorded: the object's inherent semantic front, the object's orientation relative to the current viewpoint (aligned, reversed, leftward, rightward, etc.), and the object's actual projected direction within the scene.

Occlusion levels are evaluated using a four-tier classification system: complete occlusion where the object is entirely invisible from the current viewpoint; major occlusion where primary object features are difficult to identify; minor occlusion where primary object features remain identifiable; and no occlusion where the object is fully visible. For cases of complete occlusion, the annotation system provides multi-view scene images, ensuring object visibility in at least one viewpoint to support subsequent cross-view question-answering system training.

This annotation protocol provides a structured semantic foundation for subsequent automated question-answer pair generation while ensuring data quality and consistency. Through this standardized annotation process, we effectively capture key information including spatial relationships, compositional features, semantic orientations, and occlusion states of objects within scenes.

**Examples for automatic QA generation pipeline.** Our automatic QA generation pipeline gener-

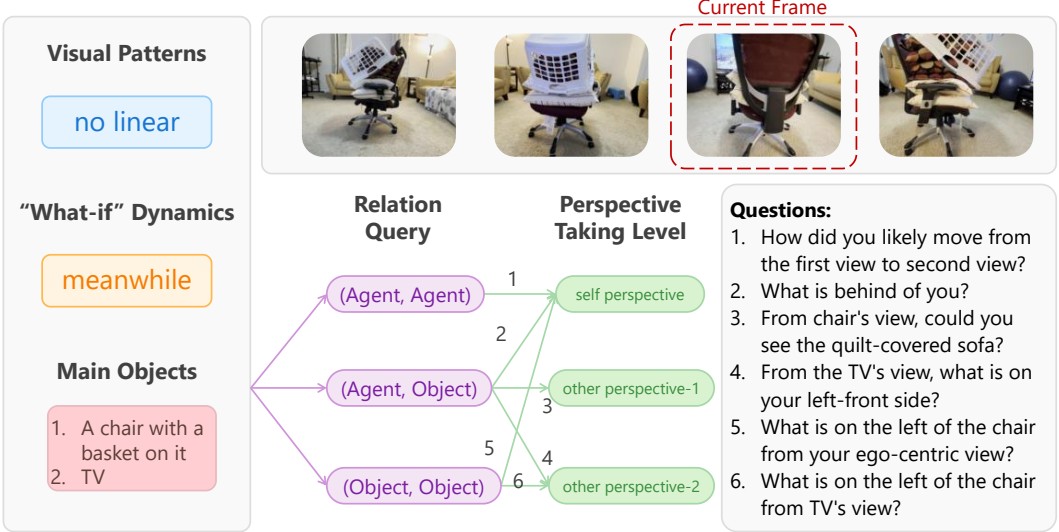

Figure 3: Example of different question-related label combinations to generate QA pairs.

ates different types of questions using combinations of labels. Each question's label combination is encoded in its ID (e.g., "among_group001_q1_1_1"), while the original object and label information is preserved in the meta_info field to track the context of question generation.

## B.2 DETAILS OF OUR MINDCUBE BENCHMARK

### B.2.1 THREE KINDS OF INVISIBILITY SETTINGS

**Rotation.** In this setting, our camera remains stationary while rotating in place, capturing 2 to 4 orthogonal views. In each view, a central object remains visible in the foreground, while all views maintain equal importance in the spatial representation.

We evaluate models' understanding of spatial invisibility by asking questions such as 'When positioned at a particular viewpoint, what should be to your left or right (given that each view only reveals what's directly ahead)?' or 'After rotating a quarter or half turn, what objects would be in front of you, to your left, behind you, or to your right?' We expect models to construct a comprehensive spatial understanding by leveraging the **sequential nature of the views and consistent spatial cues** across images (such as lighting direction), thereby demonstrating their ability to reason about the complete environment despite only having access to partial visual information from each viewpoint.

**Around.** In this setting, we leverage **occlusion** phenomena to force MLLMs beyond simple 2D spatial recognition. When viewing objects from different angles, some objects become partially or fully hidden, requiring models to:

- Maintain object permanence despite partial visibility

- Transform lateral relationships (left-right) from frontal views into depth relationships (front-back) for side views
- Integrate spatial information across multiple viewpoints to form a coherent 3D understanding

This approach prevents models from relying solely on direct visual cues and instead necessitates true 3D spatial reasoning by combining information from multiple perspectives.

**Among.** In this setting, the camera rotates around a central object, positioned between this central object and several surrounding objects. Four orthogonal views are captured, with each view showing the central object positioned in front of one of the surrounding objects.

This setup creates interesting visibility constraints across different perspectives. For instance, a surrounding object visible in one view may be invisible in another view because of the constraints imposed by the camera's field of view. Through establishing consistency relationships between these views, we can infer the relative positions of objects not directly visible from certain perspectives. When an object is not visible from a particular viewpoint, consistency and spatial reasoning can determine its position relative to the central object.

All views hold equal status in this framework, allowing for bidirectional establishment of invisibility relationships. This creates a coherent spatial reasoning system where information from each perspective contributes to a complete understanding of the three-dimensional arrangement, even when direct visual confirmation is unavailable from certain angles.

### B.2.2 LABEL TAXONOMY

We use image related labels for better analysis and question related labels for automatic QA generation with different label combinations.

**Visual Patterns.** In our taxonomy of spatial configurations, we classify visual patternss into distinct categories based on their geometric relationships. Linear arrangements refer to configurations where objects are positioned along a single axis, forming a collinear pattern. Non-linear arrangements, conversely, are characterized by objects positioned such that the connecting lines between adjacent pairs form 90-degree angles, creating rectilinear patterns. This binary classification serves as a fundamental attribute in our spatial relationship labeling scheme, enabling precise description and analysis of scene compositions across various domains.

**"What if" Dynamics.** "What if" Dynamics refers to the model's capability to comprehend and reason about dynamic perspective changes occurring within images or posed questions. We conceptualize viewpoint transitions as combinations of translation and rotation operations, resulting in four distinct categories:

- Pure Translation: Cases where the viewpoint undergoes only translational movement without rotational change.
- Pure Rotation: Scenarios involving rotational transformation of the viewpoint while maintaining its positional coordinates.
- Simultaneous Translation-Rotation(Meanwhile): Instances where both translational and rotational operations occur concurrently.
- Sequential Translation-Rotation(Sequence): Cases where translation and rotation occur in sequence rather than simultaneously. Notably, in our dataset, this category is uniquely represented through textual descriptions in the questions rather than through explicit visual transformations.

The first three categories of "What if" dynamics are visually demonstrated through changes in view representation, while the sequential category requires models to interpret text-based descriptions of perspective changes. This taxonomy provides a systematic framework for evaluating spatial reasoning capabilities across diverse viewpoint transformation scenarios.

**Relation Query.** We define three distinct categories of relation queries that capture the fundamental nature of spatial reasoning tasks:

- Agent-Agent: This pattern involves self-referential spatial positioning, where the observer must evaluate and potentially adjust their own position in space. It requires egocentric spatial reasoning and self-awareness of one's location relative to environmental constraints.

- Agent-Object: This pattern focuses on determining the orientation of an observed object relative to the observer's position. Unlike the P-P pattern, the emphasis here is on object perception rather than self-positioning, requiring the observer to make judgments about external entities while maintaining awareness of their own reference frame.

- Object-Object: This pattern involves reasoning about the spatial relationship between two discrete objects in the environment, independent of the observer's position. This allocentric spatial reasoning requires understanding relative positioning, distance, and orientation between entities without necessarily using oneself as a reference point.

These categorizations provide a structured approach to analyzing the cognitive demands of different spatial reasoning tasks and can inform both the design of spatial question answering systems and the evaluation of human spatial cognition abilities.

**Perspective Taking.** We propose a label called "Perspective Taking" that categorizes the complexity of viewpoint projection. This label distinguishes between three increasingly sophisticated levels of perspective reasoning:

- Self Perspective: Reasoning based on the current camera view or the observer's own viewpoint. This represents the baseline where no perspective shift is required.

- Other's Perspective Taking-1: The ability to determine visibility relationships from another agent's viewpoint. This involves understanding what objects are visible or occluded from a different viewpoint (e.g., determining whether a specific object is within the field of view of another camera). The another agent's viewpoint is usually determined by an object with a clear orientation in the image.

- Other's Perspective Taking-2: The ability to understand how spatial relationships transform when viewed from another agent's perspective. This more advanced capability requires mental rotation and spatial transformation to reason about relative positions (e.g., determining whether, from another viewpoint, object X appears to be positioned behind object Y).

This classification aligns with developmental psychology research on perspective-taking abilities, where Level-1 perspective taking typically develops earlier than the more cognitively demanding Level-2 perspective taking.

We provide performance across different categories and labels in Table 1 and 2. Upon detailed analysis of model performance across various capabilities, certain trends emerge. The O-O (Object-Object) task within Relation Pattern also demonstrates generally lower scores across the board, suggesting it is a less tractable problem for current models. Notably, InternVL2-8B struggles with the sequence task, exhibiting the lowest score among all evaluated models in that category.

Regarding model stability, Mantis(SigLip) demonstrates robust performance in both Object Arrangement and Relation Pattern sections, indicating a consistent capability in these spatial reasoning tasks. Similarly, Qwen2.5-VL-7B-Instruct maintains relatively stable performance within Viewpoint Dynamics. In contrast, InternVL2-8B shows a broader instability, with consistently lower overall scores and considerable performance fluctuations across different sub-categories, highlighting areas for further improvement in its generalizability and robustness.

## B.3 EXAMPLES

We show some examples in Figure 5, 6 and 4.

Table 1: Performance of VLMs on MINDCUBE across categories.(Part 1)

| Model | Overall | Object Arrangement | | Perspective Taking | | |
|---|---|---|---|---|---|---|
| | | Linear | Perp. | Self | Level1 | Level2 |
| LLaVA-Video-7B-Qwen2 | 41.96 | 30.12 | 43.11 | 42.19 | 60.76 | 33.80 |
| Mantis(SigLip) | 41.04 | **50.99** | 40.08 | 41.20 | 54.43 | 35.41 |
| GPT-4o | 38.81 | 29.16 | 39.75 | 39.07 | 46.20 | 31.86 |
| Qwen2.5-VL-3B-Instruct | 33.21 | 30.34 | 33.49 | 32.96 | 46.84 | 36.28 |
| LongVA-7B | 29.46 | 24.88 | 29.91 | 28.81 | 51.90 | **39.83** |
| Qwen2.5-VL-7B-Instruct | 29.26 | 21.35 | 30.02 | 28.77 | 46.84 | 36.81 |
| deepseek-vl2-small | **47.62** | 26.91 | **49.63** | **48.32** | 56.33 | 31.11 |
| Robobrain | 37.38 | 29.53 | 38.14 | 37.56 | 55.06 | 30.57 |
| Claude-sonnet-4 | 44.75 | 47.62 | 44.48 | 45.32 | 49.38 | 31.74 |
| Space-Mantis | 22.82 | 29.32 | 22.19 | 22.15 | 45.57 | 33.48 |
| InternVL2-8B | 18.68 | 13.11 | 19.22 | 17.89 | **64.56** | 27.99 |
| Space-Qwen | 33.28 | 26.32 | 33.95 | 33.06 | 46.84 | 35.63 |
| LLaVA-Onevision-7B | 47.43 | 44.09 | 47.75 | 48.04 | 51.27 | 33.48 |
| Spatial-MLLM | 32.06 | 20.92 | 33.13 | 31.79 | 46.84 | 35.20 |
| mPLUG-Owl3-7B | 44.85 | 26.91 | 46.59 | 45.15 | 60.13 | 35.74 |

Table 2: Performance of VLMs on MINDCUBE across categories.(Part 2)

| Model | Relation Pattern | | | Viewpoint Dynamics | | |
|---|---|---|---|---|---|---|
| | A-A | A-O | O-O | Rotation | Meanwhile | Sequence |
| LLaVA-Video-7B-Qwen2 | 36.22 | 57.61 | 26.67 | 35.71 | 30.12 | 73.45 |
| Mantis(SigLip) | 23.78 | 64.16 | 25.24 | 37.65 | 24.99 | 82.74 |
| GPT-4o | **49.30** | 48.38 | 16.70 | 32.65 | 31.09 | 59.73 |
| Qwen2.5-VL-3B-Instruct | 37.85 | 37.51 | 20.65 | 37.37 | 27.88 | 46.05 |
| LongVA-7B | 19.72 | 35.49 | 25.58 | 35.89 | 24.67 | 40.50 |
| Qwen2.5-VL-7B-Instruct | 31.41 | 34.67 | 15.63 | 38.76 | 22.87 | 43.76 |
| deepseek-vl2-small | 43.98 | **68.27** | 25.33 | 37.00 | 32.97 | **87.13** |
| Robobrain | 30.94 | 49.18 | 27.37 | 35.80 | 28.79 | 59.66 |
| Claude-sonnet-4 | 41.78 | 67.25 | 15.85 | **48.42** | **34.76** | 69.53 |
| Space-Mantis | 28.18 | 17.03 | 20.89 | 37.65 | 24.98 | 14.46 |
| InternVL2-8B | 15.67 | 12.47 | 24.58 | 36.45 | 21.78 | 7.36 |
| Space-Qwen | 31.59 | 38.14 | 26.13 | 38.02 | 28.51 | 44.58 |
| LLaVA-Onevision-7B | 42.28 | 65.87 | **29.79** | 36.45 | 33.80 | 84.38 |
| Spatial-MLLM | 27.72 | 37.75 | 25.80 | 38.39 | 26.84 | 44.19 |
| mPLUG-Owl3-7B | 47.80 | 62.29 | 18.83 | 37.84 | 31.02 | 81.55 |

## C  EVALUATION ON MINDCUBE

### C.1  PROMPT TEMPLATES FOR EVALUATION

> **Evaluation Prompt Prefix**
> Based on these images, answer the question based on this rule: You only need to provide *ONE* correct answer selecting from the options listed below. For example, if you think the correct answer is 'A. above' from ' A. above B. under C. front D. behind.', your response should only be 'A. above'.
> The Question is:

### C.2  DETAILS IN TEXT ONLY EVALUATION

In the text-only evaluation, we replace the original image input with corresponding textual descriptions and assess the performance of models based on these descriptions. The purpose of this evaluation is to highlight how much information may be lost or distorted when the visual input is sub-

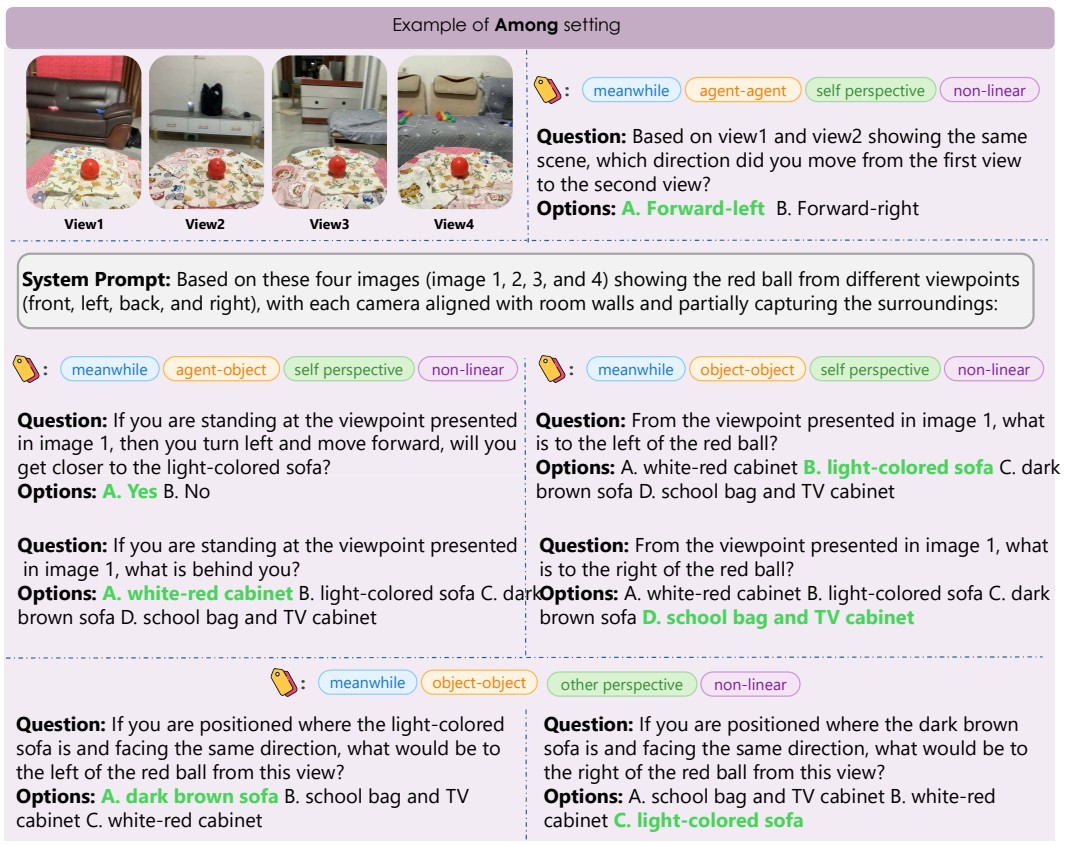

Figure 4: Example of among setting.

stituted with text-based representations, and to demonstrate the crucial role of visual data in the models' performance.

We used two types of captions: **brief** and **dense**. The brief captions provide a concise overview of the image, while the dense captions offer a more detailed description with a focus on the spatial relationships between objects. Additionally, the models are evaluated using textual descriptions (text-only evaluation) based on these captions, with no access to the actual images.

---

**Prompt for Brief Captioning**
Describe this image briefly.

---

**Prompt for Dense Captioning**
Describe this image in detail, specifically focusing on the spatial relationship between objects.

---

**Text-only evaluation Prompt Prefix**
You need to gather information about each image based on the descriptions I provide below, and answer the given questions using those textual descriptions, without directly viewing the images.

Image 1: <Caption 1>
...
Image N: <Caption N>

---

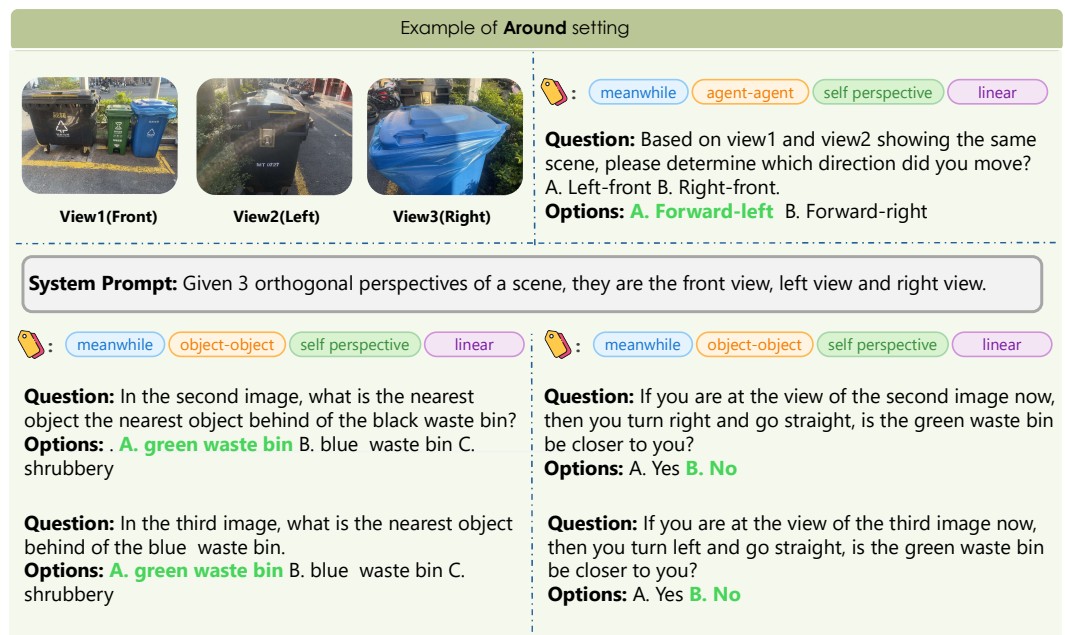

Figure 5: Example-1 of around setting.

As shown in the Table 3, all three models exhibit a noticeable performance decline when replacing the original image input with its corresponding text-based description. Specifically, the brief captions cause the most significant performance drop. For instance, RoboBrain-8B experiences a 7.83% decrease with the brief captions, and LLaVA-OneVision-7B drops by 12.91% in the same condition. Even when using dense captions, which offer more detail, there is still a performance reduction, although the decrease is slightly less pronounced compared to brief captions. In conclusion, while textual descriptions can convey some information, they fail to capture the richness and intricacies of visual data, leading to a marked reduction in performance across all models.

Table 3: Text-only (T) evaluation vs. original evaluation with image inputs (I). The results highlight a significant performance drop when the original image input is replaced with the corresponding text-based caption, particularly with the brief captions. In all cases, model performance decreases notably, underscoring that our benchmark is *vision-centric*.

| Model | Brief (T) | Dense (T) | Original (I) |
|---|---|---|---|
| RoboBrain-8B | 33.92% ↓7.83% | 35.58% ↓6.17% | 41.75% |
| LLaVA-OneVision-7B | 34.17% ↓12.91% | 35.92% ↓11.16% | 47.08% |
| Qwen2.5-VL-7B-Instruct | 27.00% ↓5.33% | 28.75% ↓3.58% | 32.33% |

C.3 HUMAN EVALUATION

We use our Tiny Benchmark— encompassing all task categories for evaluation by 5 human annotators, each of whom independently answers every question. Here is the results[4].

Table 4: Comparison of Human and GPT-4 Performance (%)

| Model/Annotator | GPT4-o | Human-max | Human-min | Human-avg |
|---|---|---|---|---|
| Accuracy | 36.54 | 94.77 | 94.20 | 94.55 |

This observation demonstrates the disparity in spatial reasoning capabilities between humans and state-of-the-art multimodal large language models, where humans exhibit superior performance in solving spatial problems that remain challenging for advanced AI systems.

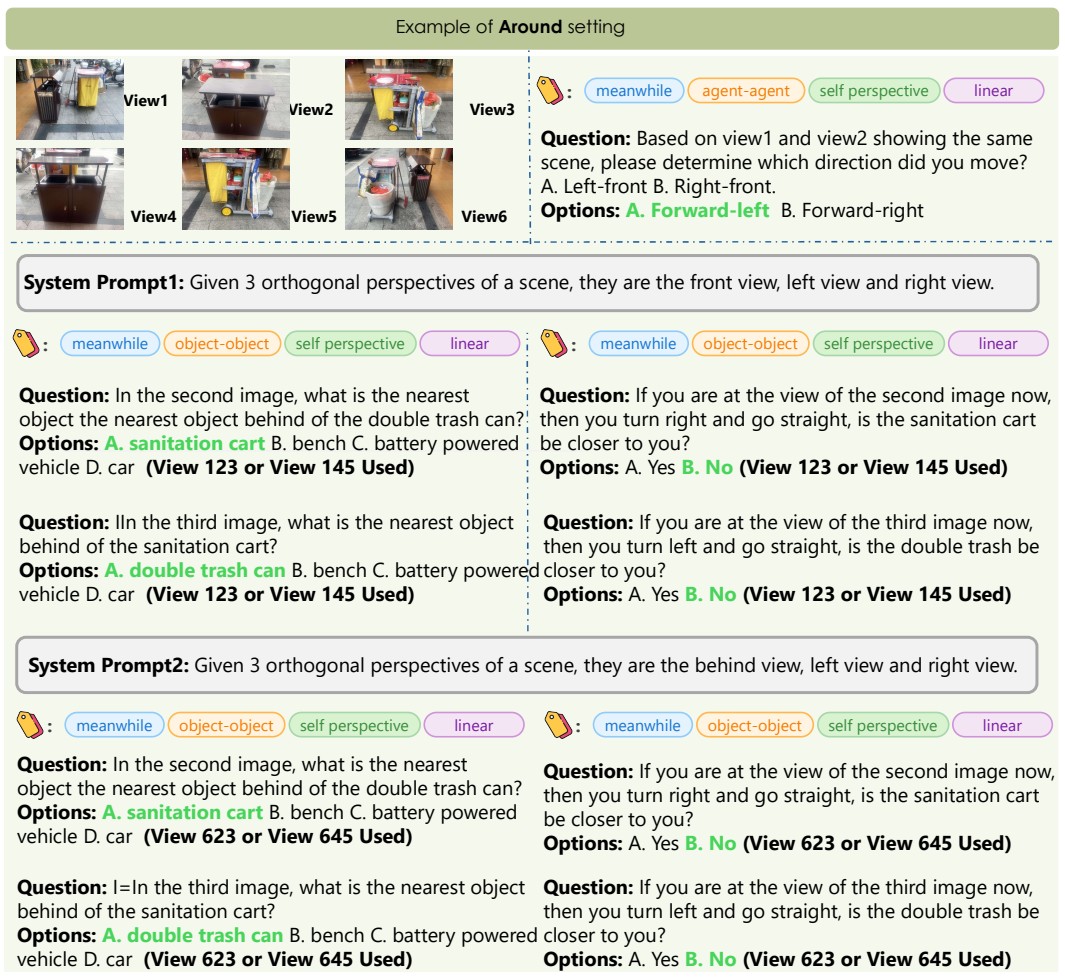

Figure 6: Example-2 of around setting.

## C.4 EVALUATION SETUP

To comprehensively evaluate model performance, we conducted experiments on a diverse suite of models. This suite includes models with native multi-image reasoning capabilities (e.g., LLaVA-Onevision (Li et al., 2024a), LLaVA-Video (Zhang et al., 2024d), mPLUG-Owl3 (Ye et al., 2024), InternVL2.5 (Chen et al., 2024b), QwenVL2.5 (Bai et al., 2025), LongVA (Zhang et al., 2024c), DeepSeek-VL2 (Lu et al., 2024)), Gemma3 Team et al. (2025), models fine-tuned on interleaved image-text data (e.g., Mantis (Jiang et al., 2024)), leading proprietary APIs (e.g., GPT-5, Claude-4-Sonnet), and models specifically fine-tuned for spatial reasoning tasks (e.g., RoboBrain (Ji et al., 2025), Space-Mantis (Chen et al., 2024a), Space-Qwen (Chen et al., 2024a), and Spatial-MLLM Wu et al. (2025a)).

## C.5 ANALYSIS IN SETTINGS

### C.5.1 AROUND

First, we examine the relationship between occlusion degree and response accuracy across four visibility levels (fully visible, mostly visible, mostly occluded, fully occluded) to determine whether performance degrades proportionally with increasing occlusion. Second, we investigate the impact of camera height variation within the same lateral viewpoint, as different vertical perspectives yield distinct occlusion patterns that may challenge the model's ability to maintain spatial coherence. These paradigms evaluate whether models perform consistently when transferring spatial

relationships across viewpoints, particularly in scenarios with significant object size discrepancies where smaller objects may be completely occluded from one angle but visible from another. This multifaceted analysis approach enables a more nuanced understanding of MLLMs' genuine 3D spatial reasoning capabilities beyond simple pattern recognition of 2D visual cues. We mainly evaluated GPT-4o and Qwen2.5-VL.

**Occlusion Degree Analysis.** Our analysis reveals a notable correlation between occlusion

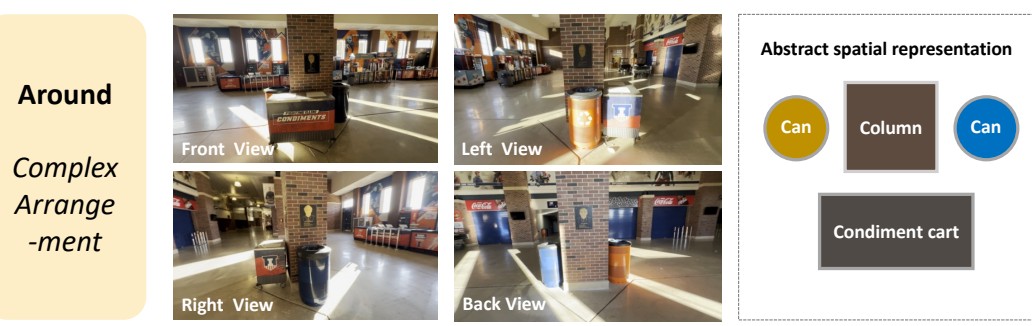

**Question1**: From the viewpoint presented in image 2, is there an blue trash can behind the brick-patterned column? A. Yes B. No.
**Question2**: From the viewpoint presented in image 3, is there an orange trash can behind the brick-patterned column? A. Yes B. No.

**Answer1**: In image 2, the viewpoint is from the left side of the brick-patterned column. Image 4 shows the back view of the column, where a blue trash can is visible. Therefore, from the viewpoint in image 2, the blue trash can is indeed behind the column.
Therefore, my answer is A. Yes.
**Answer2**:In image 3, the viewpoint shows a blue trash can next to the brick-patterned column. Image 4, which shows the back view, reveals an orange trash can next to the column. Therefore, from the viewpoint in image 3, the orange trash can is not visible because it is on the opposite side of the column.
Therefore, my answer is B. No.                                        **GPT-4o Response**

Figure 7: Example of complex spatial arrangement in around setting.

degree and model performance. Accuracy rates declined progressively with increasing occlusion, with an average decrease of 50.7% between fully visible and fully occluded conditions (p <0.01). Interestingly, the performance degradation was non-linear, with a precipitous drop occurring between the mostly visible and mostly occluded categories (28.7% decrease), suggesting a potential threshold effect in the models' spatial reasoning capabilities. Error analysis in Figure8 further revealed that models frequently defaulted to proximity-based guessing when confronted with heavily occluded objects, rather than leveraging cross-view information to reason about hidden spatial relationships.

**Camera Height Impact Analysis.** Varying camera heights significantly affected model performance through different occlusion patterns. High-angle perspectives yielded 24.8% higher accuracy than eye-level views by revealing tops of partially occluded objects and providing better scene context. This advantage was most pronounced in dense arrangements where top-down angles exposed spatial gaps between objects otherwise invisible from eye-level. Models clearly benefited from the holistic understanding afforded by elevated viewpoints, where global spatial relationships became more apparent. In contrast, eye-level perspectives with more severe occlusions led to poorer spatial reasoning, suggesting limited ability to mentally reconstruct hidden scene elements from partial visual information.

The integration of findings across all three dimensions indicates that current MLLMs possess partial but inconsistent 3D spatial reasoning capabilities. The models' performance appears heavily influenced by the visibility of key reference points across multiple viewpoints, suggesting

**Around**

*Different Height*

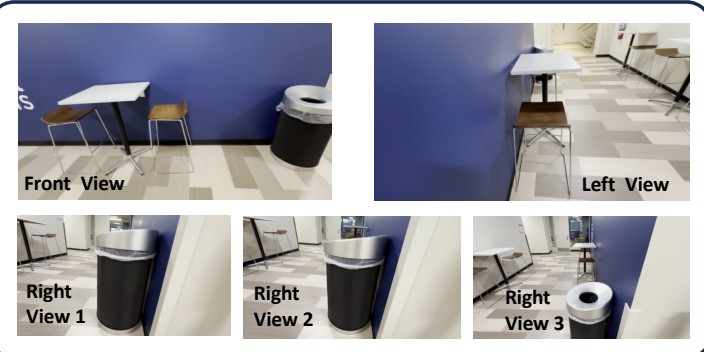

Front View

Left View

Right View 1

Right View 2

Right View 3

**Question**: Based on these three images (image 1, 2, and 3) showing the same scene from different viewpoints (front, left, and right): From the viewpoint presented in image 3, **is there a table behind the trash can?** A. Yes B. No."

Answer1: In image 3, we see the trash can positioned against the wall with no visible table behind it. The angle of the shot **does not show** any part of a table behind the trash can.
Therefore, my answer is **B. No**.

Answer2: In image 3, we see a trash can placed against a wall with a partial view of the floor and part of another object that appears to be a table. The angle suggests that if there were a table behind the trash can, it would be visible or at least partially seen. However, the image only shows the trash can and part of the floor, **without any indication of a table** being present behind it.
Therefore, my answer is **B. No**.

Answer3: In image 3, we see the trash can positioned in front of a wall with a table visible behind it. The perspective **shows that the table is indeed located** behind the trash can when viewed from this angle.
Therefore, my answer is A. Yes.

Answer1: - In image 1, we see a table and chairs adjacent to the trash can near wall.
- In image 2, the view from the left shows the table extends from the wall, and the trash can is closely beside it.
- In image 3, the view from the right toward the trash can shows the area where the table is located.
From this viewpoint, while there's no direct view behind the trash can, based on the previous images that illustrate the spatial arrangement, the table should be visible next to the trash can, not directly behind it.
Therefore, my answer is B. No.

Answer2: Image 1 shows a trash can to the right of a table with two chairs. Image 2 reinforces this by showing the table and chairs aligned along the wall, with the trash can just beyond the table. Image 3 shows the view from the side of the trash can, where the table is not visible directly behind it, suggesting it is adjacent rather than directly behind from this viewpoint.
Therefore, my answer is **B. No**

Answer3: In image 1, a table is visible next to a trash can. Image 2 confirms the same table and trash can placement from a different angle. Image 3 shows the trash can in the foreground, and a table is visible directly behind it. Therefore, my answer is **A. Yes.**

☐ Qwen-7B-Instruct Response

☐ GPT-4o Response

Figure 8: Example of different camera heights' impact in around setting.

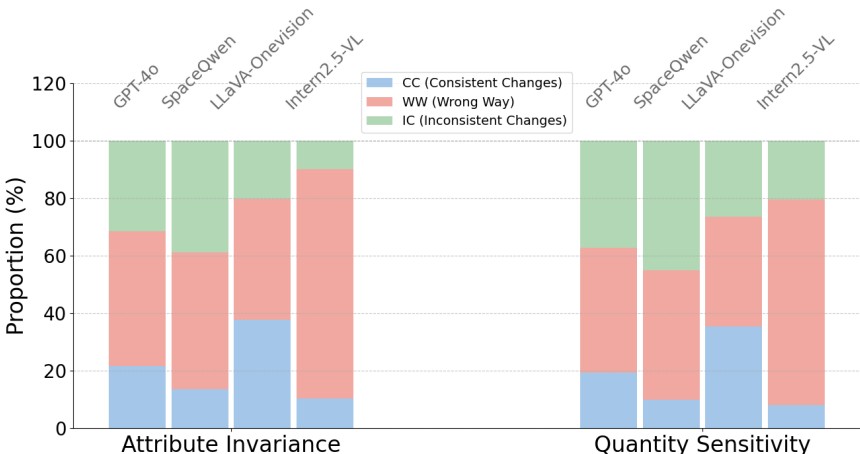

Figure 9: Paired question-answers inconsistency in two tests. We report the proportions of IC, CC and WW. Notably, SpaceQwen has a highest inconsistency(around 40%). GPT-4o and LLaVA-Onevision exhibit more balanced performance.

**a reliance on visual correspondence matching rather than true 3D mental modeling**. The observed asymmetries in spatial transformation paradigms further support this hypothesis, as a robust 3D reasoning system would demonstrate consistent performance regardless of the specific transformation required.

### C.5.2 AMONG

While the correctness of one single question indicates how often a model answers an isolated question correctly, it does not capture whether the model remains consistent when presented with objects with different visual attributes and different number of central objects while retaining the original peripheral scenes.

To investigate this, we also propose two different tests:

**Attribute Invariance Test.** We modify only the visual attributes (e.g., color, category) of the central object while keeping the spatial configuration of all objects unchanged, as shown in Figure10. A robust spatial reasoning system should maintain consistent answers, as spatial relationships remain invariant despite superficial attribute changes.

**Quantity Sensitivity Test.** We increase the number of central objects (e.g., from one to three) while retaining the original peripheral objects, as shown in Figure11. This modification is hypothesized to enhance reasoning performance, as additional central objects provide more reference points for establishing cross-view correspondences and consistency.

We also propose to look into the proportions of paired questions in tests where the answers are inconsistent with one another. First, we classify each paired instance into three scenarios: 1) CC(Both Correct) when the model answers both the primary and paired question correctly, 2) WW (Both Wrong) when it fails both versions, and 3) IC (Inconsistent) when the model answers one version correctly but fails the other.

As shown in Figure 9, we report the proportions of IC (in consistent) outcomes across 4 MLLMs in two tests — two open-source (Intern2.5-VL, LLaVA-Onevision), a spatial model (SpaceQwen) and a closed-source GPT-4o. We have several observations: 1) SpaceQwen exhibits notably high inconsistency score IC (around 40%) on both tasks, 2) LLaVA-Onevision remain fairly balanced inconsistency and high performance across tests, while InternVL vary significantly across tests.

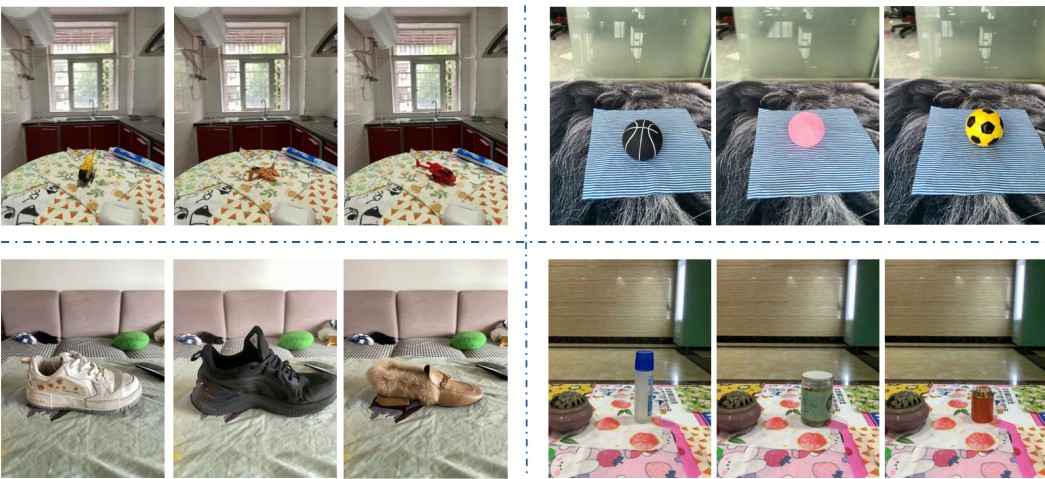

Figure 10: Examples in Attribute Invariance Test.

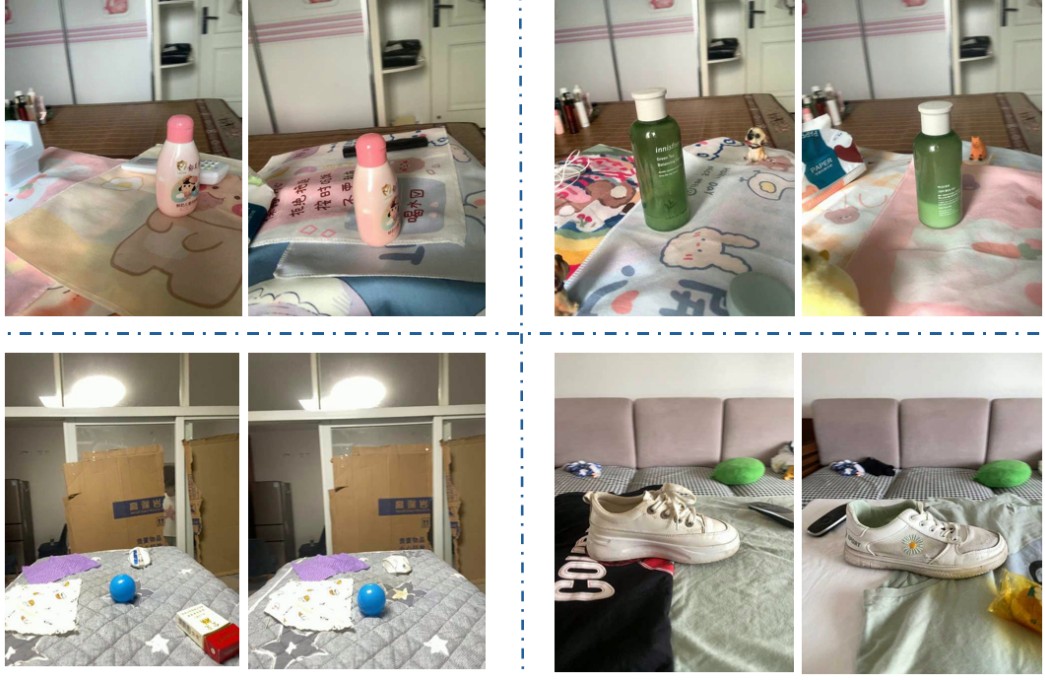

Figure 11: Examples in Quantity Sensitivity Test.

Our systematic evaluation demonstrates MLLMs can achieve attribute-invariant spatial reasoning but struggle to utilize additional reference objects effectively. This highlights the need for: (1) enhanced geometric reasoning architectures, and (2) comprehensive benchmarks evaluating both attribute invariance and quantity sensitivity in 3D spatial understanding.

## C.6 FAILURE CASE ANALYSIS

The observed pattern of errors indicates that models primarily rely on local relationship matching rather than inferring global spatial configurations, which represents a critical gap compared to human-like spatial reasoning abilities. Future architectural improvements should therefore focus on enhancing transitive spatial inference mechanisms and view-invariant scene representation to support more robust reasoning across multiple perspectives.

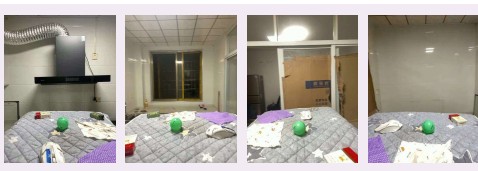 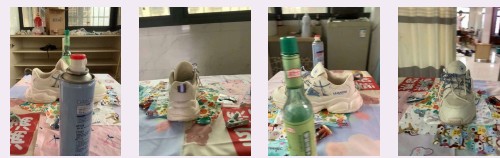

**Question:** Based on these four different viewpoints (front, left, back, and right), if you are positioned at the first viewpoint, then you turn right and move forward, wil you get closer to the window?
**Options:** A. No **B.Yes**

**<Reasoning>**:
1. The first image shows a view with a vent, likely the front view.
2. The second image has a window visible, indicating it is the view opposite to the first.
3. The third image features a large cardboard, potentially a covered window on one side.
4. The fourth image shows a plain wal, suggesting the view opposite the third image.
Based on these, the four views are likely structured in a roughly rectangular layout. Starting at the first view (front with vent), turning right would bring you to the wall (fourth image), away from the window (second image). Therefore, my answer is A. No.

**Question:** Based on these four different viewpoints (front, left, back, andright), if you are positioned at the third viewpoint, what is behind you?
**Options:** A.Washing machine **B.White wood rack** C.White + gray curtains D. White wall and window

**<Reasoning>**:
The third viewpoint shows the right side of the shoe, with the washing machine visible in the background.

Therefore, if you are positioned at the third viewpoint, the washing machine is behind you. Therefore, my answer is A. Washing machine.

Figure 12: Failure case analysis. We show GPT4-o's reasoning process. In case 1, the model is unable to establish the spatial location corresponding to each view; In case 2, the model confuses the subject of the "behind" relationship.

# D DATA STRUCTURES AS COGNITIVE SCAFFOLDS, EVALUATION METRICS, AND INPUT-OUTPUT CONFIGURATIONS

In this section, we provide detailed descriptions of the three data structures employed as cognitive scaffolds to approximate spatial mental models in VLMs, followed by formal definitions of the evaluation metrics employed across all experiments. Furthermore, we show the prompts for all the input-output configurations that were used across the following experiments.

## D.1 DATA STRUCTURES AS COGNITIVE SCAFFOLDS

The human ability to navigate and reason about space, especially with incomplete information, is largely attributed to the formation of internal spatial mental models. These models, as extensively studied in cognitive science, are not necessarily veridical, metric-perfect replicas of the environment. Instead, they are often schematic and even distorted, yet functionally effective representations. These models can be especially useful for understanding the environment spatial layouts for agentic settings Yao et al. (2023); Wang et al. (2024), such as embodied scenarios Liang et al. (2023); Driess et al. (2023); Huang et al. (2023; 2024); Li et al. (2024b); Yang et al. (2025); Tang et al. (2025a). Pioneering work by Barbara Tversky suggests that these internal constructs are more akin to "cognitive collages" – flexible assemblies of spatial information gleaned from various perspectives and experiences, rather than rigid, map-like blueprints Tversky (1993). These "cognitive collages" allow for the integration of fragmented observations and support reasoning across unseen perspectives. Johnson-Laird Johnson-Laird (1983) posits that mental models, including those for space, serve as *"structural analogs of the world,"* enabling individuals to simulate and infer spatial relationships, such as determining the relative positions of objects from sequential descriptions (e.g., "A is to the left of B; B is in front of C"). Research by Tversky Tversky et al. (1994) has also demonstrated that individuals can construct rich, multi-dimensional mental representations even from linear, descriptive texts, and subsequently query these models from various psychological viewpoints.

Inspired by these cognitive theories, we explore three distinct data structures designed to act as cognitive scaffolds for VLMs. When VLMs are presented with limited visual input, these structures aim to approximate different facets of human spatial mental modeling: dynamic updating, integrated spatial layout representation, and inferential reasoning.

### D.1.1 VIEW INTERPOLATION FOR DYNAMIC UPDATING

Human spatial mental models are not static; they are continuously updated with new sensory information and through mental simulation, such as imagining a change in viewpoint. To approximate this dynamic updating and mental animation capability Hegarty (1992), we employ view interpolation. This technique aims to bridge perceptual gaps between discrete, sparsely sampled views by generating intermediate visual frames.

**Our Setting:** In our experiments, view interpolation is implemented by inserting synthetic frames *between* consecutive views provided to the model. For instance, if "1 interpolated frame" is specified, one new frame is generated and inserted between an initial view $V_n$ and the subsequent view $V_{n+1}$ (e.g., between View 1 and View 2). Similarly, "2 interpolated frames" would mean two synthetic frames are inserted in sequence between $V_n$ and $V_{n+1}$. For the interpolated frames, we either define a heuristic function to choose from the original datasets Baruch et al. (2021); Xia et al. (2024) where we sampled our data, or we use Stable Virtual Camera Zhou et al. (2025) to generate intermediate frames for those image groups without. This approach is intended to provide a smoother perceptual experience, potentially aiding the VLM in tracking object relations and maintaining spatial consistency across viewpoint shifts. (Refer to Figure 3 in the main paper for a conceptual illustration)

### D.1.2 COGNITIVE MAPS FOR INTEGRATED SPATIAL LAYOUTS

A core aspect of spatial cognition is the ability to form an allocentric (world-centered) or survey-like understanding of an environment, capturing the relative locations of objects. Tversky Tversky (1993; 2003) highlights that such representations often involve different frames of reference and hierarchical structures. Cognitive maps in our context are 2D schematic representations that attempt to embody this integrated spatial layout.

**Our Setting:** We investigate two variants of cognitive maps, both represented as structured data (e.g., JSON-like objects), to capture the spatial layout:

- We provide a 2D grid map of the scene that is related to the question to be answered.
- The map uses a 10×10 grid, where `[0, 0]` is the top-left corner and `[9, 9]` is the bottom-right corner (i.e., bird's-eye view).
- Directions are defined as follows:
    - `up` = towards the top of the grid (decreasing y-value)
    - `right` = towards the right of the grid (increasing x-value)
    - `down` = towards the bottom of the grid (increasing y-value)
    - `left` = towards the left of the grid (decreasing x-value)
    - `inner` = into the 2D map (perpendicular to the grid, pointing away from you)
    - `outer` = out of the 2D map (perpendicular to the grid, pointing toward you)
- The map contains:
    - `objects` — a list of all important items in the scene with their `position`
    - `facing` — indicating the direction an object is oriented (when applicable)
    - `views` — representing different camera viewpoints in the scene

- **Augmented Cognitive Map:** This version explicitly integrates the observer's perspective by encoding the `positions` and `orientations` (facing directions) of the camera viewpoints within the map, alongside the objects and their locations. For instance, as depicted in our data examples (refer to Figure 3, Cognitive Map - Augmented panel), an augmented map might define a list of `objects` with their `name` and `position` (e.g., `"Tissue box": { "position": [5, 5] }`), and a separate list of `views` detailing each camera's `name` (e.g., `"View 1"`), `position` (e.g., `[3, 5]`), and `facing` direction (e.g., `"up"`).
- **Plain Cognitive Map (Object Only):** This is a more simplified, object-centric representation. It primarily focuses on the `spatial locations of objects` and, for some objects, their intrinsic `orientation` (facing direction) from a top-down survey perspective, without explicitly embedding camera view information within its structure. For example (refer to Figure 3, Cognitive Map - Plain panel), a plain map might list objects like `"Potted plant"` with

its `position` (e.g., `[5, 6]`) and `facing` direction (e.g., `"down"`), and another object like `"Sofa"` with only its `position` (e.g., `[4, 5]`). This type of map still allows for reasoning about object-to-object relationships and, where specified, object orientations, but abstracts away the explicit camera viewpoints that generated the scene understanding.

In both map types, coordinates represent positions on a 2D grid, and `facing` directions can be categorical (e.g., `"up"`, `"down"`, `"left"`, `"right"`, `"outer"`, `"inner"`). These structures aim to provide the VLM with an explicit, albeit potentially imperfect, schematic of the environment that it can then learn to generate and utilize for spatial reasoning tasks.

As for the format, our JSON format has been widely adopted as a computational model providing a flexible structure for VLMs, designed to offer a bird's-eye view representation encoding the relative positions and orientations of objects Yang et al. (2024). This representation aligns, at a high level, with the functional principles of cognitive maps in cognitive science. Our goal is to equip VLMs with a scaffold that approximates the functional role of a cognitive map to enable explicit reasoning, rather than replicating its exact neurological basis.

The use of JSON is a principled choice for interfacing with text-native VLMs, following standard practices for eliciting structured outputs. VLMs fundamentally operate on sequences of language tokens, making JSON a naturally fitting text-based format. JSON provides a structured and computationally effective means to evaluate complex spatial outputs, constituting one of the standard methods for eliciting structured knowledge from LLMs and VLMs. Although differentiable vectorized representations represent a promising research direction, current integration attempts have been widely recognized as ineffective, particularly owing to limitations in VLM comprehension.

### D.1.3 FREE FORM REASONING

Spatial mental models are not just static representations; they are actively used for inference and problem-solving Tversky et al. (1994). To approximate this procedural and inferential aspect, we utilize free-form reasoning, implemented as a natural language Chain-of-Thought (CoT) Wei et al. (2022) process. This encourages the VLM to externalize its step-by-step reasoning process when deducing an answer to a spatial query.

**Our Setting:** The VLM is prompted to generate a textual reasoning chain before outputting the final answer. This process is guided by a three-step principle, exemplified by the reasoning chain shown in Figure 3, the reasoning chain panel. For the steps shown in that example, they are: (1) *Initial Observation and Grounding:* The model first processes each available view, identifying key objects and their immediate spatial relationships within that specific viewpoint. For instance, the example chain begins with: `"In View 1, I see a potted plant, tissue box, and hand sanitizer from left to right, with a sofa behind."` This step grounds the reasoning in direct visual evidence from individual perspectives. (2) *Cross-View Integration and Environment Consolidation:* Next, the model attempts to identify consistent objects or environmental cues across the different views to recognize that they depict the same underlying 3D scene. The example reasoning continues: `"In View 2, I see the same potted plant, so both views are from the same environment."` This step is crucial for building a unified understanding of the space from discrete observations. (3) *Question-Guided Spatial Inference:* Finally, based on the specific question posed and the integrated understanding from the previous steps, the model performs step-by-step logical and spatial inferences to arrive at the answer. In the example, this involves relating the object positions across views relative to the observer's position in View 2: `"Since the hand sanitizer is rightmost in View 1, it's spatially furthest behind the potted plant when looking in View 2.`
`In View 2, the potted plant is closest to me, so the hand sanitizer is`
`the furthest from me."`

### D.2 EVALUATION METRICS

To quantitatively assess how these data structures affect the performance of VLMs in the spatial mental modeling presented in MINDCUBE, and to evaluate the quality of the generated cognitive

maps, we employed the following metrics: (1) *QA Accuracy*, and (2) *Graph Metrics for Generated Cognitive Maps*.

### D.2.1 QA ACCURACY

QA Accuracy serves as the core metric for evaluating task performance. It quantifies the proportion of questions that the vision-language model (VLM) answers correctly out of the total number of questions. A higher QA Accuracy indicates better alignment between the model's responses and the ground truth.

The metric is formally defined as:

$$\text{QA Accuracy} = \frac{N_{\text{correct}}}{N_{\text{total}}} \times 100\%$$

where $N_{\text{correct}}$ denotes the number of correctly answered questions, and $N_{\text{total}}$ is the total number of questions evaluated.

### D.2.2 GRAPH METRICS FOR COGNITIVE MAPS

To quantitatively evaluate the quality of a generated cognitive map, we use a set of structured graph-based metrics. The overall process consists of several key steps:

1. **Validity Check.** First, we ensure that the generated map is syntactically and semantically valid—i.e., it has a correct JSON format, contains interpretable object positions, and includes at least one valid object.

2. **Rotation Normalization.** Since we do not enforce a fixed orientation for generated maps (to allow for flexible generation from vision-language models), we evaluate the similarity between the generated map and the ground truth across a set of 3D rotations. We always choose the best-aligned rotation to compute our similarity scores.

3. **Structural Matching.** We define a relation graph between object pairs in each map, capturing directional and proximity-based relationships. A core part of the evaluation is determining whether these relationships in the ground truth are preserved in the generated map.

4. **Similarity Metrics.** We compute coverage (how many ground-truth objects are present), directional similarity (relative spatial relations), and facing similarity (object orientation). These are aggregated into an overall similarity score.

5. **Rotation-Invariant Isomorphism.** We also evaluate whether a generated map is graph-isomorphic to the ground truth under any allowed 3D rotation, providing a strict measure of structural correctness.

Below, we provide precise mathematical definitions for each of these components.

**Notation.** A *cognitive map* is a finite set of objects $\mathcal{O} = \{o_1, \ldots, o_n\}$ where each object $o_i$ is associated with (i) a 2-D position vector $p_i = (x_i, y_i) \in \mathbb{R}^2$ and (ii) an optional facing label $f_i \in \{\text{up}, \text{right}, \text{down}, \text{left}, \text{inner}, \text{outer}\} \cup \{\varnothing\}$. For two maps, we distinguish (1) the *ground-truth* map $(\mathcal{O}^\star, p^\star, f^\star)$ and (2) a *generated* map $(\mathcal{O}^g, p^g, f^g)$.

The set of objects that appear in both maps is $\mathcal{O}^c = \mathcal{O}^\star \cap \mathcal{O}^g$.

**Extended directional relation.** We define a directional or proximity-based relationship between any ordered object pair $(o_i, o_j)$ based on their spatial arrangement and optional facing annotations. This relation is captured via the function:

$$
\text{dir}(o_i, o_j) = \begin{cases}
\text{right} & |x_j - x_i| > |y_j - y_i| \text{ and } x_j > x_i, \\
\text{left} & |x_j - x_i| > |y_j - y_i| \text{ and } x_j < x_i, \\
\text{down} & |y_j - y_i| \geq |x_j - x_i| \text{ and } y_j > y_i, \\
\text{up} & |y_j - y_i| \geq |x_j - x_i| \text{ and } y_j < y_i, \\
\text{inner} & \|p_j - p_i\|_2 < \delta \text{ and } (f_i = \text{inner} \vee f_j = \text{outer}), \\
\text{outer} & \|p_j - p_i\|_2 < \delta \text{ and } (f_i = \text{outer} \vee f_j = \text{inner}), \\
\varnothing & \text{otherwise,}
\end{cases}
$$

with threshold $\delta = 0.5$ as in the implementation. These relations form a *relation matrix*:

$$
R(o_i, o_j) = \text{dir}(o_i, o_j).
$$

**Coverage.** Coverage measures how many ground-truth objects are successfully retrieved in the generated map:

$$
\text{Cov} = \frac{|\mathcal{O}^c|}{|\mathcal{O}^\star|} \in [0, 1].
$$

**Directional similarity.** We now evaluate how well the generated map preserves the directional relationships among object pairs from the ground truth. Define:

$$
\mathcal{P}^\star = \{(o_i, o_j) \in \mathcal{O}^c \times \mathcal{O}^c \mid i \neq j, \ R^\star(o_i, o_j) \neq \varnothing\}.
$$

Then the directional similarity score is given by:

$$
S_{\text{dir}} = \frac{\left|\{(o_i, o_j) \in \mathcal{P}^\star \mid R^g(o_i, o_j) = R^\star(o_i, o_j)\}\right|}{|\mathcal{P}^\star|} \in [0, 1],
$$

which corresponds to the proportion of directional relations in the ground truth that are correctly matched in the generated map.

**Facing similarity.** For objects with defined facing directions, we compare their orientation across the two maps:

$$
\mathcal{F}^\star = \{o_i \in \mathcal{O}^c \mid f_i^\star \neq \varnothing\}.
$$

Then:

$$
S_{\text{face}} = \frac{\left|\{o_i \in \mathcal{F}^\star \mid f_i^g = f_i^\star\}\right|}{|\mathcal{F}^\star|} \in [0, 1].
$$

**Overall similarity.** To aggregate the directional and facing similarities, we use a weighted combination:

$$
S_{\text{overall}} = \alpha \cdot S_{\text{dir}} + (1 - \alpha) \cdot S_{\text{face}} \in [0, 1],
$$

where $\alpha = 0.7$ places greater emphasis on spatial layout than orientation.

**Rotation-invariant isomorphism.** To ensure fair comparison regardless of orientation, we define a set of 3D rotations: $\mathcal{R} = \{R_1, \ldots, R_m\}$, including all 90° turns about the $z$-axis, and one 90° turn about each of the $x$- and $y$-axes.

We say the maps are *rotation-invariant isomorphic* if there exists a rotation such that their relation matrices match completely:

$$
\exists k \in \{1, \ldots, m\} \ \forall o_i, o_j \in \mathcal{O}^\star : \quad R^\star(o_i, o_j) = R^g_{(k)}(o_i, o_j),
$$

where $R^g_{(k)}$ is the relation matrix computed after applying $R_k$ to the generated map.

**Graph validity.** Finally, a generated map is deemed *valid* if: (1) It is well-formed JSON, (2) All fields conform to expected formats and constraints, and (3) At least one object has a valid position.

Together, the tuple $\left(\text{Cov}, S_{\text{dir}}, S_{\text{face}}, S_{\text{overall}}, \text{Iso}_{\text{rot}}\right)$ provides a comprehensive, rotation-aware evaluation of how closely a generated cognitive map matches ground truth structure and orientation.

### D.3 PROMPTS FOR ALL INPUT-OUTPUT CONFIGURATIONS

Below, we provide all prompts for the input-output configurations we investigate in our work.

#### D.3.1 EXAMPLE FOR Raw-QA

> **Example Prompt for Raw-QA**
>
> 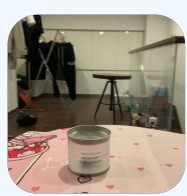 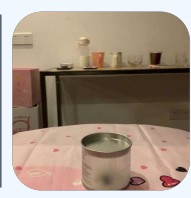 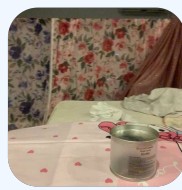 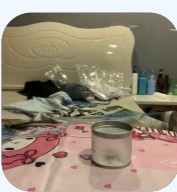
>
> [Task]
> Your task is to analyze the spatial arrangement of objects in the scene by examining the provided images, which show the scene from different viewpoints.
> [Answer Instruction]
> You only need to provide *ONE* correct answer selecting from the options listed below. For example, if you think the correct answer is 'A. Above' from 'A. Above B. Under C. Front D. Behind', your response should **only** be '<answer>A. Above</answer>'.
> [Question]
> Based on these four images (image 1, 2, 3, and 4) showing the white jar from different viewpoints (front, left, back, and right), with each camera aligned with room walls and partially capturing the surroundings: From the viewpoint presented in image 4, what is to the left of the white jar?
> A. Table with cups on it B. Clothes rack C. Bed sheet with a floral pattern D. White headboard

#### D.3.2 EXAMPLE FOR FFR

> **Example Prompt for FFR: Free-Form Reasoning**
>
> 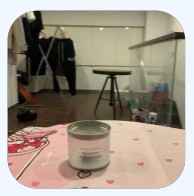 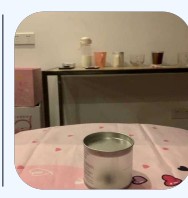 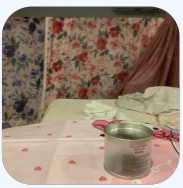 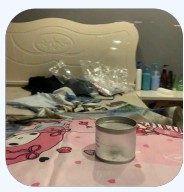
>
> [Task]
> Your task is to analyze the spatial arrangement of objects in the scene by examining the provided images, which show the scene from different viewpoints.
> [Answer Instruction]
> Please do step by step reasoning first, then give your final answer. For example, if you think the correct answer is 'A. Above' from 'A. Above B. Under C. Front D. Behind', your response should be this format: '<think>(replace with your reasoning here)</think><answer>A. Above</answer>'.
> [Question]
> Based on these four images (image 1, 2, 3, and 4) showing the white jar from different viewpoints (front, left, back, and right), with each camera aligned with room walls and partially capturing the surroundings: From the viewpoint presented in image 4, what is to the left of the white jar?
> A. Table with cups on it B. Clothes rack C. Bed sheet with a floral pattern D. White headboard

### D.3.3 EXAMPLE FOR VI-1 AND VI-2

**Prompt for VI-1: View Interpolation with 1 Frame**

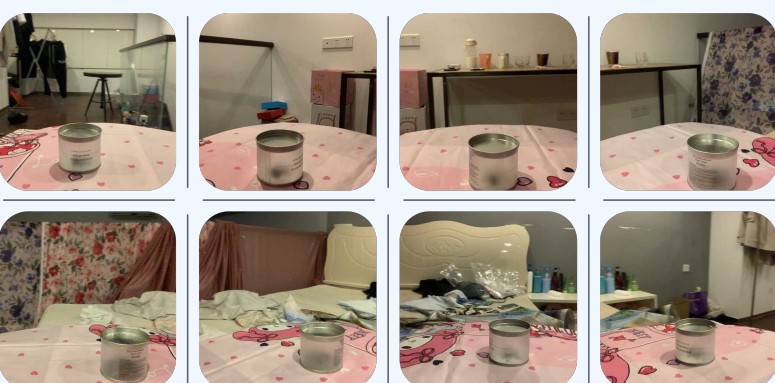

[Task]
Your task is to analyze the spatial arrangement of objects in the scene by examining the provided images, which show the scene from different viewpoints.
[Answer Instruction]
You only need to provide *ONE* correct answer selecting from the options listed below. For example, if you think the correct answer is 'A. Above' from 'A. Above B. Under C. Front D. Behind', your response should **only** be '<answer>A. Above</answer>'.
[Question]
Based on these 8 images showing the white jar from different viewpoints (from front (image 1) to left (image 3), from left (image 3) to back (image 5), from back (image 5) to right (image 7), from right (image 7) back to front (image 1)), with each camera aligned with room walls and partially capturing the surroundings: From the viewpoint presented in image 7, what is to the left of the white jar?
A. Table with cups on it B. Clothes rack C. Bed sheet with a floral pattern D. White headboard

**Prompt for VI-2: View Interpolation with 2 Frames**

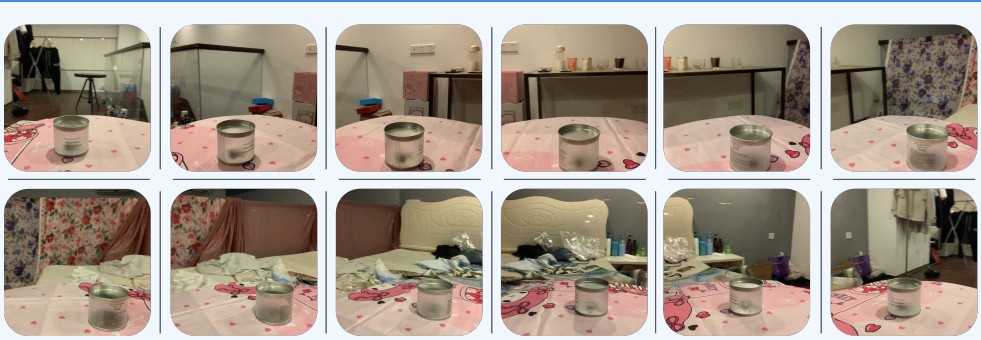

[Task]
Your task is to analyze the spatial arrangement of objects in the scene by examining the provided images, which show the scene from different viewpoints.
[Answer Instruction]
You only need to provide *ONE* correct answer selecting from the options listed below. For example, if you think the correct answer is 'A. Above' from 'A. Above B. Under C. Front D. Behind', your response should **only** be '<answer>A. Above</answer>'.
[Question]
Based on these 12 images showing the white jar from different viewpoints (from front (image 1) to left (image 4), from left (image 4) to back (image 7), from back (image 7) to right

(image 10), from right (image 10) back to front (image 1)), with each camera aligned with room walls and partially capturing the surroundings: From the viewpoint presented in image 10, what is to the left of the white jar?
A. Table with cups on it B. Clothes rack C. Bed sheet with a floral pattern D. White headboard

### D.3.4 EXAMPLE FOR `Aug-CGMap-In`

**Prompt for `Aug-CGMap-In`: Grounded Augmented Cognitive Map as Input**

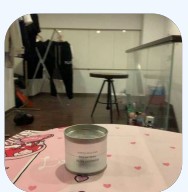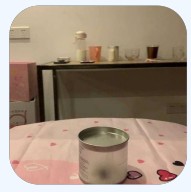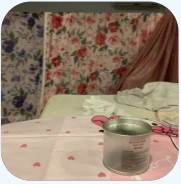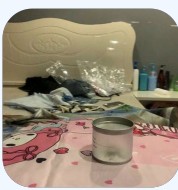

[Task]
Your task is to analyze the spatial arrangement of objects in the scene by examining the provided images, which show the scene from different viewpoints. Also, we provide you a cognitive map that shows the general layout for the scene. Please use the cognitive map to reason and answer the question.
[Answer Instruction]
You only need to provide *ONE* correct answer selecting from the options listed below. For example, if you think the correct answer is 'A. Above' from 'A. Above B. Under C. Front D. Behind', your response should **only** be '<answer>A. Above</answer>'.
[Cognitive Map Format]
We provide you a 2D grid map of the scene that is related to the question you should answer. Below is the description of the map:
- The map uses a 10x10 grid where [0,0] is at the top-left corner and [9,9] is at the bottom-right corner
- The map is shown in the bird's view
- Directions are defined as:
* up = towards the top of the grid (decreasing y-value)
* right = towards the right of the grid (increasing x-value)
* down = towards the bottom of the grid (increasing y-value)
* left = towards the left of the grid (decreasing x-value)
* inner = straight into the 2D map (perpendicular to the grid, pointing away from you)
* outer = straight out of the 2D map (perpendicular to the grid, pointing towards you)
- "objects" lists all important items in the scene with their positions
- "facing" indicates which direction an object is oriented towards (when applicable)
- "views" represents the different camera viewpoints in the scene
Below is the cognitive map of the scene related to the question. Please use it to reason and answer the question.

```json
{
  "objects": [
    {"name": "white jar", "position": [5, 5]},
    {"name": "bed sheet with a floral pattern",
    "position": [5, 8]},
    {"name": "white headboard", "position": [2, 5]},
    {"name": "clothes rack", "position": [5, 2]},
    {"name": "table with cups on it", "position": [8, 5]}
  ],
  "views": [
    {"name": "Image 1", "position": [5, 6], "facing": "up"},
    {"name": "Image 2", "position": [4, 5], "facing": "right"},
```

```
    {"name": "Image 3", "position": [5, 4], "facing": "down"},
    {"name": "Image 4", "position": [6, 5], "facing": "left"}
  ]
}
```

[Question]
Based on these four images (image 1, 2, 3, and 4) showing the white jar from different view-points (front, left, back, and right), with each camera aligned with room walls and partially capturing the surroundings: From the viewpoint presented in image 4, what is to the left of the white jar?
A. Table with cups on it B. Clothes rack C. Bed sheet with a floral pattern D. White head-board

### D.3.5 EXAMPLE FOR `AUG-CGMAP-OUT`

**Prompt for `Aug-CGMap-Out`: Ask VLM to Output Augmented Cognitive Map and Direct Answer**

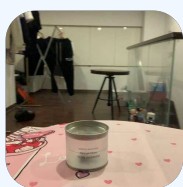 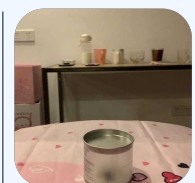 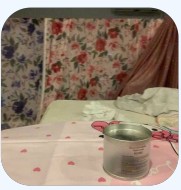 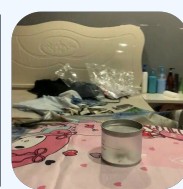

[Task]
Your task is to analyze the spatial arrangement of objects in the scene by examining the provided images, which show the scene from different viewpoints. You will then create a detailed cognitive map representing the scene using a 10x10 grid coordinate system.
[Rules]
1. Focus ONLY on these categories of objects in the scene: {white jar, bed sheet with a floral pattern, white headboard, clothes rack, table with cups on it}
2. Create a cognitive map with the following structure in the bird's view:
- A 10x10 grid where [0,0] is at the top-left corner and [9,9] is at the bottom-right corner
- up = towards the top of the grid (decreasing y)
- right = towards the right of the grid (increasing x)
- down = towards the bottom of the grid (increasing y)
- left = towards the left of the grid (decreasing x)
- inner = straight into the 2D map (perpendicular to the grid, pointing away from you)
- outer = straight out of the 2D map (perpendicular to the grid, pointing towards you)
- Include positions of all objects from the specified categories
- Estimate the center location (coordinates [x, y]) of each instance within provided categories
- If a category contains multiple instances, include all of them
- Each object's estimated location should accurately reflect its real position in the scene, preserving the relative spatial relationships among all objects
- Combine and merge information from the images since they are pointing to the same scene, calibrating the object locations accordingly
- Include camera positions and directions for each view
3. Carefully integrate information from all views to create a single coherent spatial representation.
[Answer Instruction]
1. Given the provided views and main objects mentioned in the above rules, you **MUST** present your cognitive map in the following JSON format **before your answer**:

```json
{
  "objects": [
```

```
    {"name": "object_name", "position": [x, y],
      "facing": "direction"},
    {"name": "object_without_orientation", "position": [x, y]}
  ],
  "views": [
    {"name": "View/Image 1", "position": [x, y],
      "facing": "direction"},
    {"name": "View/Image 2", "position": [x, y],
      "facing": "direction"}
  ]
}
```

2. Next, provide *ONE* correct answer selecting from the options. Your answer field must be in the format like "A. Above".

3. In general, your response's format should be like "Based on my observation, the answer is: <cogmap>(Replace with your cogmap here)</cogmap><answer>(Replace with your answer here)</answer>". Your option must be from the available options.

[Question]
Based on these four images (image 1, 2, 3, and 4) showing the white jar from different viewpoints (front, left, back, and right), with each camera aligned with room walls and partially capturing the surroundings: From the viewpoint presented in image 4, what is to the left of the white jar?

A. Table with cups on it B. Clothes rack C. Bed sheet with a floral pattern D. White headboard

### D.3.6 EXAMPLE FOR PLAIN-CGMAP-OUT

**Prompt for Plain-CGMap-Out:** Ask VLM to Output Plain Cognitive Map and Direct Answer

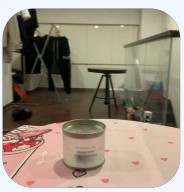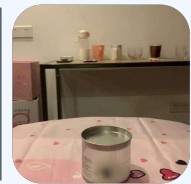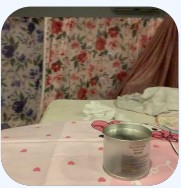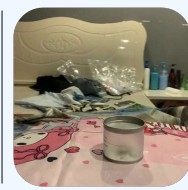

[Task]
Your task is to analyze the spatial arrangement of objects in the scene by examining the provided images, which show the scene from different viewpoints. You will then create a detailed cognitive map representing the scene using a 10x10 grid coordinate system.

[Rules]
1. Focus ONLY on these categories of objects in the scene: {white jar, bed sheet with a floral pattern, white headboard, clothes rack, table with cups on it}

2. Create a cognitive map with the following structure in the bird's view:
- A 10x10 grid where [0, 0] is at the top-left corner and [9, 9] is at the bottom-right corner
- up = towards the top of the grid (decreasing y)
- right = towards the right of the grid (increasing x)
- down = towards the bottom of the grid (increasing y)
- left = towards the left of the grid (decreasing x)
- Include positions of all objects from the specified categories
- Estimate the center location (coordinates [x, y]) of each instance within provided categories
- If a category contains multiple instances, include all of them
- Object positions must maintain accurate relative spatial relationships
- Combine and merge information from the images since they are pointing to the same scene, calibrating the object locations with grid coordinates accordingly

3. Carefully integrate information from all views to create a single coherent spatial repre-

sentation.
[Answer Instruction]
1. Given the provided views and main objects mentioned in the above rules, you **MUST** present your cognitive map in the following JSON format **before your reasoning**:

```json
{
    "object_category_1": {"position": [x, y]},
    "object_category_2": {"position": [x, y],
        "facing": "direction"},
    # if the object is asked for orientation
    ...
}
```

2. Next, provide *ONE* correct answer selecting from the options. Your answer field must be in the format like "A. Above"
3. In general, your response's format should be like "Based on my observation, the answer is: <cogmap>(Replace with your cogmap here)</cogmap><answer>(Replace with your answer here)</answer>". Your option must be from the available options.
[Question]
Based on these four images (image 1, 2, 3, and 4) showing the white jar from different viewpoints (front, left, back, and right), with each camera aligned with room walls and partially capturing the surroundings: From the viewpoint presented in image 4, what is to the left of the white jar?
A. Table with cups on it B. Clothes rack C. Bed sheet with a floral pattern D. White headboard

### D.3.7 EXAMPLE FOR `PLAIN-CGMAP-FFR-OUT`

**Prompt for `Plain-CGMap-FFR-Out`: Ask VLM to Output Plain Cognitive Map and Free-Form Reasoning**

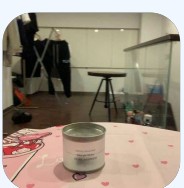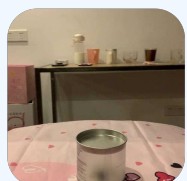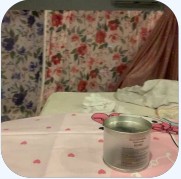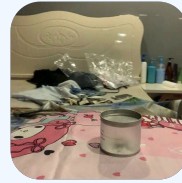

[Task]
Your task is to analyze the spatial arrangement of objects in the scene by examining the provided images, which show the scene from different viewpoints. You will then create a detailed cognitive map representing the scene using a 10x10 grid coordinate system.
[Rules]
1. Focus ONLY on these categories of objects in the scene: {white jar, bed sheet with a floral pattern, white headboard, clothes rack, table with cups on it}
2. Create a cognitive map with the following structure in the bird's view:
- A 10x10 grid where [0, 0] is at the top-left corner and [9, 9] is at the bottom-right corner
- up = towards the top of the grid (decreasing y)
- right = towards the right of the grid (increasing x)
- down = towards the bottom of the grid (increasing y)
- left = towards the left of the grid (decreasing x)
- Include positions of all objects from the specified categories
- Estimate the center location (coordinates [x, y]) of each instance within provided categories
- If a category contains multiple instances, include all of them
- Object positions must maintain accurate relative spatial relationships
- Combine and merge information from the images since they are pointing to the same scene,

calibrating the object locations with grid coordinates accordingly
3. Carefully integrate information from all views to create a single coherent spatial representation.
[Answer Instruction]
1. Given the provided views and main objects mentioned in the above rules, you **MUST** present your cognitive map in the following JSON format **before your reasoning**:

```json
{
    "object_category_1": {"position": [x, y]},
    "object_category_2": {"position": [x, y],
    "facing": "direction"},
    # if the object is asked for orientation
    ...
}
```

2. Next, please also provide your reasons step by step in details, then provide *ONE* correct answer selecting from the options. Your answer field must be in the format like "A. Above"
3. In general, your response's format should be like "Based on my observation, the answer is: <cogmap>(Replace with your cogmap here)</cogmap><think>(Replace with your reasoning here)</think><answer>(Replace with your answer here)</answer>". Your option must be from the available options.
[Question]
Based on these four images (image 1, 2, 3, and 4) showing the white jar from different viewpoints (front, left, back, and right), with each camera aligned with room walls and partially capturing the surroundings: From the viewpoint presented in image 4, what is to the left of the white jar?
A. Table with cups on it B. Clothes rack C. Bed sheet with a floral pattern D. White headboard

### D.3.8 EXAMPLE FOR AUG-CGMAP-FFR-OUT

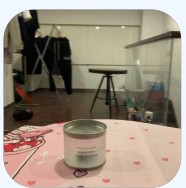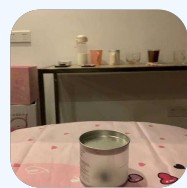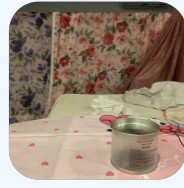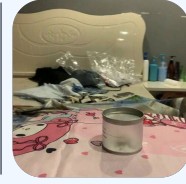

[Task]
Your task is to analyze the spatial arrangement of objects in the scene by examining the provided images, which show the scene from different viewpoints. You will then create a detailed cognitive map representing the scene using a 10x10 grid coordinate system.
[Rules]
1. Focus ONLY on these categories of objects in the scene: {white jar, bed sheet with a floral pattern, white headboard, clothes rack, table with cups on it}
2. Create a cognitive map with the following structure in the bird's view:
- A 10x10 grid where [0,0] is at the top-left corner and [9,9] is at the bottom-right corner
- up = towards the top of the grid (decreasing y)
- right = towards the right of the grid (increasing x)
- down = towards the bottom of the grid (increasing y)
- left = towards the left of the grid (decreasing x)
- inner = straight into the 2D map (perpendicular to the grid, pointing away from you)
- outer = straight out of the 2D map (perpendicular to the grid, pointing towards you)

- Include positions of all objects from the specified categories
- Estimate the center location (coordinates [x, y]) of each instance within provided categories
- If a category contains multiple instances, include all of them
- Each object's estimated location should accurately reflect its real position in the scene, preserving the relative spatial relationships among all objects
- Combine and merge information from the images since they are pointing to the same scene, calibrating the object locations accordingly
- Include camera positions and directions for each view
3. Carefully integrate information from all views to create a single coherent spatial representation.
[Answer Instruction]
1. Given the provided views and main objects mentioned in the above rules, you **MUST** present your cognitive map in the following JSON format **before your reasoning**:

```json
{
  "objects": [
    {"name": "object_name", "position": [x, y],
    "facing": "direction"},
    {"name": "object_without_orientation", "position": [x, y]}
  ],
  "views": [
    {"name": "View/Image 1", "position": [x, y],
     "facing": "direction"},
    {"name": "View/Image 2", "position": [x, y],
     "facing": "direction"}
  ]
}
```

2. Next, please also provide your reasons step by step in details, then provide *ONE* correct answer selecting from the options. Your answer field must be in the format like "A. Above"
3. In general, your response's format should be like "Based on my observation, the answer is: <cogmap>(Replace with your cogmap here)</cogmap><think>(Replace with your reasoning here)</think><answer>(Replace with your answer here)</answer>". Your option must be from the available options.
[Question]
Based on these four images (image 1, 2, 3, and 4) showing the white jar from different viewpoints (front, left, back, and right), with each camera aligned with room walls and partially capturing the surroundings: From the viewpoint presented in image 4, what is to the left of the white jar?
A. Table with cups on it B. Clothes rack C. Bed sheet with a floral pattern D. White headboard

### D.3.9 EXAMPLE FOR `CGMAP-IN-FFR-OUT`

Prompt for `CGMap-In-FFR-Out`: Input VLM with Grounded Cognitive Map and Output with Free-Form Reasoning

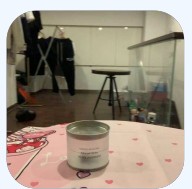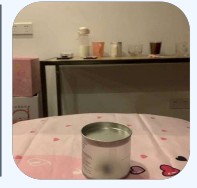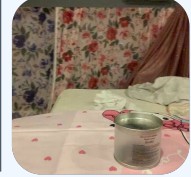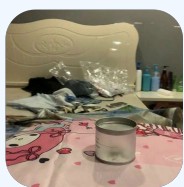

[Task]
Your task is to analyze the spatial arrangement of objects in the scene by examining the

provided images, which show the scene from different viewpoints. Also, we provide you a cognitive map that shows the general layout for the scene. Please use the cognitive map to reason and answer the question.

[Answer Instruction]

Please do step by step reasoning first, then give your final answer. For example, if you think the correct answer is 'A. Above' from 'A. Above B. Under C. Front D. Behind', your response should be this format: '<think>(replace with your reasoning here)</think><answer>A. Above</answer>'. [Cognitive Map Format]

We provide you a 2D grid map of the scene that is related to the question you should answer. Below is the description of the map:

- The map uses a 10x10 grid where [0,0] is at the top-left corner and [9,9] is at the bottom-right corner
- The map is shown in the bird's view
- Directions are defined as:
* up = towards the top of the grid (decreasing y-value)
* right = towards the right of the grid (increasing x-value)
* down = towards the bottom of the grid (increasing y-value)
* left = towards the left of the grid (decreasing x-value)
* inner = straight into the 2D map (perpendicular to the grid, pointing away from you)
* outer = straight out of the 2D map (perpendicular to the grid, pointing towards you)
- "objects" lists all important items in the scene with their positions
- "facing" indicates which direction an object is oriented towards (when applicable)
- "views" represents the different camera viewpoints in the scene

Below is the cognitive map of the scene related to the question. Please use it to reason and answer the question.

```json
{
  "objects": [
    {"name": "white jar", "position": [5, 5]},
    {"name": "bed sheet with a floral pattern",
        "position": [5, 8]},
    {"name": "white headboard", "position": [2, 5]},
    {"name": "clothes rack", "position": [5, 2]},
    {"name": "table with cups on it", "position": [8, 5]}
  ],
  "views": [
    {"name": "Image 1", "position": [5, 6], "facing": "up"},
    {"name": "Image 2", "position": [4, 5], "facing": "right"},
    {"name": "Image 3", "position": [5, 4], "facing": "down"},
    {"name": "Image 4", "position": [6, 5], "facing": "left"}
  ]
}
```

[Question]

Based on these four images (image 1, 2, 3, and 4) showing the white jar from different viewpoints (front, left, back, and right), with each camera aligned with room walls and partially capturing the surroundings: From the viewpoint presented in image 4, what is to the left of the white jar?

A. Table with cups on it B. Clothes rack C. Bed sheet with a floral pattern D. White headboard

# E   WHICH SCAFFOLDS BEST GUIDE SPATIAL THINKING IN UNCHANGED VLMS?

The primary experimental setup for evaluating how different structural scaffolds can guide spatial reasoning in unchanged VLMs, has been comprehensively described in Section 3 of the main paper.

This current section is dedicated to presenting qualitative examples of model outputs, serving as case studies for the different scaffolding configurations tested.

For precise details on the input prompt structures and specific instructions provided to the Qwen2.5-VL-3B-Instruct for each experimental condition (e.g., `Raw-QA`, `FFR`, `Plain-CGMap-FFR-Out`), please refer to Appendix D.3.

## E.1 VLM RESPONSE EXAMPLES FOR CONFIGURATIONS IN SECTION D.3

---

Output for `Raw-QA`: Input Prompt Refer to D.3.1

**C. Bed sheet with a floral pattern**

---

Output for `VI`: Input Prompt Refer to D.3.3

**B. Clothes rack**

---

Output for `Aug-CGMap-In`: Input Prompt Refer to D.3.4

**C. Bed sheet with a floral pattern**

---

Output for `FFR`: Input Prompt Refer to D.3.2

<think>To determine what is to the left of the white jar from the viewpoint presented in image 4, let's analyze the scene:
- In image 4, the white jar is placed on a table covered with a pink tablecloth featuring heart patterns.
- The table has a glass top and is positioned against a wall.
- To the left of the jar, there is a piece of furniture that appears to be a bed or a bed frame. Given this information, the object to the left of the white jar is most likely a bed or a bed frame. </think>
<answer>**C. Bed sheet with a floral pattern**</answer>

---

Output for `Aug-CGMap-FFR-Out`: Input Prompt Refer to D.3.8

```json
{
  "objects": [
    {"name": "white jar", "position": [150, 446],
        "facing": "left"},
    {"name": "bed sheet with a floral pattern",
        "position": [0, 317],
    "facing": "left"},
    {"name": "clothes rack", "position": [0, 108],
    "facing": "left"},
    {"name": "table with cups on it", "position": [188, 200],
    "facing": "left"},
    {"name": "white headboard", "position": [188, 317],
    "facing": "left"}
  ],
  "views": [
    {"name": "Image 1", "position": [0, 0], "facing": "left"},
```

```
      {"name": "Image 2", "position": [0, 0], "facing": "left"},
      {"name": "Image 3", "position": [0, 0], "facing": "left"},
      {"name": "Image 4", "position": [0, 0], "facing": "left"}
   ]
}
```

<CogMap>
<Your cognitive map>
<Reasoning>
From the viewpoint presented in image 4, we can see that the white jar is positioned in the foreground, facing left. The bed sheet with a floral pattern is located directly behind the white jar, covering the lower part of the image. The clothes rack is visible on the left side of the image, slightly above the bed sheet. The table with cups on it is situated further back, to the right of the white jar. The white headboard is also visible, but it is not as prominent as the other objects due to its position behind the bed sheet.
Therefore, my answer is A. Table with cups on it.
</Answer>
Therefore, my answer is **A. Table with cups on it**.

---

Output for `Plain-CGMap-FFR-Out`: Input Prompt Refer to D.3.7

```json
{
    "white_jar": {"position": [150, 440], "facing": "left"},
    "table_with_cups_on_it": {"position": [187, 260],
    "facing": "right"}
}
```

<CogMap>
<Your cognitive map>
<Reasoning>
From the viewpoint presented in image 4, we can see that the white jar is positioned on the bed sheet with a floral pattern. The table with cups on it is located to the right of the white jar, as indicated by its position in the image. The clothes rack is not visible in this particular view, and the white headboard is not directly related to the white jar or the table with cups on it. Therefore, the correct answer is A. Table with cups on it.
<Answer>
Therefore, my answer is **A. Table with cups on it**.

---

Output for `CGMap-In-FFR-Out`: Input Prompt Refer to D.3.9

**C. Bed sheet with a floral pattern**

## E.2 Additional Graph Metrics for Generated Graphs

To quantitatively assess the quality of the cognitive maps produced by our models, we employ a suite of graph-based metrics detailed in Section C.2.2. Table 5 presents a comparison of the cognitive maps generated by two of our methods: `Aug-CGMap-FFR-Out` and `Plain-CGMap-FFR-Out`.

The results highlight several key findings:

- **High Validity**: Both methods demonstrate a strong ability to generate syntactically and semantically valid maps, achieving high **Valid Rates** (**94.95%** for `Plain-CGMap-FFR-Out` and **92.57%** for `Aug-CGMap-FFR-Out`). This indicates that the models successfully learned the required JSON structure for the cognitive maps.

- **Structural Complexity**: Achieving perfect structural replication of the ground truth remains challenging, as shown by the modest **Isomorphism Rates**. The `Plain-CGMap-FFR-Out` method performs significantly better, with **7.43%** of its maps being structurally identical (isomorphic) to the ground truth, compared to a mere **0.10%** for the augmented map method.
- **Superior Similarity Performance**: A clear performance difference in semantic similarity is evident. The `Aug-CGMap-FFR-Out` method, which explicitly includes camera views, achieves a substantially higher **Overall Similarity** (**51.12%**) and is superior in representing both the relative directional relationships (**Avg. Dir. Sim.** of **43.57%**) and the correct orientation of individual objects (**Avg. Facing Sim.** of **68.75%**). In contrast, while `Plain-CGMap-FFR-Out` maintains higher validity and isomorphism, it lags behind in all three similarity metrics.

Table 5: Comparison of graph metrics for cognitive maps generated by different methods. The metrics evaluate the quality of the generated maps against the ground truth. **Valid Rate**: percentage of syntactically and semantically valid maps. **Isomorphism Rate**: percentage of maps that are structurally identical (isomorphic) to the ground truth, accounting for rotation. **Overall Sim. (Similarity)**: a weighted score combining directional and facing similarity ($S_{\text{overall}} = \alpha \cdot S_{\text{dir}} + (1-\alpha) \cdot S_{\text{face}}$). **Avg. Dir. Sim. (Average Directional Similarity)**: correctness of relative spatial relations between objects. **Avg. Facing Sim. (Average Facing Similarity)**: correctness of object orientations. All values are percentages (%).

| Method | Valid Rate | Isomorphism Rate | Overall Sim. | Avg. Dir. Sim. | Avg. Facing Sim. |
|---|---|---|---|---|---|
| Aug-CGMap-FFR-Out | 92.57 | 0.10 | 51.12 | 43.57 | 68.75 |
| Plain-CGMap-FFR-Out | 94.95 | 7.43 | 37.44 | 28.29 | 58.78 |

### E.3 FURTHER ANALYSIS ON VIEW INTERPOLATION

To rigorously assess the impact of view interpolation and ensure fair comparison, we conducted extensive additional experiments covering comparisons with optimal interpolation settings and scaling laws across model sizes. Please note that due to limitations in the raw dataset, the rotation setting does not support view interpolation beyond five steps; consequently, the experiments with up to 7-step view interpolation presented here are conducted exclusively on the around and among settings.

**Comparison with Optimal View Interpolation.** To ensure fairness, we compared our `Plain-CGMap` against the best possible performance of the View Interpolation (VI) baseline on Qwen2.5-VL-3B. As detailed in Table 6, our method (`Plain-CGMap-FFR-Out`, 47.41%) outperforms the baseline even at its peak performance (VI-1, 46.47%). This confirms that the Cognitive Map provides a structural advantage over simply increasing visual frame density. Furthermore, adding standard Free-Form Reasoning (`FFR`) to interpolated views harms performance as density increases (dropping from 45.53% to 40.77%), indicating that the bottleneck lies in the perception ability to organize visual floods rather than reasoning capacity alone.

| Method (Qwen2.5VL-3B) | VI-0 (Raw) | VI-1 | VI-2 | VI-3 | VI-4 | VI-5 | VI-6 | VI-7 |
|---|---|---|---|---|---|---|---|---|
| Plain-CGMap-FFR-Out | **47.41** | 44.94 | 44.59 | 43.18 | 44.35 | 43.41 | 42.82 | 45.28 |
| RawQA | 43.76 | **46.47** | 45.53 | 44.94 | 44.35 | 44.24 | 45.18 | 44.59 |
| FFR | 45.53 | 43.71 | 43.83 | 41.65 | 41.00 | 40.36 | 40.89 | 40.77 |

Table 6: Comparison with Optimal View Interpolation (Qwen2.5-VL-3B).

**Scaling Analysis: Does View Interpolation Scale?** We extended our evaluation to larger models, including Qwen2.5-VL-7B, Qwen3-VL-8B, Qwen3-VL-235B, and GPT-5, to test if higher capacity naturally resolves interpolation issues. Results for 7B/8B models (Table 7) show no consistent scaling law. While Qwen2.5-VL-7B benefits from density (peaking at VI-4), Qwen3-VL-8B exhibits unstable performance despite being architecturally advanced.

Moreover, for massive-scale models (Table 8), performance negatively correlates with view density. GPT-5 peaks at the sparse 1-frame setting (46.59%) and declines to 42.35% as density increases to

| Config | VI-0 | VI-1 | VI-2 | VI-3 | VI-4 | VI-5 | VI-6 | VI-7 |
|---|---|---|---|---|---|---|---|---|
| Qwen2.5-VL-7B | 37.80 | 34.90 | 35.60 | 45.30 | **47.40** | 46.50 | 46.80 | 46.80 |
| Qwen3-VL-8B | 33.80 | 36.60 | **37.60** | 35.20 | 35.30 | 33.90 | 35.80 | 35.80 |

Table 7: Scaling Analysis on Qwen2.5-VL-7B and Qwen3-VL-8B.

7 frames. Similarly, Qwen3-VL-235B drops to ∼36% with interpolation. This suggests that without structured mapping, interpolation artifacts act as noise rather than useful signals, even for SOTA models.

The view interpolation data is publicly available in our dataset repository.

| Model | VI-1 | VI-2 | VI-3 | VI-4 | VI-5 | VI-6 | VI-7 |
|---|---|---|---|---|---|---|---|
| **GPT-5** | **46.59** | 45.18 | 44.24 | 44.59 | 42.59 | 43.53 | 42.35 |
| **Qwen3-VL-235B** | **38.94** | 38.59 | 37.29 | 37.18 | 35.29 | 36.12 | 36.00 |

Table 8: Performance of Large-Scale Models (GPT-5, Qwen3-VL-235B) across view densities.

### E.4 EXPLICIT REASONING WITH VISUAL-OF-THOUGHT

**Explicit Visual Sketching via External Tools.** While text-structured maps (e.g., JSON) introduce symbolic order, they fundamentally remain implicit token sequences. To bridge this gap, we investigated whether externalizing spatial map into *explicit visual representations* could further enhance reasoning. Inspired by ViLaSR Wu et al. (2025b), we implemented a pipeline where the VLM uses an external plotting engine (Matplotlib) rather than generating text maps directly.

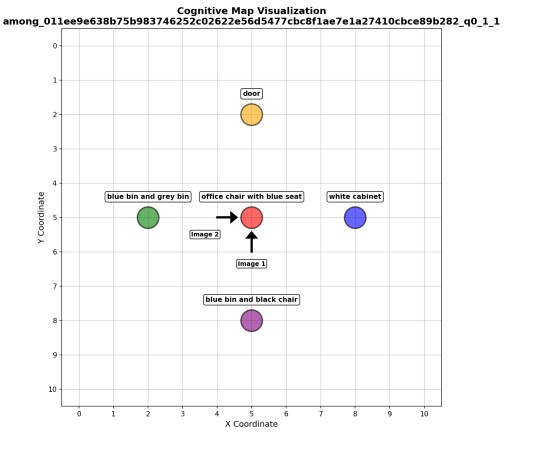 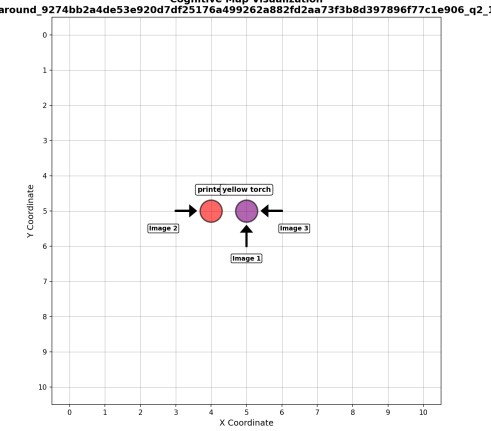

(a) Explicit Visual Map in Among Setting.  (b) Explicit Visual Map in Around Setting.

Figure 13: **Comparison of Map Representations.** The visual map, generated via external tools, renders objects and viewpoints onto a $10 \times 10$ grid, providing explicit geometric grounding.

We designed two visual-centric configurations:

1. **Visual-Map-as-Input (`Img-CGMap-In`):** Instead of ingesting raw JSON tokens, the model receives a rendered $10 \times 10$ grid image where objects and viewpoints are plotted explicitly. This tests the model's ability to comprehend provided visual layouts.

2. **Visual-Map-for-Reasoning (`Img-CGMap`):** We adopt a multi-turn tool-use framework. The model first acts as a coder to generate drawing coordinates (e.g., (obj, [x,y])),

which are executed to render a visual sketch. This sketch is then fed back to the model as a new visual prompt to guide the final spatial reasoning. **The entire multi-turn interaction pipeline and prompt design were strictly aligned with the ViSaLR framework.**

| Configuration | Map Modality | Overall | Rotation | Among | Around |
|---|---|---|---|---|---|
| Raw-QA | - | 37.81 | 34.00 | 36.00 | 45.20 |
| CGMap-In (Original) | Text (JSON) | 41.43 | **37.00** | 41.67 | 44.40 |
| **Img-CGMap-In (New)** | Visual (Grid) | **42.10** | 31.50 | **44.17** | **45.60** |
| Plain-CGMap (Original) | Text (JSON) | 41.33 | 25.00 | 39.67 | **58.40** |
| **Img-CGMap (New)** | Visual (Grid) | **43.13** | **32.75** | **41.13** | 55.06 |

Table 9: Performance comparison between implicit text-based maps (JSON) and explicit visual-based maps (Image/Grid). Visual configurations consistently outperform their textual counterparts.

**Results Analysis.** As shown in Table 9, visual map configurations consistently outperform their textual counterparts. Specifically, `Img-CGMap` achieves an overall accuracy of **43.13%**, surpassing the text-based `Plain-CGMap` (41.33%). Notably, the "Among" spatial relation benefits significantly from the visual grid (41.13% vs. 39.67%), likely because relative positioning is more intuitive in pixel space than in coordinate space. This experiment validates that leveraging external tools to create explicit visual sketches grounds the model's reasoning more effectively than symbolic text, offering a promising direction for future development.

# F    CAN WE TEACH VLMS TO BUILD AND LEVERAGE SPATIAL REPRESENTATIONS?

In the main paper, we demonstrated that prompting frozen VLMs with external scaffolds offers limited improvements. This highlighted a core limitation: the models themselves aren't effectively forming internal spatial representations or reasoning through space. To address this, we investigated whether supervised fine-tuning (SFT) could teach VLMs to build and leverage these spatial models internally. This section of the appendix provides further details on our SFT methodology, starting with the crucial step of data curation.

## F.1    SUPERVISED FINE-TUNING DATA CURATION

Effective SFT heavily relies on the quality and nature of the training data. To teach our VLMs the desired spatial reasoning capabilities, we meticulously curated two primary types of data: cognitive maps and free-form reasoning chains. These were designed to provide strong supervisory signals for the model to learn how to represent and reason about space.

### F.1.1    COGNITIVE MAP GENERATION

As discussed in Section D.1, cognitive maps serve as 2D schematic representations of object layouts. For the SFT phase, we needed to generate ground truth cognitive maps that the VLM could learn to produce. Our approach to generating these maps was grounded in the object arrangement annotations described in Section B.1. We aimed for representations that were not only accurate but also in a format that the VLM could feasibly learn to generate.

The generation process was automated via a script that processes input JSONL files, where each line item contains scene details including images and, crucially, `meta_info` describing the objects, their potential orientations, and the camera viewpoint setup. For every item, the script first identifies its specific spatial arrangement "setting" (e.g., "around," "among," "translation," or "rotation") by parsing the item's unique ID. Based on this setting, dedicated functions apply a set of predefined rules and heuristics to determine the 2D coordinates (on a 10x10 grid) and facing directions for both the objects and the camera views.

For instance, in the "around" setting, objects (typically 2-4) are placed in a predetermined linear arrangement near the grid's center (e.g., at coordinates like [4,5], [5,5]), and camera views are positioned at cardinal directions relative to these objects, based on the specific camera angles pertinent to the question. In the "rotation" setting, the camera is fixed at the center ([5,5]), and its facing direction changes across views, while object positions are defined relative to the camera's current orientation. Similar rule-based placements are implemented for "among" (objects in a cross or T-shape with views from specific angles) and "translation" (objects arranged linearly to depict relationships like "on" or "down to") settings. Object orientations, if applicable, are also assigned based on the input `meta_info`.

Finally, the generated layout of objects and views is formatted into a structured JSON string, representing the cognitive map. This JSON cogmap, along with templated instructional prompts (`cogmap_input` for VLM input format guidance and `cogmap_output` for VLM output task description), is added to the original data item. The overall generation logic is summarized in Algorithm 1.

---

**Algorithm 1** Cognitive Map Generation

---

**Require:** Dataset $D$ containing items with spatial arrangement annotations
**Ensure:** Updated dataset with cognitive maps in JSON format
1: **for all** $item \in D$ **do**
2:      $setting \leftarrow$ Extract setting type from $item.id$
3:      Initialize empty cognitive map $cogmap$
                                                              ▷ Position objects and views based on setting type
4:      **if** $setting =$ "around" **then**
5:          Position 2-4 objects in a line with coordinates like [4,5], [5,5], etc.
6:          Place views at cardinal positions based on camera angles
7:      **else if** $setting =$ "among" **then**
8:          Place center object at [5,5] and surrounding objects at [5,8], [2,5], [5,2], [8,5]
9:          Position views based on specified camera angles
10:      **else if** $setting =$ "translation" **then**
11:          Position objects according to their spatial relationships (e.g., "on", "down")
12:          Place views to highlight these spatial relationships
13:      **else if** $setting =$ "rotation" **then**
14:          Arrange objects based on rotation type (clockwise, counterclockwise, etc.)
15:          Fix camera at [5,5] with varying facing directions
16:      **end if**
                                                              ▷ Add orientation information where applicable
17:      **for all** $object \in cogmap.objects$ **do**
18:          **if** $object$ has orientation **then**
19:              Add facing direction ("up", "down", "left", "right")
20:          **end if**
21:      **end for**
22:      Format $cogmap$ as structured JSON
23:      Add formatted cognitive map to $item$
24: **end for**
25: **return** Updated dataset $D$

---

### F.1.2    FREE-FORM REASONING GENERATION

While cognitive maps provide a structured, global understanding of the scene, effective spatial reasoning also involves a procedural, step-by-step thought process. To instill this capability in our VLMs, we generated a dataset of grounded free-form reasoning chains. These chains were designed to verbalize the mental simulation process required to answer the spatial questions in MINDCUBE.

The generation of these reasoning chains was closely tied to the question-answer (QA) templates developed in Section 2. For each specific setting (e.g., rotation, among, around), we manually constructed reasoning chains following a consistent set of principles to ensure logical coherence and clear grounding in the provided visual information and the question asked.

The core principles guiding the generation of these reasoning chains were:

1. **Initial Scene Understanding.** The reasoning begins by processing each input image individually. This involves identifying key objects visible in that view and noting their explicit spatial relationships with other objects within that same view. This step emulates the initial perceptual intake a human might perform.

2. **Cross-View Consistency and Environment Integration.** After individual view analysis, the reasoning emphasizes that although different images are provided, they all depict the *same underlying spatial environment*. This is often achieved by identifying and highlighting an anchor object or a consistent set of objects that appear across multiple views, thereby helping to establish a unified mental model of the scene.

3. **Question-Driven Inference.** With a foundational understanding of the scene established from the views, the subsequent steps in the reasoning chain are directly guided by the specifics of the question. This involves: (1) **Mental Simulation:** If the question involves a hypothetical change in viewpoint or a "what-if" scenario (e.g., "what if you turn left?"), the reasoning chain explicitly verbalizes this mental transformation. (2) **Perspective Taking:** If the question requires adopting a different perspective (e.g., "from the sofa's perspective"), the reasoning chain articulates this shift. (3) **Spatial Relationship Deduction:** The chain logically deduces the queried spatial relationship by integrating information from the relevant views, applying spatial concepts (like left-of, behind, further from), and referencing the established mental model of the scene.

This structured approach to generating reasoning chains aimed to provide clear, step-by-step examples of spatial thought processes for the VLM to learn from. Figure 14, 15 and 16 show a template example combined with the filled case for ROTATION, AMONG, AROUND, respectively.

## F.2 DETAILED EXPERIMENTAL SETUP

In this section, we provide a more granular view of the experimental parameters employed during the Supervised Fine-Tuning (SFT) phase of our research. As stated in the main text, these experiments were designed to teach Vision-Language Models (VLMs) to build and leverage internal spatial representations. The base model for these SFT experiments was Qwen2.5-VL-3B-Instruct.

We utilized a consistent training script for all SFT experiments, ensuring comparability across different configurations. The primary variation across these runs was the specific dataset used (datasets variable in the script), corresponding to the different SFT task configurations discussed in Section 4.1, such as `Aug-CGMap-Out`. Other hyperparameters were kept constant to isolate the effects of the different training signals.

Table 10: Training hyperparameters for SFT experiments with Qwen2.5-VL-3B-Instruct.

| Parameter | Value |
| --- | --- |
| Dataset size | 10,000 QA pairs |
| Epochs | 3 |
| Learning rate | 1e-5 |
| Scheduler | Cosine |
| Fine-tuning type | Full-parameter |
| Batch Size | 256 |
| GPUs used | $2 \times$ NVIDIA H100 |
| Max image resolution | 90,000 pixels |
| Min image resolution | 784 pixels |
| Model Max Length | 8192 tokens |
| Weight Decay | 0 |
| Warmup Ratio | 0.03 |
| Max Grad Norm | 1 |
| Precision | BF16 |
| Optimizer | AdamW |

The core training hyperparameters are summarized in Table 10 and further detailed by the provided training script.

The training was conducted using a distributed setup managed by `torchrun` and leveraged DeepSpeed with a ZeRO Stage 3 optimization strategy for efficient full-parameter fine-tuning. Specifically, we set `NPROC_PER_NODE` to 2, utilizing two NVIDIA H100 GPUs, though the script template showed `CUDA_VISIBLE_DEVICES=0,1,2,3` and `NPROC_PER_NODE` defaulting to 4, our table and resource claims point to 2 GPUs being used for these runs. The `per_device_train_batch_size` was set to 4, and with `gradient_accumulation_steps` at 32, this resulted in an effective batch size of 256.

The learning rate was $1 \times 10^{-5}$ with a cosine learning rate scheduler and a warmup ratio of 0.03 over 3 training epochs. We enabled full fine-tuning of the vision encoder,

## Rotation – Template Type 2

### Template Reasoning Chain

This scene is observed using four images. In image 1, I can see {object_in_front_view1} as the main object in front of me. In image 2, I can see {object_in_front_view2} as the main object in front of me. In image 3, I can see {object_in_front_view3} as the main object in front of me. In image 4, I can see {object_in_front_view4} as the main object in front of me. Image 1 is the initial view. Image 2 is captured after a 90-degree clockwise rotation from image 1. Image 3 is after another 90-degree clockwise rotation (180 degrees from image 1). Image 4 is after a further 90-degree clockwise rotation (270 degrees from image 1). From the perspective of image 4: '{object_in_front_view4}' is in front, '{object_in_front_view1}' is to the right, '{object_in_front_view2}' is behind, '{object_in_front_view3}' is to the left. After turning 90 degrees to the right: '{object_in_front_view1}' is now in front, '{object_in_front_view2}' is now to my right, '{object_in_front_view3}' is now behind, '{object_in_front_view4}' is now to my left. The object located to my behind is '{object_in_front_view3}'. Therefore, from the viewpoint of image 4, after a mental turn of 90 degrees to the right, the object to my behind is '{object_in_front_view3}'. The answer is {option}

**Question:** If you are standing at the viewpoint presented in image 4 and turn 90 degrees to the right, what is to your behind ?

**Options:**

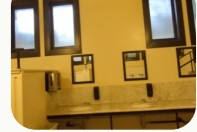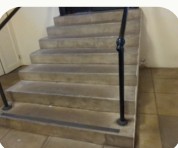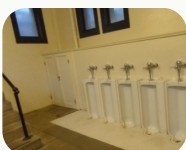

A. **Staircases**
B. Vanity unit
C. Urinal
D. Yellow signboard

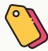 : ( rotation ) ( agent-agent ) ( self perspective ) ( non-linear )

### Curated Reasoning Chain

This scene is observed using four images. In image 1, I can see yellow signboard as the main object in front of me. In image 2, I can see vanity unit as the main object in front of me. In image 3, I can see staircases as the main object in front of me. In image 4, I can see urinal as the main object in front of me. Image 1 is the initial view. Image 2 is captured after a 90-degree clockwise rotation from image 1. Image 3 is after another 90-degree clockwise rotation (180 degrees from image 1). Image 4 is after a further 90-degree clockwise rotation (270 degrees from image 1). From the perspective of image 4: 'urinal' is in front, 'yellow signboard' is to the right, 'vanity unit' is behind, 'staircases' is to the left. After turning 90 degrees to the right: 'yellow signboard' is now in front, 'vanity unit' is now to my right, 'staircases' is now behind, 'urinal' is now to my left. The object located to my behind is 'staircases'. Therefore, from the viewpoint of image 4, after a mental turn of 90 degrees to the right, the object to my behind is 'staircases'. The answer is A. Staircases

Figure 14: Example reasoning chain template for ROTATION

## Among – Template Type 3

### Template Reasoning Chain

In this scene, I observe four images showing different perspectives. All images feature the {main_object} as the main object. In image 1, I can see {main_object} in front of the {context_obj_V1}. In image 2, I can see {main_object} in front of the {context_obj_V2}. In image 3, I can see {main_object} in front of the {context_obj_V3}. In image 4, I can see {main_object} in front of the {context_obj_V4}. By observing the main object and its surroundings across views, and noting the rotational changes, I establish their relationships. Image 1 is the initial view. Image 2 is captured after a 90-degree clockwise rotation from image 1. Image 3 is after another 90-degree clockwise rotation (180 degrees from image 1). Image 4 is after a further 90-degree clockwise rotation (270 degrees from image 1). Through analyzing these perspective changes, I construct a complete spatial understanding: When I view {context_obj_V2} behind {main_object} in the second view, it implies that in the first view, {context_obj_V2} is on the right side of {main_object}. Similarly, when I see {context_obj_V4} behind {main_object} in the fourth view, it indicates that in the first view, {context_obj_V4} is on the left side of {main_object}. To determine what lies behind me in the first view, I examine the opposite view, which is the third view. As {context_obj_V3} is observed behind {main_object} in the third view, it means that in the first view, {context_obj_V3} is positioned behind me. This way, I can fully comprehend the spatial relationships of all objects in the entire scene from the perspective of image 1. So, from the perspective of image 1: {context_obj_V2} is to the right of {main_object}, {context_obj_V3} is to my behind, and {context_obj_V4} is to the left of {main_object}. The answer is {option}.

**Question:** From the viewpoint presented in image 1, what is to the right of the black stool ?

**Options:**

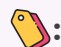
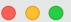

A. Desk
**B. Office Area**
C. Grey sofa
D. Two chairs on the corridor

🏷️: meanwhile   object-object   self perspective   non-linear

### Curated Reasoning Chain

In this scene, I observe four images showing different perspectives. All images feature the black stool as the main object. In image 1, I can see black stool in front of the cabinet desk along a corridor. In image 2, I can see black stool in front of the office area. In image 3, I can see black stool in front of the two chairs on the corridor. In image 4, I can see black stool in front of the grey sofa. To identify the position change across views, I focus on the main object's angle variation. Then, I analyze the angles and relative positions of other objects on the platform to back up this observation. I understand that: Image 1 is the initial view. Image 2 is captured after a 90-degree clockwise rotation from image 1. Image 3 is after another 90-degree clockwise rotation (180 degrees from image 1). Image 4 is after a further 90-degree clockwise rotation (270 degrees from image 1). Through analyzing these perspective changes, I can construct a complete spatial understanding: when I view office area behind black stool in the second view, it implies that in the first view, office area is on the right side of black stool. Similarly, when I see grey sofa behind black stool in the fourth view, it indicates that in the first view, grey sofa is on the left side of black stool. However, I am still uncertain about what lies behind me in the first view. Then, I recognize that I can examine the opposite view to find out. The opposite view of the fist view is the third view. As two chairs on the corridor is observed behind black stool in the third view, it means that in the first view, two chairs on the corridor is positioned behind me. This way, I can fully comprehend the spatial relationships of all objects in the entire scene. So, from the perspective of image 1: office area is to the right of black stool, two chairs on the corridor is to my behind, grey sofa is to the left of black stool. The answer is B. office area

Figure 15: Example reasoning chain template for AMONG

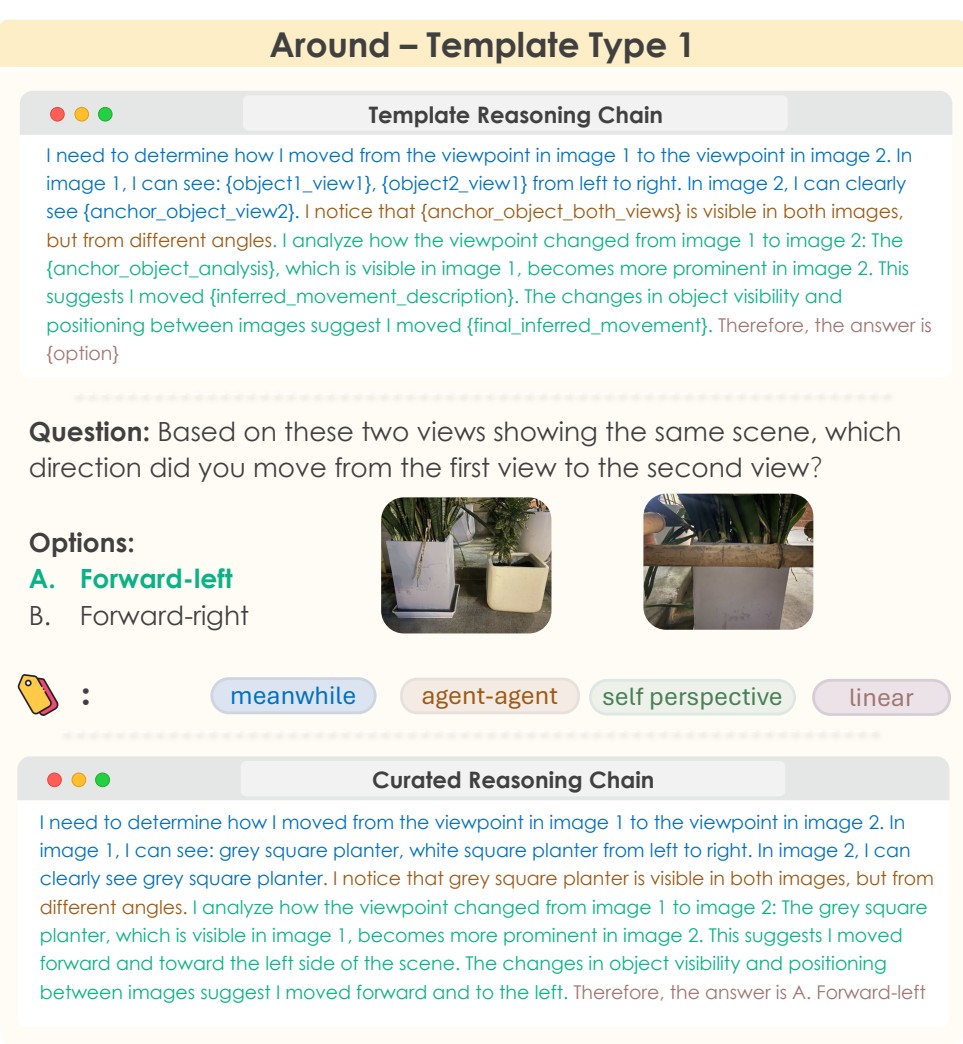

Figure 16: Example reasoning chain template for AROUND

MM-MLP, and the LLM components (`--tune_mm_vision True --tune_mm_mlp True --tune_mm_llm True`). The maximum sequence length for the model was set to 8192 tokens (`--model_max_length 8192`), and the maximum image input pixels were capped at 90,000 (`--max_pixels 90000`), with a minimum of 784. Gradient checkpointing was enabled to manage memory usage (`--gradient_checkpointing True`). Checkpoints were saved every 5 steps (`--save_steps 5`), keeping a total of 12 checkpoints (`--save_total_limit 12`).

This consistent and robust experimental framework, detailed in Sections 3.1, 3.2, and herein, allowed for a fair comparison of the different SFT strategies aimed at improving the spatial reasoning capabilities of Qwen2.5-VL-3B-Instruct.

## F.3 VLM RESPONSE EXAMPLES AFTER SFT FOR CONFIGURATIONS IN SECTION D.3

---

**Output for `Raw-QA`: Input Prompt Refer to** D.3.1

<answer>**D. White headboard**</answer>

---

**Output for `Aug-CGMap-Out`: Input Prompt Refer to** D.3.5

Based on my observation, the answer is:
<cogmap>
```json
{
  "objects": [
    {"name": "white jar", "position": [5, 5]},
    {"name": "bed sheet with a floral pattern",
        "position": [5, 8]},
    {"name": "white headboard", "position": [2, 5]},
    {"name": "clothes rack", "position": [5, 2]},
    {"name": "table with cups on it", "position": [8, 5]}
  ],
  "views": [
    {"name": "Image 1", "position": [5, 6], "facing": "up"},
    {"name": "Image 2", "position": [4, 5], "facing": "right"},
    {"name": "Image 3", "position": [5, 4], "facing": "down"},
    {"name": "Image 4", "position": [6, 5], "facing": "left"}
  ]
}
```
</cogmap><answer>**C. Bed sheet with a floral pattern**</answer>

---

**Output for `Plain-CGMap-Out`: Input Prompt Refer to** D.3.6

Based on my observation, the answer is:
<cogmap>
```json
{
    "white jar": {"position": [5, 5]},
    "bed sheet with a floral pattern": {"position": [5, 8]},
    "white headboard": {"position": [2, 5]},
    "clothes rack": {"position": [5, 2]},
    "table with cups on it": {"position": [8, 5]}
}
```

---

</cogmap><answer>**B. Clothes rack** </answer>

---

**Output for** `FFR`: Input Prompt Refer to D.3.2

<think>In this scene, I observe four images showing different perspectives. All images feature the white jar as the main object. In image 1, I can see white jar in front of the table with cups on it. In image 2, I can see white jar in front of the bed sheet with a floral pattern. In image 3, I can see white jar in front of the clothes rack. In image 4, I can see white jar in front of the white headboard. To identify the position change across views, I focus on the main object's angle variation. Then, I analyze the angles and relative positions of other objects on the platform to back up this observation. I understand that: Image 1 is the initial view. Image 2 is captured after a 90-degree clockwise rotation from image 1. Image 3 is after another 90-degree clockwise rotation (180 degrees from image 1). Image 4 is after a further 90-degree clockwise rotation (270 degrees from image 1). Through analyzing these perspective changes, I can construct a complete spatial understanding: when I view bed sheet with a floral pattern behind white jar in the second view, it implies that in the first view, bed sheet with a floral pattern is on the right side of white jar. Similarly, when I see white headboard behind white jar in the fourth view, it indicates that in the first view, white headboard is on the left side of white jar. However, I am still uncertain about what lies behind me in the first view. Then, I recognize that I can examine the opposite view to find out. The opposite view of the fist view is the third view. As clothes rack is observed behind white jar in the third view, it means that in the first view, clothes rack is positioned behind me. This way, I can fully comprehend the spatial relationships of all objects in the entire scene. So, from the perspective of image 4: table with cups on it is to the right of white jar, bed sheet with a floral pattern is to my behind, clothes rack is to the left of white jar. So the answer is B. Clothes rack</think><answer>**B. Clothes rack**</answer>

---

**Output for** `Aug-CGMap-FFR-Out`: Input Prompt Refer to D.3.8

Based on my observation, the answer is:
<cogmap>

```json
{
  "objects": [
    {"name": "white jar", "position": [5, 5]},
    {"name": "bed sheet with a floral pattern",
        "position": [5, 8]},
    {"name": "white headboard", "position": [2, 5]},
    {"name": "clothes rack", "position": [5, 2]},
    {"name": "table with cups on it", "position": [8, 5]}
  ],
  "views": [
    {"name": "Image 1", "position": [5, 6], "facing": "up"},
    {"name": "Image 2", "position": [4, 5], "facing": "right"},
    {"name": "Image 3", "position": [5, 4], "facing": "down"},
    {"name": "Image 4", "position": [6, 5], "facing": "left"}
  ]
}
```

</cogmap><think>In this scene, I observe four images showing different perspectives. All images feature the white jar as the main object. In image 1, I can see white jar in front of the table with cups on it. In image 2, I can see white jar in front of the clothes rack. In image 3, I can see white jar in front of the bed sheet with a floral pattern. In image 4, I can see white

jar in front of the white headboard. To identify the position change across views, I focus on the main object's angle variation. Then, I analyze the angles and relative positions of other objects on the platform to back up this observation. I understand that: Image 1 is the initial view. Image 2 is captured after a 90-degree clockwise rotation from image 1. Image 3 is after another 90-degree clockwise rotation (180 degrees from image 1). Image 4 is after a further 90-degree clockwise rotation (270 degrees from image 1). Through analyzing these perspective changes, I can construct a complete spatial understanding: when I view clothes rack behind white jar in the second view, it implies that in the first view, clothes rack is on the right side of white jar. Similarly, when I see white headboard behind white jar in the fourth view, it indicates that in the first view, white headboard is on the left side of white jar. However, I am still uncertain about what lies behind me in the first view. Then, I recognize that I can examine the opposite view to find out. The opposite view of the fist view is the third view. As bed sheet with a floral pattern is observed behind white jar in the third view, it means that in the first view, bed sheet with a floral pattern is positioned behind me. This way, I can fully comprehend the spatial relationships of all objects in the entire scene. So, from the perspective of image 4: clothes rack is to the right of white jar, bed sheet with a floral pattern is to my behind, table with cups on it is to the left of white jar. So the answer is A. Table with cups on it.</think><answer>**A. Table with cups on it**</answer>

---

Output for `Plain-CGMap-FFR-Out`: Input Prompt Refer to D.3.7

Based on my observation, the answer is:
<cogmap>

```json
{
    "white jar": {"position": [5, 5]},
    "bed sheet with a floral pattern": {"position": [5, 8]},
    "white headboard": {"position": [2, 5]},
    "clothes rack": {"position": [5, 2]},
    "table with cups on it": {"position": [8, 5]}
}
```

</cogmap><think>In this scene, I observe four images showing different perspectives. All images feature the white jar as the main object. In image 1, I can see white jar in front of the table with cups on it. In image 2, I can see white jar in front of the clothes rack. In image 3, I can see white jar in front of the bed sheet with a floral pattern. In image 4, I can see white jar in front of the white headboard. To identify the position change across views, I focus on the main object's angle variation. Then, I analyze the angles and relative positions of other objects on the platform to back up this observation. I understand that: Image 1 is the initial view. Image 2 is captured after a 90-degree clockwise rotation from image 1. Image 3 is after another 90-degree clockwise rotation (180 degrees from image 1). Image 4 is after a further 90-degree clockwise rotation (270 degrees from image 1). Through analyzing these perspective changes, I can construct a complete spatial understanding: when I view clothes rack behind white jar in the second view, it implies that in the first view, clothes rack is on the right side of white jar. Similarly, when I see white headboard behind white jar in the fourth view, it indicates that in the first view, white headboard is on the left side of white jar. However, I am still uncertain about what lies behind me in the first view. Then, I recognize that I can examine the opposite view to find out. The opposite view of the fist view is the third view. As bed sheet with a floral pattern is observed behind white jar in the third view, it means that in the first view, bed sheet with a floral pattern is positioned behind me. This way, I can fully comprehend the spatial relationships of all objects in the entire scene. So, from the perspective of image 4: clothes rack is to the right of white jar, bed sheet with a floral pattern is to my behind, table with cups on it is to the left of white jar. So the answer is A. Table with cups on it.</think><answer>**A. Table with cups on it**</answer>

### F.4 DETAILED GRAPH METRIC RESULTS FOR SFT GRAPH-RELATED EXPERIMENTS

This section provides a detailed look at the Supervised Fine-Tuning (SFT) training dynamics to support the main paper's conclusions. The figures below plot key metrics over training steps for four map-generation settings. A comparative analysis highlights that jointly training map generation and reasoning is the most effective strategy.

When training on map generation alone, as in the `Plain-CGMap-Out` and `Aug-CGMap-Out` settings, the graph quality metrics show rapid convergence. However, the final QA accuracy is limited, reaching 54.29% for `Plain-CGMap-Out` and 52.48% for `Aug-CGMap-Out`.

In contrast, the joint training approaches (`Plain-CGMap-FFR-Out` and `Aug-CGMap-FFR-Out`), despite a slower initial convergence on graph quality metrics, ultimately achieve far superior performance in task accuracy. The `Plain-CGMap-FFR-Out` setting proves to be the most effective, reaching a QA Accuracy of 57.81%. The `Aug-CGMap-FFR-Out` setting also yields results with QA accuracy climbing to about 54%. This demonstrates the superiority of joint training for achieving the best task accuracy, even though the generated maps are less structurally perfect than those from map-only training.

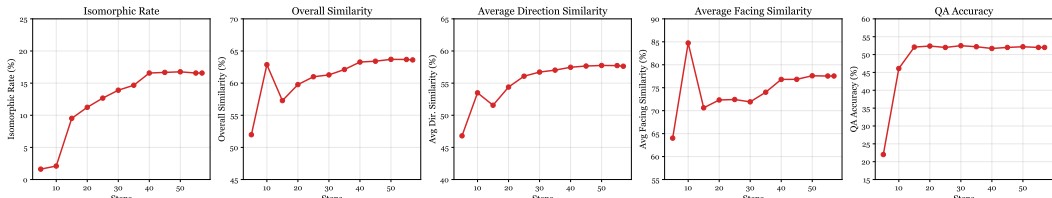

Figure 17: Training dynamics for the `Aug-CGMap-Out` setting.

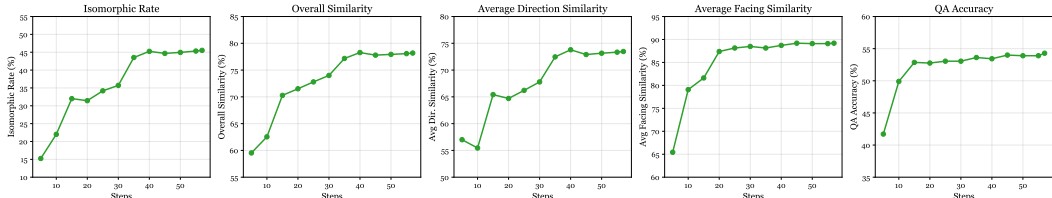

Figure 18: Training dynamics for the `Plain-CGMap-Out` setting.

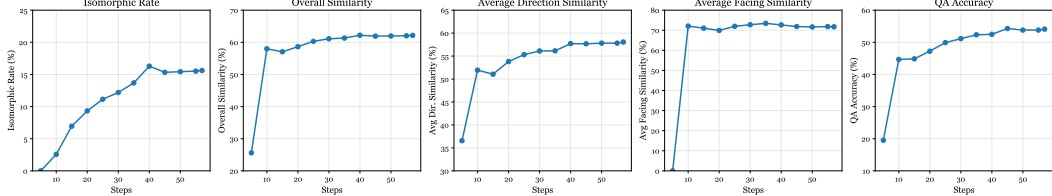

Figure 19: Training dynamics for the `Aug-CGMap-FFR-Out` setting.

### F.5 WHICH PART OF VLM IS THE BOTTLENECK FOR SPATIAL UNDERSTANDING?

To develop more efficient fine-tuning strategies, it is crucial to understand which component of a Vision-Language Model (VLM)—the vision encoder responsible for perception or the Large Language Model (LLM) responsible for reasoning—presents the primary bottleneck for spatial understanding. To investigate this, we conduct a bottleneck analysis by selectively fine-tuning different parts of the VLM and observing the impact on performance.

We evaluate four distinct training configurations on the `Raw-QA` task, with results captured at an early stage of training (step 57) to assess the initial learning dynamics. The configurations are:

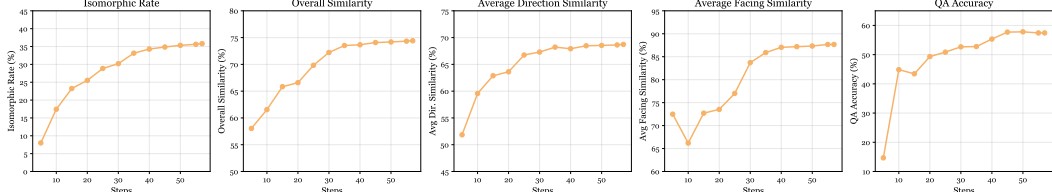

Figure 20: Training dynamics for the `Plain-CGMap-FFR-Out` setting, showing superior final performance.

(1) the baseline performance of the pre-trained model without any fine-tuning; (2) fine-tuning only the vision encoder while keeping the LLM frozen; (3) fine-tuning only the LLM while keeping the vision encoder frozen; and (4) the standard approach of fine-tuning all parts of the model.

Table 11: VLM Training Bottleneck Analysis (Step=57, in %). Performance is measured on the MINDCUBE-TINY benchmark under the Raw-QA setting.

| Training Method | Overall | Rotation | Among | Around |
|---|---|---|---|---|
| Raw-QA (no fine-tuning) | 37.81 | 34.00 | 36.00 | 45.20 |
| Freeze LLM (Vision Encoder Only) | 37.81 | 30.50 | 37.00 | 45.60 |
| Freeze Vision Encoder (LLM Only) | 51.43 | 34.00 | 50.00 | 68.80 |
| Tune All Parts | 52.67 | 34.50 | 52.50 | 67.60 |

The results, presented in Table 11, offer several key insights. First, there is a dramatic performance leap from the no-fine-tuning baseline (37.81% overall), but only when the language model is trained. Methods involving LLM fine-tuning achieve over 51% accuracy, underscoring the necessity of adapting the model's reasoning capabilities.

Most strikingly, the performance bottleneck is almost exclusively concentrated in the LLM. Tuning only the LLM (`Freeze Vision Encoder`) yields an overall accuracy of 51.43%, capturing nearly the full performance gain of end-to-end fine-tuning (52.67%). In stark contrast, tuning only the vision encoder (`Freeze LLM`) provides no improvement whatsoever over the baseline (37.81%). This indicates that the bottleneck is not shared between modules. For this spatial task, adapting the model's language-based reasoning is critical, while adapting its visual perception is surprisingly ineffective.

Intriguingly, the fact that fine-tuning only the vision encoder fails to improve performance is in itself a significant finding. A possible explanation is that the pre-trained visual features are already sufficient to extract the necessary objects and their properties. The core challenge of the task seems to lie not in *what* is seen, but in *how to reason* about the spatial relationships across a series of views—a task primarily handled by the LLM. In conclusion, our analysis suggests that the most significant gains come from adapting the reasoning module. For efficient tuning, freezing the vision encoder and focusing solely on the LLM proves to be a highly effective strategy, achieving nearly top-tier performance at a fraction of the computational cost.

## F.6 BRANCHING FROM RAW-QA SFT CHECKPOINT

In our main experiments, we fine-tuned the model for each specific task format starting from the base pre-trained VLM. A natural question arises: can a curriculum-based SFT approach further improve performance? Specifically, we investigate whether first fine-tuning the model on the simplest task format—'Raw-QA', which only requires outputting the final answer—can establish a better foundation for learning to leverage more complex reasoning formats.

To test this hypothesis, we conducted a set of branching experiments. We took the checkpoint from the model fully fine-tuned on the 'Raw-QA' task. Then, we used this specialized checkpoint as the initial weights for further fine-tuning on other scaffolding tasks, namely `Aug-CGMap-In`, FF

`Rsn`, and `Aug-CGMap-FFR-Out`. It is important to note that during this second stage of fine-tuning, the model's output for all tasks was still constrained to be only the final answer option. This setup allows us to isolate the effect of the cognitive scaffolds on the model's internal reasoning process, rather than its ability to generate complex text.

The results, presented in Table 12, show a consistent and notable improvement across all branched tasks compared to their counterparts trained from scratch. For example, both `Aug-CGMap-In` and `Aug-CGMap-FFR-Out` reach an impressive overall accuracy of 49.00%. Even the `FF Rsn` method benefits from this two-stage approach, with its overall accuracy rising to 46.82%. These findings suggest that a two-stage SFT strategy is highly effective. By first grounding the model in the fundamental objective of the task (i.e., finding the correct answer) and then teaching it to process and leverage more complex cognitive scaffolds, we can achieve superior spatial reasoning performance. This indicates that the model, once primed for the core task, becomes more adept at utilizing the provided spatial context, even if it does not explicitly generate the reasoning chain or cognitive map.

Table 12: Performance of various methods after being fine-tuned from a `Raw-QA` SFT checkpoint. This two-stage training approach led to performance gains across all methods. All accuracies are reported as percentages (%).

| Method | Overall | Rotation | Among | Around |
|---|---|---|---|---|
| Raw-QA | 46.36 | 33.50 | 51.20 | 46.75 |
| Aug-CGMap-In | 49.00 | 35.50 | 53.20 | 50.50 |
| FF Rsn | 46.82 | 37.00 | 50.60 | 47.00 |
| Aug-CGMap-FFR-Out | 49.00 | 37.00 | 53.20 | 49.75 |

### F.7 HYPERPARAMETER TUNING RESULTS

We conducted new hyperparameter tuning experiments to further validate our approach. Specifically, we performed a series of experiments to tune the hyperparameters for our Supervised Fine-Tuning (SFT) settings, as detailed in Table 13.

As shown in the table, our original hyperparameter configuration of a learning rate of $10^{-5}$, a batch size of 512, and a warmup ratio of 0.03 yielded the highest accuracy of 52.67%. The results from these experiments confirm that the hyperparameters used in our initial submission are effective and well-optimized for the task, further substantiating our claims.

Table 13: SFT Hyperparameter Tuning Results

| SFT (learning rate, batch size, warmup ratio) | Acc (%) |
|---|---|
| $(1 \times 10^{-5}, 512, 0.03)$ – Ours | **52.67** |
| $(2 \times 10^{-5}, 512, 0.03)$ | 51.71 |
| $(4 \times 10^{-5}, 512, 0.03)$ | 51.52 |
| $(1 \times 10^{-5}, 256, 0.03)$ | 50.86 |
| $(1 \times 10^{-5}, 1024, 0.03)$ | 51.90 |
| $(1 \times 10^{-5}, 512, 0.01)$ | 51.81 |
| $(1 \times 10^{-5}, 512, 0.10)$ | 50.67 |

### F.8 EFFECT OF REMOVING VIEWPOINT DESCRIPTORS

**Motivation.** In the original formulation of *Qwen2.5-VL-3B-Instruct* questions, we include coarse viewpoint descriptors (e.g., "back", "left", "right") in the textual prompt to indicate that the input images correspond to canonical, orthogonal views rather than arbitrary camera poses. This design choice was initially made for clarity and simplicity, especially given that earlier-generation VLMs exhibited near-random performance and were unable to reliably exploit such directional cues.

However, as model capabilities improve, these textual descriptors may introduce unintended shortcuts: models could partially infer spatial relations from language alone without fully aligning multi-view visual observations. To assess this effect, we construct a variant of our evaluation setting where all viewpoint descriptors are removed from the question text, forcing models to rely purely on visual consistency across views. We evaluate this variant under both frozen inference and supervised fine-tuning (SFT) settings, while keeping all other components unchanged.

**Results.** As shown in Table 15 and Table 14, the overall trends from the main paper are preserved under both frozen and SFT settings. In the frozen setting, the map-then-reason configuration (`Plain-CGMap-FFR-Out`) achieves the highest overall accuracy (42.67%), outperforming `Raw-QA` (38.57%) by 4.1 points. Notably, this relative gain is more pronounced than in the setting with viewpoint descriptors, suggesting that explicit spatial scaffolding plays a more critical role when textual shortcuts are unavailable.

Under SFT (Table 15), all configurations converge to a narrower performance range, with `Plain-CGMap-FFR-Out` maintaining a slight edge (best: 54.0%) over `Raw-QA` (best: 52.8%). The reduced gap suggests that without viewpoint descriptors, the task places greater demands on fine-grained visual alignment, making it harder for models to learn robust spatial representations from limited supervised data alone. Nevertheless, the map-then-reason approach remains the top-performing configuration across both evaluation settings, consistent with our main findings.

Table 14: QA accuracy (%) of *Qwen2.5-VL-3B-Instruct* on MINDCUBE-TINY **without viewpoint descriptors** under different configurations for frozen VLMs.

| Config. | Overall | Rotation | Among | Around |
|---|---|---|---|---|
| Raw-QA | 38.57 | 34.00 | 36.83 | 46.40 |
| Aug-CGMap-In | 32.00↓ | 34.50 | 30.50 | 33.60 |
| FFR | 39.14↑ | 32.50 | 33.83 | 57.20 |
| Aug-CGMap-FFR-Out | 38.95↑ | 19.50 | **41.00** | 49.60 |
| Plain-CGMap-FFR-Out | **42.67**↑ | 28.50 | 39.83 | **60.80** |
| CGMap-In-FFR-Out | 40.67↑ | **34.50** | **42.67** | 40.80 |

### F.9 WHY RANDOMIZED OBJECT ORDER BETTER EVALUATES GENUINE SPATIAL UNDERSTANDING

This subsection extends the discussion in Section 4.4 on the choice of object presentation order in cognitive map supervision. Under fixed spatial order, models achieve notably higher map reconstruction quality (e.g., 89.05% vs. 45.52% isomorphism rate for `Plain-CGMap-Out`), as the deterministic sequence provides a predictable pattern that the model can exploit without necessarily developing robust internal spatial representations. In contrast, randomized order forces the model to construct its own spatial understanding from scratch for each example, without relying on ordering shortcuts. We therefore adopt randomized order as our primary setting (Table 4), as it more faithfully reflects the model's ability to build genuine internal representations—the central goal of our investigation. We present the fixed-order results in Section 4.4 for completeness, noting that the core conclusions hold under both settings.

## G CAN REINFORCEMENT LEARNING FURTHER REFINE SPATIAL THOUGHT PROCESSES?

As discussed in the main paper, while Supervised Fine-Tuning (SFT) establishes a strong foundation for spatial reasoning, reinforcement learning (RL) presents an avenue for further optimizing spatial thought processes through outcome-driven feedback. The core inquiry is whether guiding VLMs with rewards can lead to the development of more precise spatial mental models and enhanced reasoning capabilities. This section of the appendix provides a more detailed exposition of the experimental setup employed for the RL phase of our research. Additionally, we present case studies to offer qualitative insights into how RL refines the models' spatial representations and reasoning chains.

Table 15: Best QA accuracy (%) across SFT training steps **without viewpoint descriptors**. Each value reports the peak accuracy observed during training (up to step 57).

|            | Raw-QA | Aug-CGMap-Out | Plain-CGMap-Out | FFR  | Aug-CGMap-FFR-Out | Plain-CGMap-FFR-Out |
|------------|--------|---------------|-----------------|------|-------------------|---------------------|
| **Best Acc.** | 52.8   | 51.6          | 52.8            | 52.2 | 53.8              | **54.0**            |
| **@ Step**    | 45     | 50            | 45              | 55   | 20                | 25                  |

## G.1 DETAILED EXPERIMENTAL SETUP

For the reinforcement learning (RL) phase of our research, we employed the VAGEN framework. The core policy optimization algorithm used was Group Relative Policy Optimization (GRPO). To ensure consistency and allow for direct comparison with earlier stages of our work, key components from the Supervised Fine-Tuning (SFT) experiments were retained. Specifically, the base Vision-Language Model (VLM) for all RL configurations was Qwen2.5-VL-3B-Instruct, and evaluations were performed on the MINDCUBE-TINY benchmark. All previously established evaluation metrics were also retained.

In consideration of computational costs, each distinct RL configuration was trained for a duration of 0.5 epoch. The primary hyperparameters governing the RL training process were set as follows:

- **Training Batch Size:** 32
- **Maximum Prompt Length:** 1024 tokens
- **Maximum Response Length:** 512 tokens
- **Actor Learning Rate:** $1 \times 10^{-6}$
- **Critic Learning Rate:** $1 \times 10^{-5}$
- **Number of Trajectories per Rollout:** 8
- **Maximum Turns per Trajectory:** 1

As detailed in Section 5.1 of the main paper, we investigated three RL task configurations:

1. **RL-FFR (from scratch):** The Qwen2.5-VL-3B-Instruct model was trained to generate free-form reasoning chains without prior SFT for this specific task format.

2. **RL-Aug-CGMap-FFR-Out (from scratch):** The model was trained to jointly produce augmented cognitive maps and associated free-form reasoning, also starting from the base pre-trained VLM.

3. **RL-Aug-CGMap-FFR-Out (from SFT):** For this configuration, the RL training was initialized using the weights from the strongest performing SFT checkpoint, specifically the Aug-CGMap-FFR-Out SFT model.

The reward function was designed to be sparse yet directly indicative of desired behaviors. A reward of +1 was assigned if the model's output was structurally valid (e.g., the generated cognitive map adhered to the predefined schema). A more significant reward of +5 was given if the model provided the correct final answer to the spatial reasoning question. This dual-component reward was intended to steer the learning process towards generating both well-formed intermediate spatial representations and achieving high end-task accuracy.

## G.2 RL REWARD DESIGN ABLATION

A comprehensive reward ablation study can provide more convincing evidence for our approach. Inspired by the Deepseek-R1 approachGuo et al. (2025), our reward design incorporates a dual-component structure: a small score assigned for valid format generation and a larger score for correct answer generation. This design encourages both structural correctness and factual accuracy in the model's output.

To investigate the efficacy of this reward structure, we conducted an ablation study with three different reward configurations, as detailed in Table16. Our baseline, represented by a frozen Visual-Language Model (VLM), achieved an accuracy of 40.57%. Our proposed setting, which assigns +1 for a valid format and +5 for a correct answer, achieved an accuracy of 49.52%. The "Answer Only" setting, which assigns +5 for a correct answer but 0 for format, resulted in a similar accuracy of

50.76%. In contrast, the "Format Only" setting, which provides a score of +5 for a valid format but 0 for the answer, yielded a lower accuracy of 45.05%.

These results confirm that the reward component for a correct answer is the most critical factor in our RL reward design. The high value assigned to the correct answer score (+5) is highly effective at improving the model's performance, a finding that is consistent with recent work on RL for Large Language Models.

Table 16: RL Reward Ablation Study

| RL Reward Setting (format score, answer score) | Acc (%) |
|---|---|
| Frozen VLM (baseline) | 40.57 |
| Our Setting (+1, +5) | 49.52 |
| Answer Only (+0, +5) | 50.76 |
| Format Only (+5, +0) | 45.05 |

## G.3 VLM RESPONSE EXAMPLES AFTER RL FOR CONFIGURATIONS IN SECTION D.3

Output for `FFR`: Input Prompt Refer to D.3.2

<think>From image 4, the white jar is positioned on a surface with a floral-patterned bed sheet. To the left of the jar, there is a table with a pink tablecloth featuring heart shapes and a Hello Kitty design. There are also cups on the table. Therefore, the object to the left of the white jar is the table with the cups.</think><answer>**A. Table with cups on it**</answer>

Output for `RL-Aug-CGMap-FFR-Out (from scratch)`: Input Prompt Refer to D.3.8

```
<cogmap>
{
  "objects": [
    {"name": "white jar", "position": [265, 436],
        "facing": "left"}
  ],
  "views": [
    {"name": "View/Image 1", "position": [265, 436],
      "facing": "left"},
    {"name": "View/Image 2", "position": [265, 436],
      "facing": "left"},
    {"name": "View/Image 3", "position": [265, 436],
      "facing": "left"},
    {"name": "View/Image 4", "position": [265, 436],
      "facing": "left"}
  ]
}
```
<Your cognitive map></cogmap>
<think>The white jar is located in the center of the 10x10 grid, facing to the left. The clothes rack is to the left of the white jar, as seen in the image 2 view. The bed sheet with a floral pattern is located to the right of the white jar, as seen in the image 1 view. The white headboard is not visible in any of the provided images.
Based on the provided images, the white jar is located in the center of the 10x10 grid, facing to the left. The clothes rack is to the left of the white jar, as seen in the image 2 view. The bed sheet with a floral pattern is located to the right of the white jar, as seen in the image 1 view. The white headboard is not visible in any of the provided images.</think><answer>**A. Table with cups on it**.</answer>

---

**Output for** `RL-Aug-CGMap-FFR-Out (from SFT)`**: Input Prompt Refer to** D.3.8

Based on my observation, the answer is:
<cogmap>

```json
{
  "objects": [
    {"name": "white jar", "position": [5, 5]},
    {"name": "bed sheet with a floral pattern",
        "position": [5, 8]},
    {"name": "white headboard", "position": [2, 5]},
    {"name": "clothes rack", "position": [5, 2]},
    {"name": "table with cups on it", "position": [8, 5]}
  ],
  "views": [
    {"name": "Image 1", "position": [5, 6], "facing": "up"},
    {"name": "Image 2", "position": [4, 5], "facing": "right"},
    {"name": "Image 3", "position": [5, 4], "facing": "down"},
    {"name": "Image 4", "position": [6, 5], "facing": "left"}
  ]
}
```

</cogmap><think>In this scene, I observe four images showing different perspectives. All images feature the white jar as the main object. In image 1, I can see white jar in front of the clothes rack. In image 2, I can see white jar in front of the table with cups on it. In image 3, I can see white jar in front of the bed sheet with a floral pattern. In image 4, I can see white jar in front of the white headboard. To identify the position change across views, I focus on the main object's angle variation. Then, I analyze the angles and relative positions of other objects on the platform to back up this observation. I understand that: Image 1 is the initial view. Image 2 is captured after a 90-degree clockwise rotation from image 1. Image 3 is after another 90-degree clockwise rotation (180 degrees from image 1). Image 4 is after a further 90-degree clockwise rotation (270 degrees from image 1). Through analyzing these perspective changes, I can construct a complete spatial understanding: when I view table with cups on it behind white jar in the second view, it implies that in the first view, table with cups on it is on the right side of white jar. Similarly, when I see white headboard behind white jar in the fourth view, it indicates that in the first view, white headboard is on the left side of white jar. However, I am still uncertain about what lies behind me in the first view. Then, I recognize that I can examine the opposite view to find out. The opposite view of the fist view is the third view. As bed sheet with a floral pattern is observed behind white jar in the third view, it means that in the first view, bed sheet with a floral pattern is positioned behind me. This way, I can fully comprehend the spatial relationships of all objects in the entire scene. So, from the perspective of image 4: clothes rack is to the right of white jar, table with cups on it is to my behind, bed sheet with a floral pattern is to the left of white jar. So the answer is C. Bed sheet with a floral pattern.</think><answer>**C. Bed sheet with a floral pattern**</answer>

