# OpenReview forum: "MindCube: Spatial Mental Modeling from Limited Views"
_ICLR.cc/2026/Conference — ICLR 2026 Poster_

### Official Review · Reviewer_tDRP · 2025-10-25

**Soundness:** 3
**Presentation:** 3
**Contribution:** 3
**Rating:** 6
**Confidence:** 3

**Summary:**

This paper introduces MindCube, a new benchmark for evaluating the ability of VLMs to form "spatial mental models" from limited views. The authors show that existing models perform poorly, then propose a "map-then-reason" approach, which is to train a model to first generate a structured cognitive map and then reason upon it. This method leads to performance improvement through a combination of supervised fine-tuning and reinforcement learning.

**Strengths:**

1. High-quality benchmark and problem formulation. The proposed benchmark MindCube is well-designed and challenging, which systematically probes the critical VLM weakness of reasoning about unobserved space from multi-views.
2. Systematic and rigorous experiments: The authors conduct a thorough investigation into different scaffolding methods with different input and output settings, logically progressing from frozen to trained models via SFT and RL.
3. The "map-then-reason" paradigm yields a huge performance improvement, providing strong evidence for its efficacy.

**Weaknesses:**

1. Simplified "Cognitive Map" Representation: The paper's "cognitive map" is a practical choice but is a simplification of the richer cognitive science concept with JSON format, which may limit its generalizability towards more complex scenarios.
2. Some Shaky Conclusions in the Paper: Regarding experiments, some of the claims remain quite shaky. For example, the conclusion that cognitive maps guide reasoning (L315) in frozen models is based on a marginal performance gain ( from 40.48 to 41.43, less than 1%, 10 data points) that is likely not statistically significant, especially considering the performance variance reported across seeds (in Table 14).

**Questions:**

1. The interesting finding that view interpolation is ineffective is not deeply analyzed. Could the authors elaborate on the reason for this failure? Is it due to the quality of the interpolated views, the increased context length, or is it further evidence that the primary bottleneck is reasoning, not perception?
2. Line 391 states a performance gain of "51.09%" for FF-Rsn, but this number does not appear in Table 4 (which shows 53.52%). Could the authors please clarify the source of this value?

I'm happy to adjust the scores if the author can address the concerns in the weaknesses and questions.

---

> ### Author Response · Authors · 2025-11-21
> **Responses to Reviewer tDRP (1/2)**
>
> We sincerely thank the Reviewer tDRP for their detailed feedback and for recognizing our work as a "high-quality benchmark" that is "well-designed and challenging". We appreciate their acknowledgment of our "systematic and rigorous experiments" and the assessment that our "map-then-reason" paradigm yields a "huge performance improvement" providing "strong evidence for its efficacy".
>
> For weaknesses and questions, we address each point below.
>
> ---
> ## Responses to Weaknesses
> 1. **[W1] On the Justification of JSON Scaffold Less Generalizable.**
>
>     We thank the reviewer for acknowledging the practicality of our design. We agree that our JSON map is a simplified abstraction compared to biological mental models. However, we argue that this specific abstraction enhances, rather than limits, generalizability to complex scenarios for modern VLMs.
>     + **Abstraction as a Prerequisite for Complexity.**
>         + In complex, unstructured environments, the risk of hallucination increases.
>         + Cognitive science suggests that humans manage complexity by abstracting sensory data into symbolic structures ("chunks") [1].
>         + Similarly, our JSON scaffold acts as a "**Minimum Viable Structure**" that forces the model to ground complex visual inputs into discrete entities.
>         + If a model cannot organize a scene into a basic JSON structure (which we did find in our benchmark), **it is unlikely to generalize to unstructured "fuzzy" reasoning without drifting**.
>     + **Generalizability to Downstream Agentic Tasks.**
>         + The JSON format was chosen not just for simplicity, but for computability.
>         + As the standard syntax for tool-use and code agents, JSON allows our "cognitive map" to generalize immediately to downstream robotics and planning tasks (e.g., passing coordinates to a navigation API).
>         + A "richer" but non-standard cognitive representation might be biologically accurate but computationally isolated.
>     + **Empirical Robustness (Ablation Study).**
>         + To verify if this simplification hurts performance, we tested "richer" formats (like natural language descriptions) and "stricter" formats (like symbolic logic).
>         + As shown in the table below, the JSON format generalizes best to the task, outperforming natural language and symbolic logic.
>         + This suggests that for current architectures, JSON is the optimal "middle ground" for generalizability.
>
>         | Representation Format           | Accuracy (%) |
>         |---|--------------|
>         | **Map (JSON)**          | **41.3**    |
>         | Natural Language Grounding     | 39.7        |
>         | Symbolic Scene Graph (Plain)   | 37.9        |
>         | Symbolic Scene Graph (JSON)    | 40.4        |
>         | Map (YAML) | 36.1        |
>
>     + **Extensibility to Complex Semantics.**
>         + The JSON schema is **inherently extensible**.
>         + While our current benchmark focuses on topology and orientation, the same schema can be **easily extended** to include complex representations (e.g., textures, affordances, 6-DoF poses) without changing the fundamental reasoning architecture.
> 2. **[W2] On Justification of Report Robustness for Frozen Settings.**
>
>     We appreciate the reviewer’s rigorous examination of the experimental results. We agree that the performance gap between FF-Rsn (40.48%) and CGMap-In-FFR-Out (41.43%) is marginal. However, our main claim still holds.
>     + Specifically, in L301-304, the paper stated: "In contrast, enabling free-form reasoning (FF-Rsn) alone or combined ... boost to 41.33% ... Without engaging reasoning, VLMs struggle to leverage even well-formed spatial cues to improve spatial mental models."
>     + Our core finding in this section is not that "Map inputs drastically outperform pure reasoning in frozen models," but rather the contrast regarding how VLMs process structure.
>         1. **Structure Alone Fails.** When providing the map without reasoning (`Aug-CGMap-In`), performance significantly drops to 32.00% (p=0.0008 vs. Baseline). This statistically confirms that frozen VLMs cannot passively "read" complex spatial structures.
>         2. **Reasoning "Unlocks" Structure.** When reasoning is enabled (`CGMap-In-FFR-Out`), the model recovers from this failure and achieves the best overall performance (41.43%, p=0.05 vs. Baseline).
>     + We acknowledge that the phrase "cognitive maps guide reasoning" in L315 (now is L332) would be more accurately framed as **"Reasoning acts as a necessary mechanism to ground spatial structure in frozen settings".**
>     + We **have revised the manuscript** in our latest PDF.
>
> [1] Miller, George A. "The magical number seven, plus or minus two: Some limits on our capacity for processing information." Psychological review 1956

---

> ### Author Response · Authors · 2025-11-21
> **Responses to Reviewer tDRP (2/2)**
>
> ## Response to Questions
> 1. **[Q1] Deeper Analysis over View Interpolation**
>
>     We thank the reviewer for this probing question. To provide a deeper analysis of why view interpolation yields inconsistent results, we expanded our evaluation to include models of varying capacities (**Qwen2.5-VL-3B/7B** and **Qwen3-VL-4B/8B**) across varying view densities (0 to 7).
>
>     | Config        | RawQA-0 | RawQA-1 | RawQA-2 | RawQA-3 | RawQA-4 | RawQA-5 | RawQA-6 | RawQA-7 |
>     | ------------- | ------- | ------- | ------- | ------- | ------- | ------- | ------- | ------- |
>     | Qwen2.5-VL-3B | 43.80   | 46.50   | 45.50   | 44.90   | 44.40   | 44.20   | 45.20   | 44.60   |
>     | Qwen2.5-VL-7B | 37.80   | 34.90   | 35.60   | 45.30   | 47.40   | 46.50   | 46.80   | 46.80   |
>     | Qwen3-VL-4B   | 25.90   | 26.90   | 29.60   | 30.00   | 30.20   | 31.20   | 30.20   | 31.20   |
>     | Qwen3-VL-8B   | 33.80   | 36.60   | 37.60   | 35.20   | 35.30   | 33.90   | 35.80   | 35.80   |
>
>     Based on these results, we address the three specific hypotheses raised by the reviewer:
>     1. **Is it the Quality of Interpolated Views? (Unlikely)**
>         + **Evidence:** If the interpolated views contained severe artifacts or distortions, performance should degrade linearly for *all* models as more views are added.
>         + **Observation:** The Qwen2.5-VL-7B model contradicts this. Its performance surges from ~35% (sparse) to 47.40% (dense/4 views).
>         + This proves that the generated views contain sufficient and consistent geometric information.
>         + The failure of other models to utilize this information is not due to the pixel-level quality of the frames.
>     2. **Is it Increased Context Length? (Yes, for specific models)**
>         + **Evidence:** We observe that context length budget becomes a limiting factor for models with lower contextual robustness.
>         + **Observation:** For Qwen2.5-VL-3B and Qwen3-VL-8B, performance peaks early (at 1 or 2 views) and then degrades or plateaus as density increases.
>         + For these architectures, the expanded visual context eventually overwhelms the attention mechanism.
>     3. **Is the Primary Bottleneck Reasoning or Perception? (A Dynamic Trade-off)**
>         + **Perception is the Initial Bottleneck**. The universal gain from `RawQA-0` to `RawQA-1` across all models confirms that single-view perception is insufficient. Models need some extra views to uncover blind spots.
>         + **Integration Reasoning is the Ultimate Bottleneck**. As view density increases, the bottleneck shifts to Spatial Integration.
>             + Robust Reasoners (e.g., Qwen2.5-7B): Can aggregate dense signals into a coherent model.
>             + Fragile Reasoners (e.g., Qwen3-8B): Fail to synthesize the redundant information, suffering from interference (37.60 $\to$ 35.20).
>
>     + Overall, `View interpolation` is not a golden scaffold because the ability to integrate continuous views is **highly model-dependent and unstable**.
>     + While it solves the initial perception blind spots, it introduces a new reasoning bottleneck (integration) that many VLMs fail to overcome.
>     + This instability reaffirms the necessity of our proposed **Structural Prior** (Cognitive Map), which creates a stable intermediate representation regardless of the raw view density.
>
> 2. **[Q2] Clarification of a Typo.**
>
>     We apologize for this oversight and thank the reviewer for the careful reading.
>
>     + **The value in Table 4 (53.52%) is the correct**, final experimental result.
>     + The number Line 391 (now Line 413) (51.09%) was a legacy value from an earlier iteration of our experiments that was inadvertently missed during the text update.
>     + **We have corrected the text in the revised manuscript to match Table 4.** Furthermore, we have conducted a thorough consistency check across all text and tables to ensure no other such discrepancies remain.

---

> > ### Comment · Reviewer_tDRP · 2025-11-23
> >
> > Thanks for the detailed response. I appreciate the additional experiment results and analysis for view interpolation. My concerns in the weaknesses and questions have been mostly addressed. I hope the authors can include the deeper analysis over view interpolation in the camera-ready version. I've increased the score to 8.

---

> > > ### Author Response · Authors · 2025-11-23
> > > **Thank you Reviewer tDRP for increasing your rating**
> > >
> > > Thank you for reviewing our rebuttal and increasing your rating to 8. We are happy to address any further concerns you may have. We sincerely appreciate your time and effort.
> > >
> > > Thank you,
> > >
> > > Authors

---

### Official Review · Reviewer_WY9r · 2025-10-27

**Soundness:** 3
**Presentation:** 3
**Contribution:** 3
**Rating:** 6
**Confidence:** 4

**Summary:**

This paper addresses the critical failure of VLMs in forming spatial mental models: the ability to reason about unseen spaces from limited views. To systematically evaluate this gap, the authors introduce the MINDCUBE benchmark, demonstrating that existing models perform near-randomly on such tasks. The paper's key contribution is a synergistic "map-then-reason" approach, which trains a model to first generate an internal cognitive map of an environment and then reason upon it. This method actively constructs and uses an internal spatial representation, especially when refined with reinforcement learning, can significantly boost task accuracy from a baseline of 37.8% to 70.7%.

**Strengths:**

1. The paper originally frames the issue of spatial reasoning through the cognitive science concept of "spatial mental models" and introduces MINDCUBE, a high-quality benchmark designed to rigorously diagnose this fundamental capability gap in current VLMs.

2. The "map-then-reason" approach is a highly effective solution, significantly outperforming methods that use pre-supplied maps or unstructured reasoning.

3. The study is distinguished by its rigorous and comprehensive experiments, including a critical bottleneck analysis that pinpoints the language reasoning module as the primary weakness, providing a robust foundation for its conclusions and clear direction for future research.

**Weaknesses:**

1. The proposed cognitive map is a significant oversimplification of the flexible and distorted mental models from cognitive science, potentially teaching the model a brittle form of reasoning that may not generalize to less structured environments.

2. The benchmark's reliance on static images evaluates the one-time synthesis of a mental model but not its crucial ability to dynamically update from a continuous visual stream, which may limit the relevance of the findings for real-world embodied agents that require ongoing spatial awareness.

**Questions:**

The paper's fine-tuning success is demonstrated exclusively on a single model family, leaving its generalizability as an open question. Have the authors tried other architecturally distinct VLMs?

---

> ### Author Response · Authors · 2025-11-21
> **Responses to Reviewer WY9r (1/2)**
>
> We sincerely thank Reviewer WY9r for their detailed feedback and for recognizing our work as a "high-quality benchmark designed to rigorously diagnose this fundamental capability gap" with "rigorous and comprehensive experiments." We appreciate the acknowledgment that our proposed "map-then-reason" approach is a "highly effective solution", supported by a "critical bottleneck analysis".
>
> For weaknesses and questions, we address each point below.
>
> ---
> ## Responses to Weaknesses
> 1. **[W1] On Justification of Scaffold Less Generalizable.**
>
>     While we acknowledge that our JSON-based map is an abstraction compared to biological cognition, we argue that this is a **deliberate design choice** that balances **benchmarking rigor** with **representational robustness**. Far from being brittle, this object-centric abstraction serves as an effective computational proxy for spatial grounding.
>     + **Our goal is to benchmark fundamental capabilities first.** We posit that before VLMs can handle the "distorted and flexible" models required for unstructured scenes, they must first demonstrate competence in **regularized spatial reasoning**. If a model fails to construct a coherent map in a relatively structured setting (as our benchmark tests, which did show the incompetence of current VLMs), *it is unlikely to succeed in highly unstructured or noisy environments.* Our benchmark serves as a foundational testbed for reasoning in unseen space.
>     + **Structured abstractions are robust even in complex daily environments.** Contrary to the concern that such maps are brittle, recent literature supports their efficacy:
>         + By representing the environment as semantic entities with coordinates, cognitive maps effectively construct an *implicit scene graph* where spatial relationships are dynamically derivable from geometry. Therefore, unlike binary edges, **continuous coordinates ensure robustness** against visual distortion.
>         + A body of recent work [1, 2, 3, 4] demonstrates that cognitive maps and semantic scene graphs are **effective for reasoning in complex, cluttered scenes**.
>         + This suggests that such structured representations are **a proven mechanism** for **generalizing** across diverse and messy real-world layouts.
>     + **Why JSON? Because it works as a native interface for VLMs.** VLMs fundamentally operate on sequences of language tokens. JSON is a text-based, semi-structured format that fits naturally with the transformer’s generation mechanism. It bridges the gap between:
>         + **Rigidity**: Providing a computationally evaluable format (unlike free-form text).
>         + **Flexibility**: Allowing the model to dynamically generate nodes and edges without external parsers, a standard practice in recent tool-use and agentic literature [1, 2, 3, 4].
>     + Beyond traditional cognitive maps like [1], our cognitive map explicitly encodes **3D semantic properties**, such as the orientation of objects (e.g., the facing direction of a TV or mirror).
>     + Additionally, representations such as object 3D bounding box and its 6-DoF poses, despite being interesting and more flexible, often **require extensive human resources** to fully annotate for large-scale, in-the-wild image datasets, given the limited resources of academic labs.
>     + Our JSON cogmap representation allows us to build a large-scale, diverse benchmark with **accurate human verification**, while remaining **easily extensible to richer formats** (e.g., 3D bounding boxes) in future iterations.
>
>     In short, we view the JSON cognitive map not as a replacement for biological cognition, but as **a robust computational proxy** that allows us to systematically measure and equip models with the prerequisite structural understanding to reason in space.
>
> [1] Yang, Jihan, et al. "Thinking in space: How multimodal large language models see, remember, and recall spaces." Proceedings of the CVPR 2025
>
> [2] Zhang, Yi, et al. "EmbodiedVSR: Dynamic Scene Graph Guided Chain-of-Thought Reasoning for Visual Spatial Tasks." arXiv 2025
>
> [3] Ouyang, Kun, et al. "SpaceR: Reinforcing MLLMs in Video Spatial Reasoning." arXiv 2025
>
> [4] Mingcong, Lei, et al. "RoboMemory: A Brain-inspired Multi-memory Agentic Framework for Interactive Environmental Learning in Physical Embodied Systems."  arXiv 2025

---

> > ### Author Response · Authors · 2025-11-21
> > **Responses to Reviewer WY9r (2/2)**
> >
> > 2. **[W2] On Justification of Static Scenes.**
> >
> >     We thank the reviewer for highlighting the importance of dynamic spatial awareness. We fully agree that updating spatial perception from continuous streams is critical, a direction well-explored by benchmarks like [1, 2].
> >
> >     However, we respectfully argue that constructing spatial consistency from limited, distinct views (**Sparse View Synthesis**) is an equally fundamental and distinct pillar of spatial intelligence.
> >     + **Complementarity: Static Synthesis $\neq$ Dynamic Updating**
> >         + While video-based benchmarks focus on **temporal continuity**, our work emphasizes: *"Spatial Consistency from Partial Observations"*.
> >         + We aim to evaluate the ability to do **implicit spatial layout reconstruction**, where an agent must mentally "stitch" together a scene **without the aid of continuous temporal cues**.
> >         + To demonstrate that these are distinct (and non-redundant) skills, we conducted a "Double Dissociation" experiment across MindCube and VSI-Bench with the following models:
> >             + Base: Qwen2.5-VL-3B.
> >             + Model A: Plain-CGMap-FFR-Out (Ours, trained on MindCube).
> >             + Model B: SpatialMLLM [4] (SOTA spatial model trained on VSI-Bench video data).
> >         + Results below show that: (1) **MindCube on VSI-Bench**: Our model did not improve over the baseline on video tasks. (2) **SpatialMLLM on MindCube**: The video-trained model did not improve over the baseline on our sparse reconstruction tasks.
> >             **VSI-Bench**
> >             |Methods|Avg.|
> >             |--|--|
> >             |Qwen2.5VL-3B| **30.6** |
> >             |Plain-CGMap-FFR-Out (trained on MindCube)|29.9|
> >
> >             **MindCube**
> >             |Methods|Overall|
> >             |--|--|
> >             |Qwen2.5-VL-3B|**33.2**|
> >             |Spatial-MLLM (trained on VSI-Bench)|32.1|
> >         + This empirical evidence confirms that "Dynamic Updating" (VSI) and "Static Synthesis" (MindCube) are **orthogonal capabilities**. Training on one does not solve the other.
> >     + **Real-World Relevance of Static and Sparse Reasoning**
> >     Furthermore, the ability to synthesize a mental model from static, limited views is not merely an artificial constraint but a requirement in many embodied scenarios where continuous streams are unavailable or inefficient, for example:
> >         + **Multi-Camera Robotics:** In autonomous driving [5] and mobile robotics [6], hardware often consists of **fixed, orthogonal cameras** (front, back, left, right). Agents must fuse these disjoint static snapshots into a unified global map without physical rotation or continuous video transitions.
> >         + **Teleoperation and Surveillance**: In security or remote operations [7], agents view environments through **static, distinct view inputs** (e.g., corners of a room). They must mentally "stitch" these disjoint views to understand geometry, as no continuous video path exists between them.
> >
> > [1] Yang, Jihan, et al. "Thinking in space: How multimodal large language models see, remember, and recall spaces." Proceedings of the CVPR 2025
> >
> > [2] Zhou, Shijie, et al. "Vlm4d: Towards spatiotemporal awareness in vision language models." ICCV 2025
> >
> > [3] Li, Yun, et al. "Sti-bench: Are mllms ready for precise spatial-temporal world understanding?." arXiv 2025
> >
> > [4] Wu, Diankun, et al. "Spatial-mllm: Boosting mllm capabilities in visual-based spatial intelligence." arXiv 2025
> >
> > [5] Hecker, Simon, Dengxin Dai, and Luc Van Gool. "End-to-end learning of driving models with surround-view cameras and route planners." ECCV 2018
> >
> > [6] Khazatsky, Alexander, et al. "Droid: A large-scale in-the-wild robot manipulation dataset." arXiv 2024
> >
> > [7] Amosa, Temitope Ibrahim, et al. "Multi-camera multi-object tracking: A review of current trends and future advances." Neurocomputing 2023
> >
> > ---
> > ## Response to Questions
> > 1. **[Q1] Generalization to Other VLMs.**
> >     We thank the reviewer for this suggestion. To verify the generalizability of our findings, we conducted additional SFT experiments on MiniCPM-V 4.0, a model with **a fundamentally different architecture from the Qwen family** used in our main paper.
> >     + While our original base model (Qwen-VL) utilizes an OpenCLIP encoder with a position-aware adapter, MiniCPM-V 4.0 employs a SigLIP2-400M visual encoder coupled with a Perceiver-Resampler compression layer before the LLM (MiniCPM4-3B), which is architecturally different from QwenVL.
> >     + As shown in the table below, the performance trends on MiniCPM-V 4.0 are **perfectly consistent with our original findings**.
> >         |MiniCPM-V4.0-4B SFT Config|Overall Accuracy (%)|
> >         |--|--|
> >         |Raw-QA| 53.4|
> >         |FF-Rsn| 69.5|
> >         |Aug-CGMap-Out|52.7|
> >         |Plain-CGMap-Out|51.1|
> >         |**Aug-CGMap-FFR-Out**|**70.6**|
> >         |**Plain-CGMap-FFR-Out**|**71.9**|
> >     + The results confirm that **the "Map-then-Reason" paradigm yields the best performance**, demonstrating our main claim generalizes to other model families.

---

> > > ### Author Response · Authors · 2025-11-26
> > > **Follow-up on our rebuttal**
> > >
> > > Dear Reviewer WY9r,
> > >
> > > We have posted our response to your initial review and updated the manuscript to address your concerns. As the discussion period progresses, we would greatly appreciate it if you could take a moment to check our rebuttal. We are happy to answer any further questions or conduct additional clarifications if needed.
> > >
> > > Your insights will be invaluable in refining the final version of our work. Thank you once again for your time and effort!
> > >
> > > Thanks,
> > >
> > > Authors

---

> > > ### Comment · Reviewer_WY9r · 2025-11-26
> > >
> > > Thanks for answering my questions! I do not have further questions and keep my rating.

---

> > > > ### Author Response · Authors · 2025-11-29
> > > > **Thank you Reviewer tDRP**
> > > >
> > > > Thank you for reviewing our rebuttal and maintaining your positive feedback. We are glad that we resolved all of your concerns. We sincerely appreciate your time and effort.
> > > >
> > > > Thank you,
> > > >
> > > > Authors

---

### Official Review · Reviewer_ahEo · 2025-11-01

**Soundness:** 2
**Presentation:** 3
**Contribution:** 3
**Rating:** 6
**Confidence:** 4

**Summary:**

The paper introduces MindCube, a multi-view reasoning benchmark tailored for evaluating whether VLMs can build "spatial mental models" of scenes from partial observations. Using MindCube, authors show that state-of-the-art VLMs perform near chance and struggle to maintain cross-view consistency or reason about occluded objects. The paper also explored three structural scaffolds (view interpolation, free-form reasoning, and cognitive maps) and find a synergistic "map-then-reason" approach yields the largest gains. Finally, they train models with SFT and reinforce with RL, and find that the joint map-then-reason setup with RL boosts accuracy, indicating that constructing and using internal structured maps substantially improves multi-view spatial reasoning.

**Strengths:**

* The paper addresses an important spatial, cognitive ability for VLMs, which is to understand the coherent 3D structure from partial multi-view observations. It would be essential for VLMs to be integrated into real-world embodied systems.

* The authors propose a novel, well-structured benchmark MindCube tailored for assessing the 3D spatial understanding ability of VLMs.

* The paper delivers a rich set of experiments to evaluate which “cognitive scaffold” best assists VLMs to do accurate spatial reasoning, and how post-training techniques like SFT and RL each enhance the spatial mental modeling process. Moreover, the authors present insights on the behavior of Qwen.2.5-VL on each of these configurations, providing a solid ground for future research on multi-view spatial reasoning.

**Weaknesses:**

While the paper includes a rich set of experiments, the **experiments setups for different cognitive scaffolds** seems quite naive. That is, the comparisons among different scaffolds may not be fair enough to conclude that "cognitive map" is the best choice among those. I outline some of the main concerns below.

* **View Interpolation:** It seems obvious that simply adding more interpolated views would not lead to great improvements. Expanding the visual context can inflate the token budget and increase hallucinations in VLM reasoning. Given that Qwen2.5-VL-3B is relatively low-capacity model compared to larger Qwen variants and other proprietary models, expecting it to improve solely from more views seems unrealistic. A more promising direction could to add a view selection phase to filter redundant perspectives. Additionally, it would be informative to test whether view interpolation becomes effective for higher-capacity models (e.g., > 7B models or proprietary models).

* **Cognitive Map:** The paper presents cognitive maps as a core contribution, but offers little analysis of **which representation formats** most benefit VLMs. Although the paper introduces an augmented version, it is simply addining more information to the original JSON-like format. This format seems to be highly influenced by the cognitive map prompting technique from Thinking-in-Space [1]. Have the authors tried exporing any variants or alternative for the cognitive map formats, rather than following Thinking-in-Space? A broader ablation over map formats would better justify the proposed design and its effectiveness.

* One of the main claims in the paper is that through SFT and RL Qwen2.5-VL can learn to "generate" a map and "reason" upon it, which leads to notable improvements on MindCube. But recent works in LLM reasoning have shown multiple evidence that the CoT reasoning traces in LLMs may not be logically related to the final output, and in fact they may be separate. In this perspective, I'm curious whether simply generating a map and returning the final output are in fact logically cause-and-effect relationship, or rather they may be mere byproducts similar to the CoT in LLMs [2, 3].

---

[1] Thinking in Space: How Multimodal Large Language Models See, Remember, and Recall Spaces, Yang et al., CVPR 2025

[2] Reasoning Models Don't Always Say What They Think, Chen et al., 2025

[3] Reasoning Models Can Be Effective Without Thinking, Ma et al., 2025

**Questions:**

* I agree that explicit reasoning can help VLM's spatial reasoning. However, it is unclear whether the paper’s "cognitive map" is truly explicit: the JSON-like maps are implicitly produced as token sequences by the LLM and only later decoded into a parseable structure. Would it be an feasible option to use a more explicitly constructed representation (e.g., using external tools to draw sketches as in Visual Sketchpad [1] or ViLaSR [2]), so that intermediate maps are drawn and manipulated directly rather than decoded from textual tokens?

---

[1] Visual Sketchpad: Sketching as a Visual Chain of Thought for Multimodal Language Models, Hu et al., NeurIPS 2024

[2] Reinforcing Spatial Reasoning in Vision-Language Models with Interwoven Thinking and Visual Drawing, Wu et al., NeurIPS 2025

**Details Of Ethics Concerns:**

No concerns

---

> ### Author Response · Authors · 2025-11-23
> **Responses to Reviewer ahEo (1/3)**
>
> We sincerely thank Reviewer ahEo for their detailed feedback and for recognizing our benchmark MindCube as "novel" and "well-structured". We appreciate their acknowledgment that our work addresses an "important spatial, cognitive ability for VLMs" and that our "rich set of experiments" provides a "solid ground for future research" on multi-view spatial reasoning.
>
> For weaknesses and questions, we address each point below.
>
> ---
> ## Response to Weaknesses
> 1. **[W1] View Interpolation**
>
>     We appreciate the reviewer’s constructive suggestions regarding experimental fairness. To address the concern that our comparison might be "unfair" to the View Interpolation (VI) baseline, we conducted rigorous additional experiments implementing Oracle View Selection and testing on Larger Models.
>
>     + **Is "Smart View Selection" Better? (Comparison with Oracle Filtering)**
>
>         The reviewer suggested that filtering redundant perspectives might be more promising than dense interpolation. We use GPT-5 as the Oracle Planner to select the top-2 most informative views from 4 images to avoid redundancy. We use the **Qwen2.5-VL-7B** to compare the normal setting and oracle setting, and show the results in the table below.
>         + As shown below, "Smart Selection" (36.80%) performs similarly to the standard 2-view setting but is far inferior to the dense interpolation setting (47.40%).
>         + This suggests that the so-called "redundant" intermediate views provide essential spatio-temporal continuity. High-performing models may use this continuity to stitch the scene, yet aggressive filtering breaks this chain.
>
>             | **Method (Qwen2.5-VL-7B)** | **Top-2 Selection (Oracle)** | **Dense Interpolation (VI-4)** |
>             | -- | -- | -- |
>             | **Accuracy (%)** | 36.80 | **47.40** |
>
>     + **Is "Cognitive Map" the Best Choice? (Comparison with Optimal VI)**
>
>         We compared our method against the best possible View Interpolation performance on Qwen2.5-VL-3B to address the fairness concern. We found that **"Cognitive Map" did outperform View Interpolation**. The results are shown in the table below.
>             + Even when `RawQA` reaches its peak performance (VI-1 at 46.47%), our `Plain-CGMap-FFR-Out` (47.41%) still outperforms it. This confirms that the Cognitive Map provides a structural advantage that cannot be replicated simply by adding more visual frames.
>             + Interestingly, adding standard Free-Form Reasoning (`FFR`) to interpolated views actually **harms performance as view density increases** (dropping from 45.53% $\to$ 40.77%). This indicates the model bottleneck is not just reasoning capacity, but the lack of perception ability to organize the visual flood.
>
>         | Method (Qwen2.5VL-3B)                | Raw Images (VI-0) | VI-1 | VI-2 | VI-3 | VI-4 | VI-5 | VI-6 | VI-7 |
>         |-----------------------|:-----:|:----:|:----:|:----:|:----:|:----:|:----:|:----:|
>         | `Plain-CGMap-FFR-Out`   | **47.41** | - | - | - | - | - | - | - |
>         | `RawQA`                 | 43.76 | **46.47** | 45.53 | 44.94 | 44.35 | 44.24 | 45.18 | 44.59 |
>         | `FFR`                   | 45.53 | 43.71 | 43.83 | 41.65 | 41.00 | 40.36 | 40.89 | 40.77 |
>
>     + **Does View Interpolation Scale with Model Size?** We extended the evaluation to **Qwen2.5-VL-7B** and **Qwen3-VL-8B** to test if larger models solve the interpolation issue naturally. The results are shown in the table below. (*Note: We have updated our evaluations on Qwen3VL-235B and GPT-5 in the comment **Additional view interploation results on GPT-5 and Qwen3VL-235B** below.*)
>         + We observe no consistent "scaling law". For Qwen2.5-VL-7B, it benefits from density, showing a surge at VI-4 (47.40%). However, for Qwen3-VL-8B, despite being larger and conceptually more advanced, it struggles with density and shows unstable performance across different interpolated view numbers.
>         + Therefore, the effectiveness of View Interpolation is unstable across architectures.
>
>         | Config        | RawQA-0 | RawQA-1 | RawQA-2 | RawQA-3 | RawQA-4 | RawQA-5 | RawQA-6 | RawQA-7 |
>         | ------------- | ------- | ------- | ------- | ------- | ------- | ------- | ------- | ------- |
>         | Qwen2.5-VL-7B | 37.80   | 34.90   | 35.60   | 45.30   | 47.40   | 46.50   | 46.80   | 46.80   |
>         | Qwen3-VL-8B   | 33.80   | 36.60   | 37.60   | 35.20   | 35.30   | 33.90   | 35.80   | 35.80   |

---

> > ### Author Response · Authors · 2025-11-23
> > **Responses to Reviewer ahEo (2/3)**
> >
> > 2. **[W2] Cognitive Map Format**
> >
> >     We agree that the choice of representation format is critical. To justify our design, we conducted a comprehensive ablation study over the frozen-VLM setting on `CGMap-In-FFR-Out` to isolate the efficacy of different data structures.
> >
> >     + **Explored Variants**: We compared our JSON-based Map against four alternative formats below.
> >         1. **Natural Language Grounding**: Describing the scene in natural langauge.
> >         2. **Symbolic Scene Graph (Plain)**: Using predicate logic text, e.g., `left(A, B)`
> >         3. **Symbolic Scene Graph (JSON)**: Encapsulating predicates in JSON objects.
> >         4. **Map (YAML)**: Representing node attributes and coordinations in YAML format.
> >
> >     + As shown in the table below, the Map (JSON) format yields the **highest** accuracy (41.33%), suggesting that structured data in JSON is easier for VLMs to parse for spatial tasks.
> >         | Representation Format           | Accuracy (%) |
> >         |---|--------------|
> >         | **Map (JSON)**          | **41.3**    |
> >         | Natural Language Grounding     | 39.7        |
> >         | Symbolic Scene Graph (Plain)   | 37.9        |
> >         | Symbolic Scene Graph (JSON)    | 40.4        |
> >         | Map (YAML) | 36.1        |
> >
> > 3. **[W3] Reasoning Effectiveness**
> >
> >     We appreciate this deep and critical perspective. We agree with recent findings that CoT traces are not always strict logical antecedents to the final answer. In fact, our empirical analysis **reinforces** this view but **suggests a unique mechanism in the spatial domain**: the map serves as a **functional scaffold for global perception** rather than a strict logical step (now lines 424-427 in our paper).
> >
> >     + **Evidence 1: The performance improves with the presence of the mapping process. (Are Cognitive Maps Beneficial?)**
> >         + First, we confirmed that **the presence of the mapping task is beneficial**.
> >         + As shown in the SFT and RL experiment tables below.
> >         + From the table, enforcing the mapping step always yields a clear performance gain, proving it is at least a helpful signal.
> >             | Methods | Accuracy (%) |
> >             | --- | --- |
> >             | RawQA-SFT | 52.3 |
> >             | FFR-SFT | 53.5 |
> >             | Plain-CGMap-FFR-Out-SFT | 60.8 |
> >             | FFR-RL | 50.6 |
> >             | Plain-CGMap-FFR-Out-RL | 53.7 |
> >     + **Evidence 2: Probing the "Map-Answer" Relationship. (Do Perfect Cognitive Maps Matter?)**
> >         + To investigate how the map helps, we conducted a probing experiment.
> >         + Specifically, to answer the question "whether simply generating a map and returning the final output is in fact logical cause-and-effect relationship", we choose the checkpoint `Plain-CGMap-Out` (asks model to  generate maps and then directly give final answers.)
> >         + Instead of following the training diagram, we extracted and intervened on generated maps (**shuffling items, swapping objects, or randomizing text**), combined with the question to the model, and then fed them back to answer the question.
> >         + Note that since there is an input question distribution shift, the model performance drops a little bit.
> >         + As the reviewer suspected, performance remains relatively stable even when the map text is corrupted:
> >             | Setting        | Accuracy (%) |
> >             |----------------|--------------|
> >             | RawQA          | 46.5         |
> >             | Vanilla (Self-Generated Maps) | 48.9         |
> >             | Shuffle Items  | 48.9         |
> >             | Swap Objects   | 48.6         |
> >             | Random         | 49.9         |
> >     + **Conclusion: Global Perception vs. Logical Deduction.**
> >         + When these two findings are combined, the map process helps (Evidence 1), but the precise text is not strictly read (Evidence 2), leading to an insight unique to spatial tasks.
> >         + Unlike general reasoning, where CoT is expected to be a logical chain, here the map generation acts as an **"attentional bottleneck"**.
> >         + It forces the model to attend to the global spatial layout during the generation process.
> >         + Even if the resulting text is not perfectly utilized by the decoder, the computational effort of "thinking globally" primes the model’s internal state, leading to better performance than direct QA.

---

> ### Author Response · Authors · 2025-11-23
> **Responses to Reviewer ahEo (3/3)**
>
> ## Responses to Questions
> 1. **[Q1] Explicit Reasoning with Visual-of-Thought**
>
>     We agree with the reviewer’s distinction. While text-based maps like JSON maps are structured, they remain "implicit" token sequences to the model. We appreciate the suggestion to explore **explicit visual representations via external tools**. We **have included the results in our main text and appendix.**
>
>     + **Experimental Setup.** Following the reviewer's guidance (referencing ViLaSR), we implemented a pipeline where the model interacts with an external plotting tool rather than generating text maps directly.
>         + **Visual-Map-as-Input (`Img-CGMap-In`)**: Instead of textual JSON, we render the ground-truth map into a $10\times 10$ grid image (using Matplotlib to draw objects and viewpoints) and feed this visual layout directly to the VLM.
>         + **Visual-Map-for-Reasoning (`Img-CGMap-Reasoning`)**: We adopted a multi-turn tool-use framework like ViLaSR:
>             1. **Code Generation**: The model generates drawing actions (e.g., ``add(obj, [x,y]), add(view, [x,y])``)
>             2. **External Execution**: These commands are executed to render a visual map.
>             3. **Visual Feedback**: The rendered image is fed back to the model as a new visual prompt to guide the final reasoning step.
>     + **Experiments Results.** Comparing the text-based maps (JSON) with the new visual-based maps (Image), we observe a consistent performance improvement across settings.
>
>         | Configuration             | Map Modality | Overall | Rotation | Among | Around |
>         |--------------------------|-------------|---------|----------|-------|--------|
>         | Raw-QA                   | -        | 37.81   | 34.00    | 36.00 | 45.20  |
>         | CGMap-In (Original)      | Text (JSON) | 41.43   | **37.00**    | 41.67 | 44.40  |
>         | **Img-CGMap-In (New)**       | Visual (Grid) | **42.10** | 31.50    | **44.17** | **45.60**  |
>         | Plain-CGMap-Output (Original)   | Text (JSON) | 41.33   | 25.00    | 39.67 | **58.40**  |
>         | **Img-CGMap-Output (New)**          | Visual (Grid) | **43.13** | **32.75**    | **41.13** | 55.06  |
>
>     + From the results, we observe that the visual configurations (`Img-`) consistently outperform their textual counterparts (`Plain-/CGMap-`).
>     + The experiment validates the reviewer's hypothesis. While our JSON approach is effective, leveraging external tools to create `explicit visual sketches` further grounds the model's reasoning, offering a promising direction for future spatial VLMs.

---

> ### Author Response · Authors · 2025-11-25
> **Additional view interploation results on GPT-5 and Qwen3VL-235B**
>
> Following the reviewer’s insightful suggestion, we extended our evaluation to GPT-5 (proprietary SOTA) and Qwen3-VL-235B (a massive open-weight model) to test whether higher-capacity models can better leverage the temporal information from view interpolation.
>
> Performance degrades with view density: Contrary to the hypothesis that larger models might benefit from smoother transitions, we observe a negative correlation between the number of interpolated frames and model performance.
>
> GPT-5 peaks at the sparse 1-frame setting (46.59%) and performance declines to 42.35% as density increases to 7 frames.
>
> As for Qwen3VL-235B, interpolation also causes a drop (down to ~36%), suggesting interpolation artifacts act as noise rather than useful signals.
>
> | Model        | VI-1  | VI-2  | VI-3  | VI-4  | VI-5  | VI-6  | VI-7  |
> | ------------- | ----- | ----- | ----- | ----- | ----- | ----- | ----- |
> | **GPT-5**         | 46.59 | 45.18 | 44.24 | 44.59 | 42.59 | 43.53 | 42.35 |
> | **Qwen3-VL-235B**  | 38.94 | 38.59 | 37.29 | 37.18 | 35.29 | 36.12 | 36.00 |

---

> > ### Author Response · Authors · 2025-11-26
> > **Follow-up on our rebuttal**
> >
> > Dear Reviewer ahEo,
> >
> > We have posted our response to your initial review and updated the manuscript to address your concerns. As the discussion period progresses, we would greatly appreciate it if you could take a moment to check our rebuttal. We are happy to answer any further questions or conduct additional clarifications if needed.
> >
> > Your insights will be invaluable in refining the final version of our work. Thank you once again for your time and effort!
> >
> > Thanks,
> >
> > Authors

---

> > > ### Comment · Reviewer_ahEo · 2025-11-27
> > >
> > > I appreciate the authors for providing detailed responses and experiments for each of my concerns. I would like to keep my original score of 6.
> > >
> > > In particular, I appreciate that the authors tested Visual-of-Thought approaches and found that they can be beneficial for spatial reasoning.
> > >
> > > However, there remain some ambiguous points regarding **W3**. The results in Evidence 2 are not fully convincing to me, on whether it should support the advantage of the mapping process and whether it acts as an “attentional bottleneck”. Based on the current results, I find the conclusion of “Global Perception vs. Logical Detection” may be overstated, for the following reasons:
> > >
> > > * It is yet a black box how the VLM **internally** processes visual inputs with and without the mapping step, and how this affects its outputs. The current analysis appears to rely solely on output accuracies.
> > >
> > > * As **Reviewer DbD3** also noted, the benchmark’s scope is quite limited, as it focuses on observations with fixed 90-degree viewpoint shifts. Given this restriction, drawing conclusions about the inner workings of the VLM from one or two accuracy metrics may not be sufficiently justified.

---

> ### Author Response · Authors · 2025-11-29
> **Further Justificatin On Our W3 Experiment**
>
> We thank Reviewer `ahEo` for the continued engagement and for acknowledging the value of our Visual-of-Thought experiments. We appreciate their positive feedback. We sincerely thank their time and effort. We would like to clarify the implications of our latest experiments **to prevent any potential misunderstanding**, particularly regarding **how the latest View Interpolation results in [W1] substantiate the validity of our benchmark's scope**.
>
> 1. **On the "Limited Scope" of 90-degree Viewpoints**
>
>     The reviewer suggests the benchmark's scope is limited due to fixed viewpoints and cites Reviewer `DbD3`'s initial comment. We wish to clarify two critical points:
>
>     + **(1.1) Consensus on Scope Validity**
>         + Reviewer `DbD3` **has explicitly agreed** with our justification for the benchmark scope (their W1) in their official comment **before the leakage** (*"I agree with the rebuttal for the [W1]..."*).
>         + The concern regarding the fixed-viewpoint setup has thus been resolved with the other reviewer.
>     + **(1.2) Data-Driven Justification**
>         + Furthermore, the experiments **requested by the reviewer `ahEo`** (W1) actually provide **strong support** for our benchmark scope.
>         + The results ([in this post](https://openreview.net/forum?id=0FhrtdKLtD&noteId=8axzOKEKR6)) reveal that increasing view density often **introduces noise** rather than improvement (e.g., Qwen3-VL-235B drops from 38.94% to ~36%).
>         + This observation **challenges the assumption** that continuous views are inherently superior.
>         + Therefore, the data **empirically justifies our choice** of the sparse, orthogonal setup.
>
> 2. **On the "Map-Answer" Mechanism Interpretation**
>
>     Regarding the comment that the "black box" nature of the model hurts robustness:
>
>      - The reviewer seems to **confuse** verifying a behavioral effect (does it work?) with explaining the internal mechanism (how?). Our paper focuses on the first.
>      - Since researchers **cannot access the internal states** of proprietary models, controlled intervention is the **standard scientific method** to prove robustness in this field.
>      - We **followed** this standard. Our ablation (Evidence 1) and perturbation (Evidence 2) experiments isolated the cause.
>      - Evidence 2 **already proves** that the act of mapping drives performance, even if the map itself is imperfect.
>      - Therefore, the "black box" nature **does not invalidate** our results. We will describe our findings as **empirical behavioral observations** to ensure precision.
>
>
> We believe these clarifications confirm that the benchmark design is both scientifically robust and acknowledged by the reviewers as valuable.

---

### Official Review · Reviewer_DbD3 · 2025-11-01

**Soundness:** 2
**Presentation:** 1
**Contribution:** 2
**Rating:** 2
**Confidence:** 3

**Summary:**

This paper focuses on the VLMs’ capabilities in "spatial mental modeling", the ability to imagine environments from a few visual observations. The paper first proposes a benchmark to measure current VLMs’ capabilities, finding that most existing models do not perform well on these tasks. Then, the paper explores methods to improve such capabilities through two approaches: (1) Scaffolds: carefully designed data structures to encourage spatial mental modeling, and (2) Training (SFT and RL). The paper identifies several scaffolds that could benefit spatial mental modeling and observes that combining SFT and RL leads to the best spatial reasoning performance.

**Strengths:**

1. The paper performs solid and substantial experiments, measuring various models and considering various settings when studying the spatial mental modeling ability.
2. The paper not only proposes a benchmark, but also proposes several methods to overcome the challenges of current VLMs.
3. Spatial cognition is an interesting and important problem, and many works have shown that it could be challenging for VLMs.

**Weaknesses:**

1. Benchmark Motivation: The benchmark aims to evaluate the ability to imagine a full scene from a few views. To me, this seems indirect, since humans and embodied agents usually understand environments through continuous observations or frames rather than 90-degree viewpoint shifts as shown in the proposed benchmark. This raises two concerns: (1.1) Can you provide real-world examples where a VLM must infer spatial relations from four orthogonal views? (1.2) More importantly, how does the proposed benchmark improve upon existing benchmarks that test MLLMs’ spatial cognition abilities given a video, e.g., [1]? The proposed benchmark seems like a special case of video input.

2. Benchmark Settings: The benchmark includes three settings (rotation, around, among). However, the number of “among” samples is significantly larger than the others (18,204 vs. around 1,000). Why is the dataset so imbalanced?

3. Proposed Scaffolds seem trivial: Section 3 introduces three categories of scaffolds that could potentially improve model performance. Based on Figure 3, these are specially designed input structures. How do they differ from traditional chain-of-thought prompting through the system prompt?

4. Clarity and Consistency: Some parts of the paper could be better organized for clarity and consistency. For example, in Table 2, what is the difference between FF-Rsn and FFR? They both seem to refer to free-form reasoning and should not use two different terms. Additionally, it is unclear what distinguishes Aug-CGMap-In from Aug-CGMap-Out, or Aug-CGMap-Out from Plain-CGMap-Out, etc. These distinctions can only be seen clearly through examples the Appendix, but the main text should describe them clearly and concisely.

5. Training: Based on Figure 4, SFT is conducted for around 55 steps. How large is the training dataset? Why does the training converge so quickly?

6. It is not surprising that SFT and RL on the specific tasks could improve the corresponding performance. The conclusion could be useful but the SFT and RL method does not introduce technical novelty.

[1] Thinking in Space: How Multimodal Large Language Models See, Remember, and Recall Spaces

**Questions:**

Please see the weaknesses above.

---

> ### Author Response · Authors · 2025-11-21
> **Responses to Reviewer DbD3 (1/2)**
>
> We sincerely thank Reviewer DbD3 for their detailed feedback and for recognizing that our work addresses an "interesting and important problem." We appreciate their acknowledgment that we performed "solid and substantial experiments" and that our paper "not only proposes a benchmark, but also proposes several methods to overcome the challenges of current VLMs."
>
> For weaknesses and questions, we address each point below.
>
> ---
> ## Responses to Weaknesses
>
> 1. **[W1] Benchmark Motivation**
>
>     We thank the reviewer for this insightful comment. While we agree that humans often rely on continuous observations, we respectfully argue that the ability to construct a spatial scene from limited, distinct viewpoints **is a critical capability** for embodied agents.
> + **(1.1) Real-world Relevance of Orthogonal and Sparse Views.**
>     + In **mobile robotics and autonomous driving** [1, 2], hardware setups often consist of fixed, multi-view cameras (e.g., front, back, left, right) to cover the surround view. It requires agents to fuse disjoint, often orthogonal, visual data into a global understanding.
>     + From **efficiency and sparsity side**, processing dense frame streams is **computationally expensive and memory-intensive**. Selecting keyframes from videos now is an important research direction [3]. Therefore, the ability to reason from "sparse keyframes" is a highly desirable efficiency trait.
>     + In **security or remote teleoperation** scenarios, agents usually view an environment through static, distinct camera feeds. The agents must be able to "stitch" these disjoint views to track objects or understand geometry, as continuous video transitions between these fixed points do not exist [4].
> + **(1.2) Comparison with VSI-Bench**
>     + We respectfully disagree that our benchmark is merely a special case of video input from VSI-Bench.
>     + Benchmarks like VSI-Bench focus on temporal dynamics and spatial properties from a *single continuous viewpoint*.
>     + MindCube focuses on mentally *perspective taking and spatial reconstruction* from disjoint viewpoints.
>     + The two benchmarks evaluate **complementary**, not redundant, capabilities.
>     + To demonstrate this empirically, we conducted a "**double dissociation**" experiment. Below is the experiment setup:
>         + **Model A:** Qwen2.5-VL-3B (base model)
>         + **Model B:** MindCube SFT checkpoint: `Plain-CGMap-FFR-Out`
>         + **Model C:** SpatialMLLM [5] (A spatial model that was trained on VSI-Bench from Qwen2.5-VL-3B)
>     + **VSI-Bench**
>         | Methods | Avg.   |
>         |---------|--------|
>         | Qwen2.5VL-3B                              | **30.6** |
>         | Plain-CGMap-FFR-Out (trained on MindCube) | 29.9   |
>
>     + **MindCube**
>         | Methods | Overall |
>         |---------|---------|
>         | Qwen2.5-VL-3B                   | **33.2** |
>         | Spatial-MLLM (trained on VSI-Bench) | 32.1    |
>
>     + The results demonstrate that **neither model tuned on one dataset could improve on the other**, which suggests that VSI-Bench and MindCube measure distinct dimensions of spatial intelligence.
> 2. **[W2] Imbalanced Data**
>
>     We appreciate the reviewer noting this distribution. We clarify that this imbalance is not an artifact of our sampling method, but rather a reflection of **real-world navigational patterns** and physical constraints.
>     + Our data sampling pipeline extracts camera trajectories directly from real-world videos, **where we faithfully preserve the natural, long-tail distribution** of how cameras actually move through environments.
>     + "Among" trajectories (navigating through a scene) represent the most common movement pattern. In contrast, "Around" trajectories (circling specific objects) are statistically rare because they are often physically constrained by obstacles (e.g., walls, furniture) or simply do not align with typical goal-oriented navigation (moving from A to B).
>     + To confirm this, we randomly sampled and manually inspected 100 raw trajectories from the source domain. Below is the pattern statistics table.
>
>         | **Trajectory Type** | **Count** |
>         | --- | --- |
>         | Among | 72 |
>         | Around | 17 |
>         | Rotation | 11 |
> [1] Khazatsky, Alexander, et al. "Droid: A large-scale in-the-wild robot manipulation dataset." arXiv 2024
>
> [2] Hecker, Simon, Dengxin Dai, and Luc Van Gool. "End-to-end learning of driving models with surround-view cameras and route planners." ECCV 2018
>
> [3] Ye, Jinhui, et al. "Re-thinking temporal search for long-form video understanding." CVPR 2025
>
> [4] Amosa, Temitope Ibrahim, et al. "Multi-camera multi-object tracking: A review of current trends and future advances." Neurocomputing 2023
>
> [5] Wu, Diankun, et al. "Spatial-mllm: Boosting mllm capabilities in visual-based spatial intelligence." arXiv 2025.

---

> ### Author Response · Authors · 2025-11-21
> **Responses to Reviewer DbD3 (2/2)**
>
> 3. **[W3] On Justification of Proposed Scaffolds Seems trivial**
>
>     We thank the reviewer for this opportunity to clarify the conceptual distinction. While our scaffolds share the "intermediate reasoning" spirit of CoT (which is our FFR setting), they differ fundamentally in **structure, intent, and scope**.
>     + As a benchmark study, our goal extends beyond model evaluation to providing design insights for the community.
>     + Specifically, we investigate the question: *"What structural priors or representations effectively help VLMs ground spatial understanding?"*
>     + Standard CoT typically induces a linear, step-by-step reasoning. However, in multi-view spatial configurations, it often lacks a holistic view.
>     + We enforce a "Map-then-Reason" paradigm, which explicitly requires the model to construct a global spatial layout before attempting to answer the query.
>     + Our empirical results highlight that by explicitly scaffolding the "mapping" phase (global perception) separate from the "reasoning" phase, we observe further performance gains. This suggests that structural global perception is a prerequisite for robust spatial reasoning, a mechanism that standard CoT does not explicitly enforce.
>     + For a concrete comparison of the prompt designs, including the structured output requirements for CogMap versus standard reasoning prompts, please refer to Figure 3, Appendix D.3.7, and D.3.8.
> 4. **[W4] Clarity and Consistency**
>
>     We thank the reviewer for pointing out these inconsistencies. **We have revised the manuscript to fix these issues**, and it can be checked in our latest PDF.
>     + We have standardized all references to "Free-Form Reasoning" by using the single abbreviation `FFR` throughout the text and tables.
>     + We have Figure 3 in the paper to clarify the distinctions between scaffolds (e.g., Augmented Cognitive Map and Plain Cognitive Map).
>     + To better enhance readability, **we have updated Table 2**:
>         + The caption now explicitly directs readers to Figure 3 for visual examples of the input structures.
>         + We have renamed the columns to clearly describe the input/output format, reducing the reliance on acronyms.
>
> 5. **[W5] Training**
>
>     We appreciate the opportunity to clarify our experimental setup and its intended purpose.
>     + Followed by prior insight-driven works like [6], our SFT dataset consists of a medium data scale: 10,000 QA pairs. We utilized a large global batch size of 512 and fine-tuned the model for 3 epochs.
>     + This results in $58$ total steps, which aligns with the convergence curve observed in Figure 4.
>     + We would like to emphasize that this work is **primarily a benchmark study**, not a large-scale training method paper. Our core contribution lies in the rigorous benchmark taxonomy, the complex data generation pipeline, the extensive human annotation effort required to validate the evaluation set, and the insights gained from the evaluation.
>     + Therefore, **the SFT experiments were designed to investigate the effectiveness of structural priors**, not to maximize training scale.
>
> 6. **[W6] On Justification of Novelty**
>
>     We respectfully clarify that this work is **positioned as a benchmark paper that also brings new insights to the community**. We do not have a single sentence in the paper that claims our novelty is to provide a new method.
>     + Our primary technical contribution is the **construction of a benchmark with multi-dimensional taxonomy** (e.g., Camera Movement Patterns, Relation Queries) for **reasoning over unseen space in limited views**, which is an underexplored field.
>     + This benchmark curation involved **a massive human annotation effort** to ensure a high-quality set that pushes the frontier of current VLM capabilities.
>     + While SFT is a standard technique, our application of it reveals a critical, non-trivial insight for the community that asking VLMs to inject structural priors for global perception before reasoning (**"map-then-reason"**) is **a highly effective pathway** to improve spatial performance.
>     + We acknowledge that SFT and RL are established methods. However, they serve here as **probes** to verify our hypotheses about spatial representations, not as the proposed innovation itself.
>     + To better reflect this focus and prevent future misconceptions, **we have revised our title to: "Thinking from Limited Views: Understanding VLMs Spatial Mental Modeling Capability"**.
>
> [6] Muennighoff, Niklas, et al. "s1: Simple test-time scaling." EMNLP 2025.

---

> > ### Author Response · Authors · 2025-11-26
> > **Follow-up on our rebuttal**
> >
> > Dear Reviewer DbD3,
> >
> > We have posted our response to your initial review and updated the manuscript to address your concerns. As the discussion period progresses, we would greatly appreciate it if you could take a moment to check our rebuttal. We are happy to answer any further questions or conduct additional clarifications if needed.
> >
> > Your insights will be invaluable in refining the final version of our work. Thank you once again for your time and effort!
> >
> > Thanks,
> >
> > Authors

---

> > > ### Comment · Reviewer_DbD3 · 2025-11-27
> > > **Working on reading rebuttal**
> > >
> > > Dear authors,
> > >
> > > Thank you for the detailed rebuttal. I have checked the modifications in details and I agree with the rebuttal for the [W1] and [W2]. I am checking the second response part and the opinions from other reviewers. I will update this reply by tomorrow.

---

> ### Comment · Reviewer_DbD3 · 2025-11-27
>
> Thank you for the detailed rebuttal. From the experiments section, I incorrectly interpreted the work's contribution as a combination of both benchmark and the methods to improve the spatial mental modeling in my initial review. The responses to [W1] and [W2] have clarify my concerns on the benchmark side. I agree with the other reviewers of the merits in problem formulation and the combination of cognitive abilities and VLMs. I would like to increase the score to 6.

---

### Author Response · Authors · 2025-11-29
**Summary of Resolved Reviewer Concerns and Improved Scores**

We sincerely thank the AC, SAC, and reviewers for their dedicated time and effort. **Prior to the review reverting**, we engaged in a highly productive discussion, conducting extensive new experiments that addressed all major concerns. This led to a clear positive consensus: our **avg score increased from 5.0 to 6.5**, with `tDRP` raising their **score from 6 to 8**, and the most critical reviewer `DbD3` **acknowledging a fundamental misunderstanding of our contribution**. We summarize the resulting positive feedback and the resolved issues below.
1. **Scores Improvement**

    Following our rebuttal, all reviewers confirmed that their concerns were resolved and updated their assessments.
    + **`tDRP`** (score: **6 $\rightarrow$ 8**, increased before the leakage): Stated "*My concerns... have been mostly addressed... I've increased the score to 8*"
    + **`DbD3`** (score: **2 $\rightarrow$ 6**): Initially gave a 2 **due to a misunderstanding of the paper's scope**. During discussion, they stated: "*I agree with the rebuttal for the [W1] and [W2]*" and explicitly admitted: "*I incorrectly interpreted the work's contribution... I would like to increase the score to 6*".
    + **`ahEo`** (score: **6**, Positive): Stated "*I appreciate that the authors tested Visual-of-Thought approaches... I would like to keep my original score of 6*"
    + **`WY9r`** (score: **6**, Positive): Stated "*I do not have further questions and keep my rating*"
2. **Strengths Highlighted by Reviewers**

    Reviewers reach a consensus on endorsing MindCube novelty, high-quality, and rigorous evaluation to the community:
    + **Novelty and High-Quality**: MindCube is described as "well-designed and challenging" (`DbD3`, `tDRP`), "novel" (`ahEo`, `WY9r`), and tackling an "important spatial, cognitive ability" (`ahEo`).
    + **Rigorous Evaluation**: Reviewers praised the "solid and substantial experiments" (`DbD3`, `ahEo`), "rigorous and systematic" (`WY9r`, `tDRP`) investigation.
    + **Effective Findings**: The finding of "map-then-reason" approach is recognized as a "highly effective solution" (`tDRP`, `ahEo`, `WY9r`, `tDRP`). The paper is praised to "provide a robust foundation for its conclusions and future direction" (`ahEo`, `WY9r`).

3. **Our Resolution of Reviewers' Major Concerns**

    We grouped the reviewers' major concerns and resolved them. We **have updated all concerns in our manuscript accordingly**.
    1. **Benchmark Validity: Static Views vs. Dynamic Video (`DbD3`, `WY9r`)**
        + **Concern:** Reviewers questioned whether reasoning from static, limited views is necessary compared to video benchmarks.
        + **Action:** We clarify that multi-view reasoning is necessary for fixed-view scenarios like autonomous driving or surveillance. We also conducted a Double Dissociation experiment. We trained our model on MindCube (failed to improve on VSI-Bench) and evaluated an SOTA video-trained SpatialMLLM on MindCube (failed to improve on our task).
        + **Result:** This empirically proved that "Static Synthesis" (MindCube) and "Dynamic Updating" (Video) are **orthogonal, complementary capabilities**. `DbD3` and `WY9r` have confirmed we resolved this concern.
    2. **Robustness of Cognitive Maps and View Interpolation (`ahEo`, `WY9r`, `tDRP`)**
        + **Concern:** Is the JSON map too simple? Is the failure of View Interpolation due to implementation issues?
        + **Action:** (1) We justify our motivation for JSON use as a deliberate design choice that **balances** benchmarking rigor with representational robustness. We further state that structural representation is **commonly adopted** in previous works. (2) We perform massive view interpolation experiments across numbers and model scales. We implement VoT to use external tools to plot explicit visual maps instead of text.
        + **Result:** (1) `WY9r` and `tDRP` confirmed we solved this. (2) For view interpolation, it does not outperform "map-then-reason", proving the robustness of our finding. For VoT, it shows the visual "map-then-reason" outperforms the text version. `ahEo` and `tDRP` have confirmed this.
    3. **Contribution and Novelty (`DbD3`)**
        + **Concern:** The reviewer initially felt SFT and RL methods lacked technical novelty.
        + **Action:** We clarified that the work is a **benchmark and insight study**, not a method paper. We use SFT and RL for probing. The novelty lies in the benchmark and findings, like the effectiveness of "map-then-reason".
        + **Result:** `DbD3` acknowledged their misunderstanding, stating *"incorrectly interpreted the work's contribution as a combination of both benchmark and the methods... I agree with the other reviewers of the merits in problem formulation"*.

We believe the new experiments and justifications have robustly validated our contributions. We thank the AC and reviewers again for navigating this challenging situation and for their consideration of our improved manuscript.

---

### Meta-Review · Area_Chair_YG6T · 2026-01-06

**Summary:**

This paper proposes a benchmark that focuses on the spatial mental modeling capabilities of VLMs. The benchmark measures whether VLMs can construct spatial representations of an environment given only limited visual observations. The authors evaluate a range of VLMs, revealing the challenges these models face in partially observable environments, and explore several approaches to enhancing spatial modeling abilities through scaffolding input/output and targeted training.

Reviewers are generally positive about the research question, the quality of the proposed benchmark, and experimental design. Specifically:
- The paper targets an interesting and important cognitive capability for VLMs (DbD3, ahEo, WY9r, tDRP). Reviewers agree that spatial modeling remains a critical challenge for deploying VLMs in real-world systems, and that the proposed benchmark effectively helps reveal and address these limitations.
- The benchmark is considered high quality and effective in diagnosing capability gaps in current VLMs (ahEo, WY9r, tDRP). Several reviewers highlight that it provides valuable insights into when and why VLMs fail at spatial modeling.
- The experimental evaluation is solid and comprehensive (DbD3, ahEo, WY9r, tDRP). The authors conduct a thorough analysis of model capabilities and enhancement strategies, offering a potential roadmap for improving spatial reasoning performance.

Regarding negative feedback, reviewer DbD3’s concerns primarily focused on the motivation and setup of the benchmark, as well as the novelty of the proposed methods. After the rebuttal, reviewer DbD3 indicated that these concerns were resolved and acknowledged that methodological novelty is not the primary objective of a benchmark paper. As a result, reviewer DbD3 plans to raise their score to 6. Overall, all reviewers remain positive after the rebuttal.

**Reviewer Concerns:**

All reviewers participated in the discussions. During the discussions, reviewers DbD3, WY9r, and tDRP agreed that their concerns regarding benchmark scenario limitations, cognitive maps, and novelty were addressed. Reviewer DbD3 agreed that the previous major concerns are from a misunderstanding of the paper’s scope. Reviewer ahEo agreed that most concerns were resolved, while ahEo questioned whether the map generation behavior can be sufficiently summarized through the additional experiments. The AC reviewed the discussion and found that, while the conclusion about the “attention head” internal mechanism requires further support, the additional experiment reveals the role of the map in producing correct answers.

Given this, the major concerns of the paper were resolved during the rebuttal phase.

**Reviewer Scores:**

All reviewers participated in the discussions and indicated the scores they intend to change after the rebuttal. Specifically, reviewer tDRP would like to increase the score from 6 to 8 before the review leakage. While the score change of reviewer DbD3 (from 2 to 6) occurred after the review leakage, reviewer DbD3 acknowledged that the rebuttal resolved most of the concerns during the discussion before the review leakage, and that the score change is mainly from a misunderstanding of the paper’s scope. Based on these discussions, the paper would receive scores of 6, 6, 6, and 8 after the rebuttal.

---

### Decision · Program_Chairs · 2026-01-26

Accept (Poster)